# Differential DNA methylation of vocal and facial anatomy genes in modern humans

David Gokhman et al.[#]

Changes in potential regulatory elements are thought to be key drivers of phenotypic divergence. However, identifying changes to regulatory elements that underlie human-specific traits has proven very challenging. Here, we use 63 reconstructed and experimentally measured DNA methylation maps of ancient and present-day humans, as well as of six chimpanzees, to detect differentially methylated regions that likely emerged in modern humans after the split from Neanderthals and Denisovans. We show that genes associated with face and vocal tract anatomy went through particularly extensive methylation changes. Specifically, we identify widespread hypermethylation in a network of face- and voice-associated genes (*SOX9*, *ACAN*, *COL2A1*, *NFIX* and *XYLT1*). We propose that these repression patterns appeared after the split from Neanderthals and Denisovans, and that they might have played a key role in shaping the modern human face and vocal tract.

[#]A full list of authors and their affiliations appears at the end of the paper.

The advent of high-coverage ancient genomes of archaic humans (Neanderthal and Denisovan) introduced the possibility to identify the genetic basis of some unique modern human traits[1]. A common approach is to carry out sequence comparisons and detect non-neutral sequence changes. However, out of ~30,000 substitutions and indels that reached fixation in modern humans, less than 100 directly alter amino acid sequence[1], and as of today, our ability to estimate the biological effects of the remaining ~30,000 noncoding changes is very restricted. Whereas many of them are probably nearly neutral, many others may affect gene function, especially those in regulatory regions such as promoters and enhancers. Such changes in regulatory elements may have a sizeable impact on human evolution, as alterations in gene regulation are thought to underlie much of the phenotypic variation between closely related groups[2]. Because of the limited ability to interpret noncoding variants, direct examination of regulatory layers such as DNA methylation has the potential to enhance our understanding of the evolutionary origin of human-specific traits far beyond what can be achieved using sequence comparison alone[3].

In order to gain insight into changes in regulatory elements that might underlie human evolution, we previously developed a method to reconstruct DNA methylation maps of ancient genomes based on analysis of patterns of damage to ancient DNA[4]. We used this method to reconstruct the methylomes of a Neanderthal and a Denisovan, which were then compared to a partial methylation map of a present-day osteoblast cell line. However, the ability to identify differentially methylated regions (DMRs) between the human groups was constrained by the incomplete reference map (providing methylation information for ~10% of CpG sites), differences in outputs of sequencing platforms, lack of an outgroup, and a restricted set of skeletal samples (see Supplementary Methods).

To study the evolutionary dynamics of DNA methylation along the hominin tree on a larger scale, here we use 69 skeletal DNA methylation maps from modern humans, archaic humans, and chimpanzees to identify 588 genes whose methylation state is unique to modern humans. We then analyze the function of these genes by investigating their known anatomical effects, and validate this using over 50 orthogonal tests and controls. We find that the most extensive DNA methylation changes are observed in genes that affect vocal and facial anatomy, and that this trend appears to be unique to modern humans.

## Results

### Generating DNA methylation maps.
We reconstructed ancient DNA methylation maps of eight individuals: in addition to the previously published Denisovan and Altai Neanderthal methylation maps[4], we reconstructed the methylomes of the Vindija Neanderthal (~52 thousand years ago, kya)[5], and three anatomically modern humans: the Ust'-Ishim individual (~45 kya, Western Siberia)[6], the Loschbour individual (~8 kya, Luxemburg)[7], and the Stuttgart individual (~7 kya, Germany)[7]. We also sequenced to high-coverage and reconstructed the methylomes of the La Braña 1 individual from Spain (~8 kya, 22×) (which was previously sequenced to low-coverage)[8] and an individual from Barçın Höyük, Western Anatolia, Turkey (I1583, ~8.5 kya, 24×), which was previously sequenced using a capture array[9].

To this set we added 52 publicly available partial bone methylation maps from present-day individuals, produced using 450K methylation arrays (see Supplementary Methods). To obtain full present-day bone maps, we produced whole-genome bisulfite sequencing (WGBS) methylomes from the femur bones of two individuals (Bone1 and Bone2, Supplementary Fig. 1). Hereinafter, ancient and present-day modern humans are collectively referred to as anatomically modern humans (AMHs), while the Neanderthal and Denisovan are referred to as archaic humans. As an outgroup, we produced methylomes of six chimpanzees: one WGBS, one reduced representation bisulfite sequencing (RRBS) and four 850K methylation arrays (Supplementary Fig. 1, Supplementary Tables 1, 2). Together, these data establish a unique and comprehensive platform to study DNA methylation dynamics in recent human evolution (Supplementary Data 1).

### Identification of DMRs.
We developed a DMR-detection method for ancient methylomes, which accounts for potential noise introduced during reconstruction, as well as differences in coverage and deamination rates. To minimize the number of false positives and to identify DMRs that are most likely to have a regulatory effect, we applied a strict threshold of >50% difference in methylation across a minimum of 50 CpGs. This also filters out environmentally-induced DMRs which typically show small methylation differences and limited spatial scope[10]. Using this method, we identified 9679 regions overall that showed methylation differences between any of the high-quality representative methylomes of the Denisovan, the Altai Neanderthal, and the Ust'-Ishim anatomically modern human. These regions do not necessarily represent evolutionary differences between the human groups. Rather, many of them could be attributed to variability within populations or to factors separating the three individuals (e.g., Ust'-Ishim is a male whereas the archaic humans are females). One common approach to minimize such false positives is to match samples in as many parameters as possible: age, sex, bone type, technology, and disease state. However, this is rarely fully accomplished, and samples often remain unmatched in one or more categories. A stricter approach is to leverage the variability between samples to estimate the maximum contribution of their confounders. To this end, and to minimize the contribution of within-population variability, we used the 59 additional human maps to filter out regions where variability in methylation is detected. Importantly, our samples come from both sexes, from individuals of various ages and ancestries, from patients and healthy individuals, and from a variety of skeletal parts (femur, skull, phalanx, tooth, and rib). We adopted a conservative approach, whereby we take only loci where methylation in one hominin group is found completely outside the range of methylation in the other groups (Fig. 1a). Hence, as the differences between groups is considerably larger than the contribution of these factors, this procedure is expected to account for these potentially confounding factors, and the remaining DMRs are expected to represent true evolutionary differences (Fig. 1a–c and Supplementary Figs. 2, 3, Supplementary Methods). This step resulted in a set of 7649 DMRs that discriminate between the human groups, which we ranked according to their significance level. For the top candidates (see later), we also ascertain that the differential methylation is observed when matching samples from the same bone type, age, disease state, and technology.

Next, using the chimpanzee samples, we were able to determine for 2825 of these DMRs the lineage where the methylation change occurred (Fig. 1d). Of these DMRs, 873 are AMH-derived, 939 are archaic-derived, 443 are Denisovan-derived, and 570 are Neanderthal-derived (Fig. 2a and Supplementary Fig. 3, Supplementary Data 2). To study the derived biology of AMHs, and to focus on DMRs that are based on the most extensive set of maps, we concentrated on the 873 AMH-derived DMRs. We found that these DMRs are located 58× closer to AMH-derived sequence changes than expected by chance (0.092 Mb vs. a median of 5.3 Mb, $P < 10^{-5}$, permutation test, Fig. 2b). This suggests that some of the methylation changes

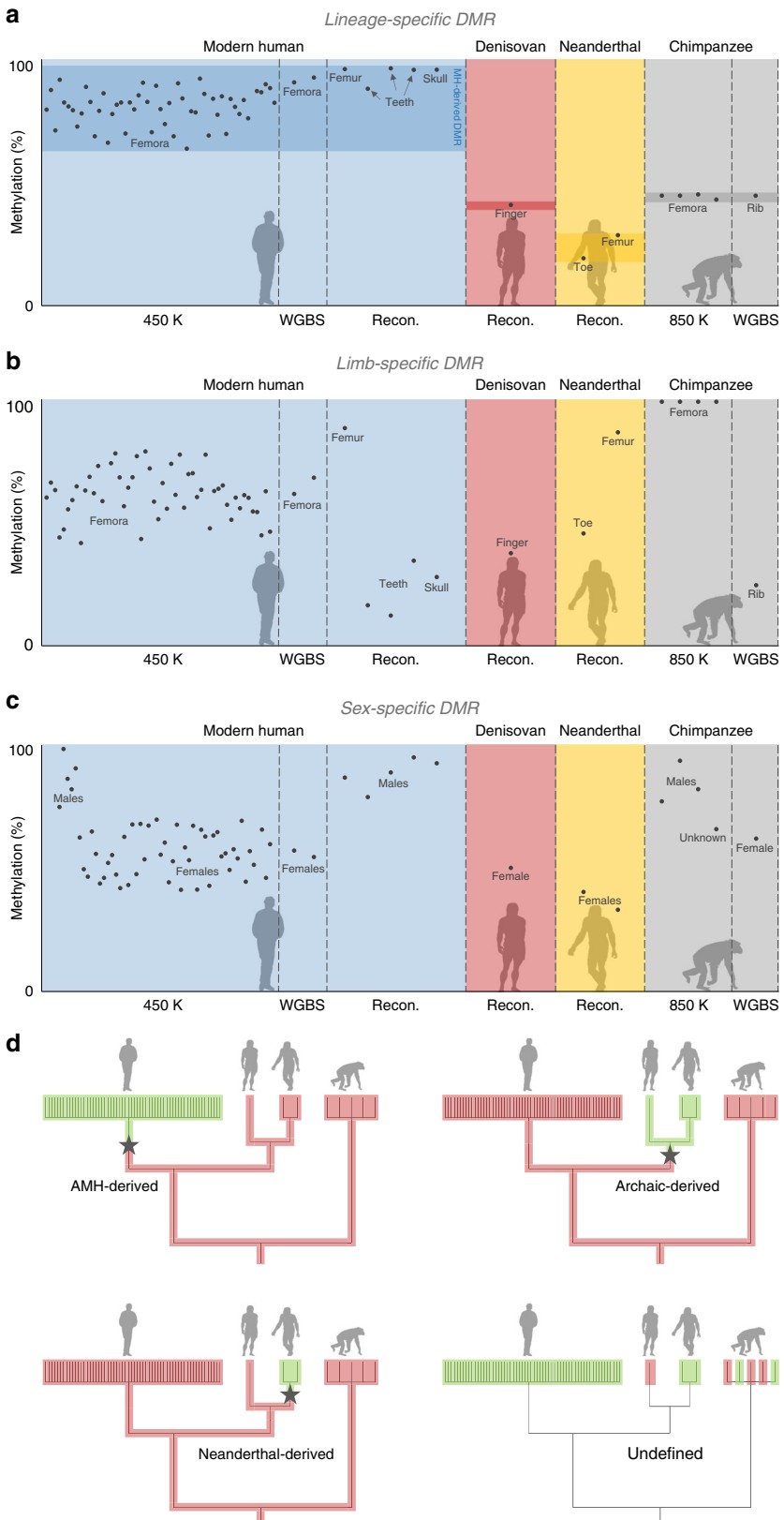

might be associated with cis-regulatory sequence variants that arose along the AMH lineage.

**Face and voice-associated genes are derived in AMHs.** We defined differentially methylated genes (DMGs) as genes that overlap at least one DMR along their body or in their promoter,

up to 5 kb upstream. The 873 AMH-derived DMRs are linked to 588 AMH-derived DMGs (Supplementary Data 2). To gain insight into the function of these DMGs, we first analyzed their gene ontology (GO). As expected from genes that show differential methylation in the skeleton between human groups, AMH-derived DMGs are enriched with terms associated with the

**Fig. 1 Variability filtering and lineage assignment. a** Methylation levels across AMH, Denisovan, Neanderthal, and chimpanzee samples in DMR#278 (chr4:38,014,896–38,016,197). This is an example of a lineage-specific DMR, defined as a locus in which all samples of a group are found outside the range of methylation in the other groups. Chimpanzee samples were used during the following step of lineage assignment. **b** A putative limb-specific DMR (chr3:14,339,371–14,339,823) which was removed from the analysis, as it does not comply with our definition of lineage-specific DMRs. Femur, toe, and finger samples are hypermethylated compared to other skeletal elements. Toe and finger are found at the bottom range of limb samples, suggesting some variation in this locus within limb samples too. **c** A putative sex-specific DMR (chr3:72,394,336–72,396,901) which was removed from the analysis. Males are hypermethylated compared to females. **d** Lineage assignment using chimpanzee samples. Only DMRs that passed the previous variability filtering steps were analyzed. Each bar at the tree leaves represents a locus in a sample. Methylation levels of the locus in each sample are marked with red (methylated) and green (unmethylated). The lineage where the methylation change has likely occurred (by parsimony) is marked by a star. Branch lengths are not scaled.

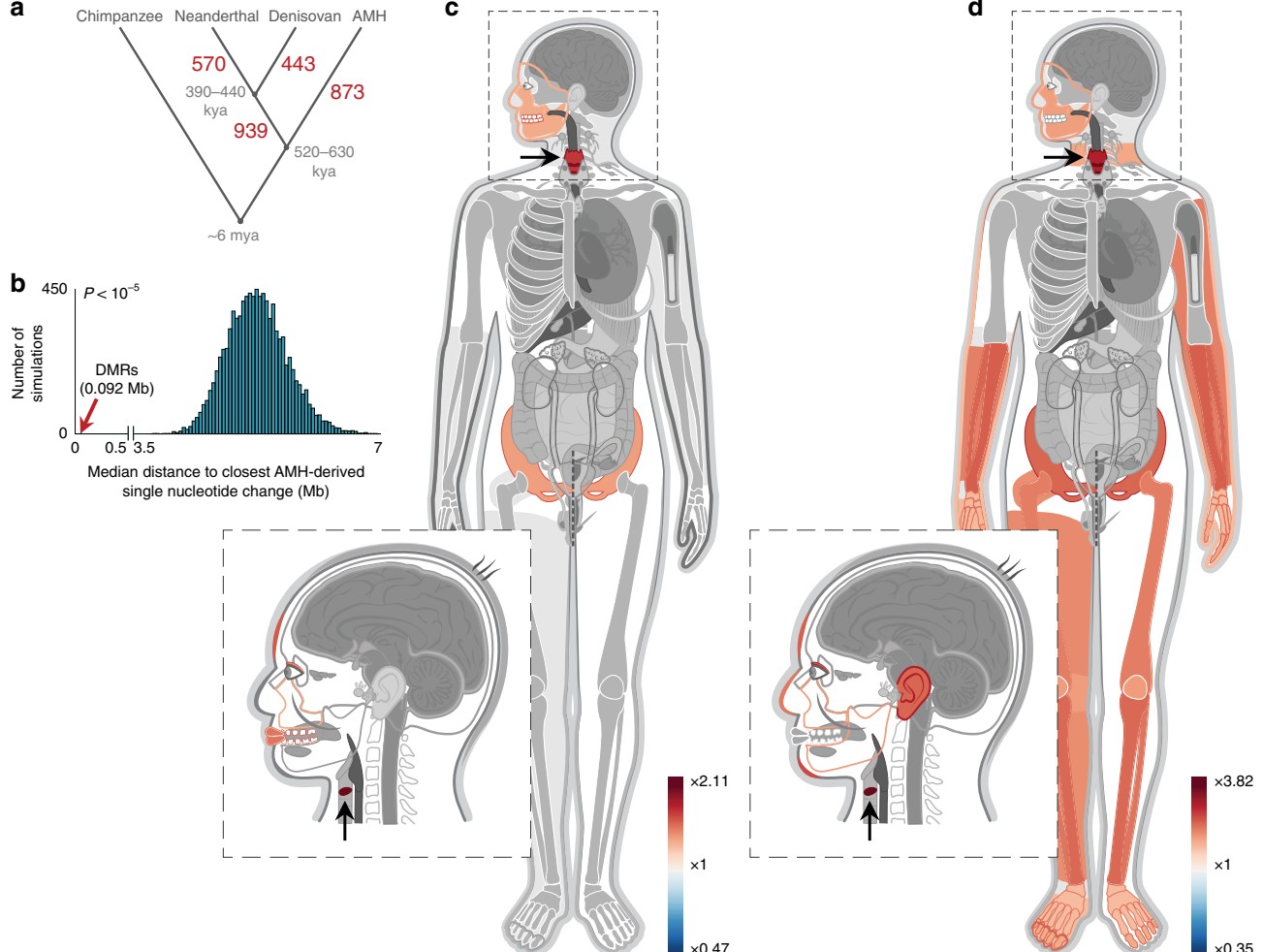

**Fig. 2 Genes affecting voice and face are the most over-represented within AMH-derived DMRs. a** The number of DMRs that emerged along each of the human branches. Divergence times are in thousands of years ago (kya). **b** Distribution of median distances (turquoise) of DMRs to randomized single nucleotide changes that separate AMHs from archaic humans and chimpanzees. Genomic positions of single nucleotide changes were allocated at random. This was repeated 10,000 times. Red arrow marks the observed distance of DMRs, showing that they tend to be significantly closer to AMH-derived single nucleotide changes than expected by chance. This suggests that some of these sequence changes might be associated with the changes in methylation. **c** A heat map representing the level of enrichment of each anatomical part within the AMH-derived DMRs. Only body parts that are significantly enriched (FDR < 0.05) are colored. Three skeletal parts are significantly over-represented: the face, pelvis, and larynx (voice box, marked with arrows). **d** Enrichment levels of anatomical parts within the most significant (top quartile, Q statistic) AMH-derived DMRs, showing a more pronounced enrichment of genes affecting vocal and facial anatomy.

skeleton (e.g., endochondral bone morphogenesis, trabecula morphogenesis, and palate development). Also notable are terms associated with the muscular, cardiovascular, and nervous systems (Supplementary Data 3).

To acquire a more precise understanding of the possible functional consequences of these DMGs, we used Gene

ORGANizer, which links human genes to the organs they phenotypically affect[11]. Unlike tools that use GO terms or RNA expression data, Gene ORGANizer is based entirely on curated gene-disease and gene-phenotype associations from monogenic diseases. It relies on direct phenotypic observations in human patients whose conditions are associated with known gene

perturbations. Hereinafter, we refer to genes as associated with an organ if they have been shown to have a phenotypic effect on that organ in patients where this gene is dysfunctional. An enrichment or depletion in Gene ORGANizer is detected if the group of genes analyzed shows a significant deviation in the organs they are known to phenotypically affect, compared to the rest of the genome. Using Gene ORGANizer, we found 11 organs that are over-represented within the 588 AMH-derived DMGs, eight of which are skeletal parts that can be divided into three regions: the face, larynx (voice box), and pelvis (Fig. 2c and Supplementary Data 4). The strongest enrichment was observed in the laryngeal region (×2.11 and ×1.68, FDR = 0.017 and 0.046, for the vocal folds (vocal cords) and larynx, respectively), followed by facial and pelvic structures, including the teeth, forehead, jaws, and pelvis. Interestingly, the face and pelvis are considered the most morphologically divergent regions between Neanderthals and AMHs[12] and our results reflect this divergence through gene regulation changes. To gain orthogonal evidence for the enrichment of the larynx and face within these AMH-derived DMGs, we carried out a number of additional analyses: First, we analyzed gene expression patterns and found that the supralaryngeal vocal tract (the pharyngeal, oral, and nasal cavities, where sound is filtered to specific frequencies) is the most enriched body part (1.7× and 1.6×, FDR = $5.6 \times 10^{-6}$ and FDR = $7.3 \times 10^{-7}$, for the pharynx and larynx, respectively, hypergeometric test, Supplementary Data 3). Second, 44 of the AMH-derived DMRs overlap previously reported putative enhancers of human craniofacial developmental genes (5.1× compared to expected, $P < 10^{-4}$, permutation test)[13,14]. Third, Palate development is the third most enriched GO term among AMH-derived DMGs (Supplementary Data 3). Fourth, DMGs significantly overlap genes associated with craniofacial features in the GWAS catalog[15] ($P = 3.4 \times 10^{-4}$, hypergeometric test).

To test whether this enrichment remains if we take only the most confident DMRs, we limited the analysis to DMGs where the most significant DMRs are found (top quartile, Q statistic). Here, the over-representation of voice-affecting genes is even more pronounced (2.82× and 2.26×, for vocal folds and larynx, respectively, FDR = 0.028 for both, Fig. 2d and Supplementary Data 4).

Next, we reasoned that skeleton-associated genes might be over-represented in analyses that compare bone DNA methylation maps, hence introducing potential biases. To test whether this enrichment might explain the over-representation of the larynx, face, and pelvis, we compared the fraction of genes associated with these organs within all skeletal genes to their fraction within the skeletal genes in the AMH-derived DMGs. We found that genes associated with the face, larynx, and pelvis are significantly over-represented even within skeletal AMH-derived DMGs ($P = 1.0 \times 10^{-5}$, $P = 1.3 \times 10^{-3}$, $P = 2.1 \times 10^{-3}$, $P = 0.03$, for vocal folds, larynx, face, and pelvis, respectively, hypergeometric test). Additionally, using a permutation test, we found that the enrichment levels within AMH-derived DMGs are significantly higher than expected by chance for the laryngeal and facial regions, but not for the pelvis ($P = 8.0 \times 10^{-5}$, $P = 3.6 \times 10^{-3}$, $P = 8.2 \times 10^{-4}$, and $P = 0.115$, for vocal folds, larynx, face and pelvis, respectively, permutation test, Supplementary Fig. 4, Supplementary Methods). Thus, we found that the enrichment in the facial and laryngeal regions is not a by-product of a general enrichment in skeletal parts, and we hereinafter focus on genes associated with these two regions.

Finally, we ruled out the options that our DMR-detection algorithm, number of samples, filtering process or biological factors such as gene length, cellular composition, pleiotropy or developmental stage might underlie the enrichment of these organs (see Supplementary Methods). Perhaps most importantly,

none of the other branches shows enrichment of the larynx or the vocal folds; Neanderthal-derived and Denisovan-derived DMGs show no significant enrichment in any organ, and archaic-derived DMGs are over-represented in the jaws, lips, limbs, scapulae, and spinal column, but not in the larynx or vocal folds (Supplementary Fig. 4e, Supplementary Data 4). In addition, DMRs that separate chimpanzees from all humans (archaic and modern, Supplementary Data 2) do not show enrichment of genes associated with the larynx or face, compatible with the notion that this trend emerged along the AMH lineage.

Taken together, we conclude that DMGs that emerged along the AMH lineage are uniquely enriched in genes associated with the voice and face, and that this is unlikely to be an artifact of (a) inter-individual variability resulting from age, sex, disease, or bone type; (b) significance level of DMRs; (c) the reconstruction or DMR-detection processes; (d) number of samples used; (e) pleiotropic effects; (f) the types of methylation maps used; (g) the comparison of skeletal methylomes; (h) gene length distribution; or (i) biological factors such as cellular composition and developmental state.

Our analyses identified 56 DMRs in genes associated with the facial skeleton, and 32 in genes associated with the laryngeal skeleton. The face-associated genes are known to shape mainly the protrusion of the lower and midface, the size of the nose, and the slope of the forehead. Interestingly, these traits are considered some of the most derived between Neanderthals and AMHs[12]. The larynx-associated genes have been shown to underlie various phenotypes in patients, ranging from slight changes to the pitch and hoarseness of the voice, to a complete loss of speech ability[11] (Supplementary Data 5). These phenotypes were shown to be driven primarily by alterations to the laryngeal and vocal tract skeleton. Methylation patterns in differentiated cells are often established during earlier stages of development, and the closer two tissues are developmentally, the higher the similarity between their methylation maps[3,16,17]. This is also evident in the fact that DMRs identified between species in one tissue often exist in other tissues as well[16]. Importantly, the laryngeal skeleton, and particularly the arytenoid cartilage to which the vocal folds are anchored, share an origin from the somatic layer of the lateral plate mesoderm with the cartilaginous tissue of the limb bones prior to their ossification. Thus, it is likely that many of the DMRs identified here between limb samples also exist in their closest tissue—the laryngeal skeleton. This is further supported by the observation that these DMGs are consistent across all examined skeletal samples, including skull, femur, rib, tibia, and tooth. Furthermore, we directly measured methylation levels in a subset of the DMRs in primary chondrocytes and show that their patterns extend to these cells as well (see below).

**Extensive changes within face and voice-associated genes.** The results above suggest that methylation levels in many face-associated and voice-associated genes have changed in AMHs since the split from archaic humans, but they do not provide information on the extent of changes within each gene. To do so, we scanned the genome in windows of 100 kb and computed the fraction of CpGs which are differentially methylated in AMHs (hereinafter, AMH-derived CpGs). We found that the extent of changes within voice-associated DMGs is most profound, more than 2× compared to other DMGs (0.132 vs. 0.055, FDR = $2.3 \times 10^{-3}$, t-test, Supplementary Data 6). Face-associated DMGs also present high density of AMH-derived CpGs (0.079 vs. 0.055, FDR = $2.8 \times 10^{-3}$, t-test). In archaic-derived DMGs, on the other hand, the extent of changes within voice-associated and face-associated genes is not different than expected (FDR = 0.99, t-test, and Supplementary Data 6). To control for possible biases,

we repeated the analysis using only the subset of DMRs in genes associated with the skeleton. Here too, we found that voice-associated AMH-derived DMGs present the highest density of changes (2.5× for vocal folds, 2.4× for larynx, $FDR = 1.4 \times 10^{-3}$ for both, t-test, Supplementary Data 6), and face-associated DMGs also exhibit a significantly elevated density of changes (1.4×, $FDR = 0.04$, t-test).

We also found that compared to other AMH-derived DMRs, DMRs in voice-associated and face-associated genes tend to be 40% closer to candidate positively selected loci in AMHs[18] ($P < 10^{-4}$, permutation test).

Strikingly, when scanning the genome for hotspots of methylation changes, all top five skeleton-related loci are found within genes known to associate with lower and midfacial protrusion, as well as the voice (ACAN, SOX9, COL2A1, XYLT1, and NFIX)[11,19] (Fig. 3 and Supplementary Fig. 4f). This is particularly surprising considering that genome-wide, less than 2% of genes (345) are known to affect the voice, ~3% of genes (726) are known to associate with lower and midfacial protrusion, and less than 1% (182) are known to associate with both[11,19].

The three skeletal DMGs with the highest density of AMH-derived CpGs are the extra-cellular matrix genes ACAN and COL2A1, and their key regulator SOX9, which together form a network that regulates skeletal growth, the transition from cartilage to bone, and spatio-temporal patterning of skeletal development, including the facial and laryngeal skeleton in humans[19,20] and mouse[21]. SOX9 was also shown to be one of the top genes associated with variation in craniofacial morphology within-AMHs[22]. SOX9 is regulated by a series of upstream enhancers identified in mouse and human[23]. In human skeletal samples, hypermethylation of the SOX9 promoter was shown to downregulate its activity, and consequently its targets[24]. This was also demonstrated repeatedly in non-skeletal tissues of human[25,26] and mouse[27,28]. We found substantial hypermethylation in AMHs in the following regions: (a) the SOX9 promoter; (b) seven of its proximal and distal skeletal and skeletal progenitor enhancers[23]; (c) the targets of SOX9: ACAN (DMR #80) and COL2A1 (DMR #1, the most significant AMH-derived DMR, which spans 32 kb and covers almost the entire COL2A1 gene, from its 1st intron to its 54th exon and 3′UTR region); and (d) an upstream lincRNA (LINC02097). Notably, regions (a), (b), and (d) overlap the longest DMR on the AMH-derived DMR list, spanning 35,910 bp (DMR #11, Fig. 4). Additionally, a more distant putative enhancer, located 345 kb upstream of SOX9, was shown to bear strong active histone modification marks in chimpanzee craniofacial progenitor cells; whereas, in humans these marks are almost absent (~10× lower than chimpanzee, suggesting downregulation, Fig. 4b)[13]. Importantly, human and chimpanzee non-skeletal tissues (i.e., brain and blood) exhibit very similar methylation patterns in these genes, suggesting the DMRs are skeleton-specific. Finally, the amino acid sequence coded by each of these genes is identical across the hominin groups[1], suggesting that the observed changes are purely regulatory. Together, these observations support the idea that SOX9 became downregulated in AMH skeletal tissues, likely followed by downregulation of its targets: ACAN and COL2A1.

XYLT1, the 4th highest skeleton-related DMG, is an enzyme involved in the synthesis of glycosaminoglycan. Loss-of-function mutations, hypermethylation of the gene and its consequent reduced expression underlie the Desbuquois dysplasia skeletal syndrome, which was shown to affect the cartilaginous structure of the larynx, and drive a retraction of the face[29,30]. Very little is known about XYLT1 regulation, but interestingly, in zebrafish it was shown to be bound by SOX9 (ref. [31]).

To quantitatively investigate the potential phenotypic consequences of these DMGs, we tested what fraction of their known phenotypes are also known as traits that differ between modern and archaic humans. We found that four of the top five most differentially methylated genes (XYLT1, NFIX, ACAN, and COL2A1) are in the top 100 genes with the highest fraction of divergent traits between Neanderthals and AMHs. Remarkably, COL2A1, the most divergent gene in its methylation patterns, is also the most divergent in its phenotypes: no other gene in the genome is associated with as many divergent traits between modern humans and Neanderthals[32] (63 traits, Supplementary Data 7, see Supplementary Methods). This suggests that these extensive methylation changes are possibly linked to phenotypic divergence between archaic and AMHs.

**NFIX methylation patterns suggest downregulation in AMHs.** In order to investigate how methylation changes affect expression levels, we scanned the DMRs to identify those whose methylation levels are strongly correlated with expression across 22 human tissues[33]. We found 90 such AMH-derived DMRs ($FDR < 0.05$, Supplementary Data 2). DMRs in voice-associated genes are significantly more likely to be correlated with expression compared to other DMRs (2.05×, $P = 6.65 \times 10^{-4}$, hypergeometric test). Particularly noteworthy is NFIX, one of the most derived genes in AMHs (ranked 5th among DMGs affecting the skeleton, Fig. 3a, b). NFIX contains two DMRs (#24 and #167, Fig. 5a), and in both, methylation levels are tightly linked with expression (correlation of 81.7 and 73.8%, $FDR = 3.5 \times 10^{-6}$ and $8.6 \times 10^{-5}$, respectively, Pearson's r, Fig. 5b). In fact, NFIX is one of the top ten DMGs with the most significant correlation between methylation and expression in human. The association between NFIX methylation and expression was also shown previously across several mouse tissues[34,35]. To further examine this, we investigated a dataset of DNMT3A-induced methylation of human MCF-7 cells. Forced induction of methylation in this study was sufficient to repress NFIX expression by over 50%, placing NFIX as one of the genes whose expression is most affected by hypermethylation[36] (ranked in the 98th percentile, $FDR = 1.28 \times 10^{-6}$). We further validated the hypermethylation of NFIX across the skeleton by comparing four human cranial samples to four chimpanzee cranial samples through bisulfite-PCR ($P = 0.01$, t-test, Supplementary Fig. 5, Supplementary Data 1, Supplementary Table 3, Supplementary Methods). Together, these findings suggest that the observed hypermethylation of NFIX in AMHs reflects downregulation that emerged along the AMH lineage. Indeed, we found that NFIX, as well as SOX9, ACAN, COL2A1, and XYLT1 are hypermethylated in human femora compared to baboon[37] ($P = 1.4 \times 10^{-5}$ and $P = 8.1 \times 10^{-9}$, compared to baboon femora bone and cartilage, respectively, t-test). Also, all five genes show significantly reduced expression in humans compared to mice (Fig. 5c). Taken together, these observations suggest that DNA methylation is a primary mechanism in the regulation of NFIX, and serves as a good proxy for its expression. Interestingly, NFI proteins were shown to bind the upstream enhancers of SOX9 (ref. [38]), hence suggesting a possible mechanism to the simultaneous changes in the five top genes we report.

## Discussion

We have shown here that genes associated with vocal and facial anatomy have different DNA methylation patterns in recent AMH, compared to Neanderthals and Denisovans. The differences in DNA methylation are manifested both in the number of divergent genes and in the extent of changes within each gene. Notably, the DMRs we report capture substantial methylation changes (over 50% between at least one pair of human groups), span thousands or tens of thousands of bases, including in

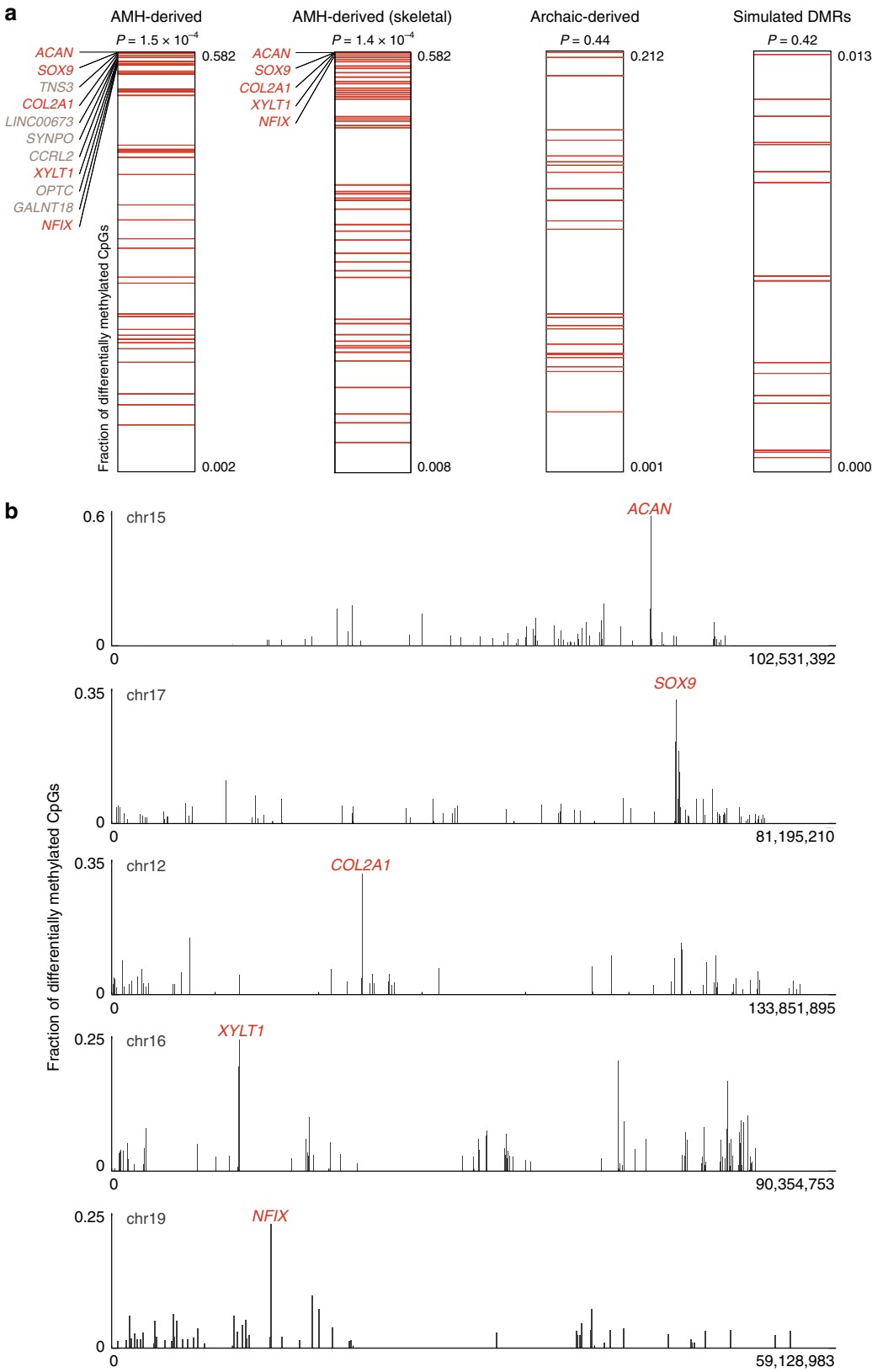

promoters and enhancers. Many of these methylation changes are associated with changes in expression. We particularly focused on changes that might affect regulation of the five most derived skeletal genes on the AMH lineage: *SOX9, ACAN, COL2A1, XYLT1*, and *NFIX*, whose downregulation was shown to underlie a retracted face, as well as changes to the structure of the

larynx[20,29,39–42]. The results we report, which are based on ancient DNA methylation patterns, provide means to analyze the genetic mechanisms that underlie the evolution of the human face and vocal tract.

One of the limitations of analyzing regulatory maps is their tissue-specific, sex-specific, age-specific, disease-specific, and

**Fig. 3 The extent of differential methylation is highest among genes associated with the voice. a** Within each lineage, the fraction of differentially methylated CpGs was computed as the number of derived CpGs per 100 kb centered around the middle of each DMR. DMRs were ranked according to the fraction of derived CpG positions in their vicinity. DMRs in genes associated with the voice are marked with red lines. In AMHs, DMRs in voice-affecting genes tend to be ranked significantly higher. Although known voice-associated genes comprise less than 2% of the genome, three of the top five AMH-derived DMRs, and all top five skeleton-related AMH-derived DMRs are in genes known to associate with the voice. In archaic-derived DMRs and in simulated DMRs, voice-associated genes do not show higher ranking compared to the rest of the DMGs. *t*-test *P*-values are shown for each group. **b** The fraction of differentially methylated CpGs along the five chromosomes containing *ACAN*, *SOX9*, *COL2A1*, *XYLT1*, and *NFIX*. In each of these chromosomes, the most extensive changes are found within these genes. All five genes control facial projection and the development of the larynx.

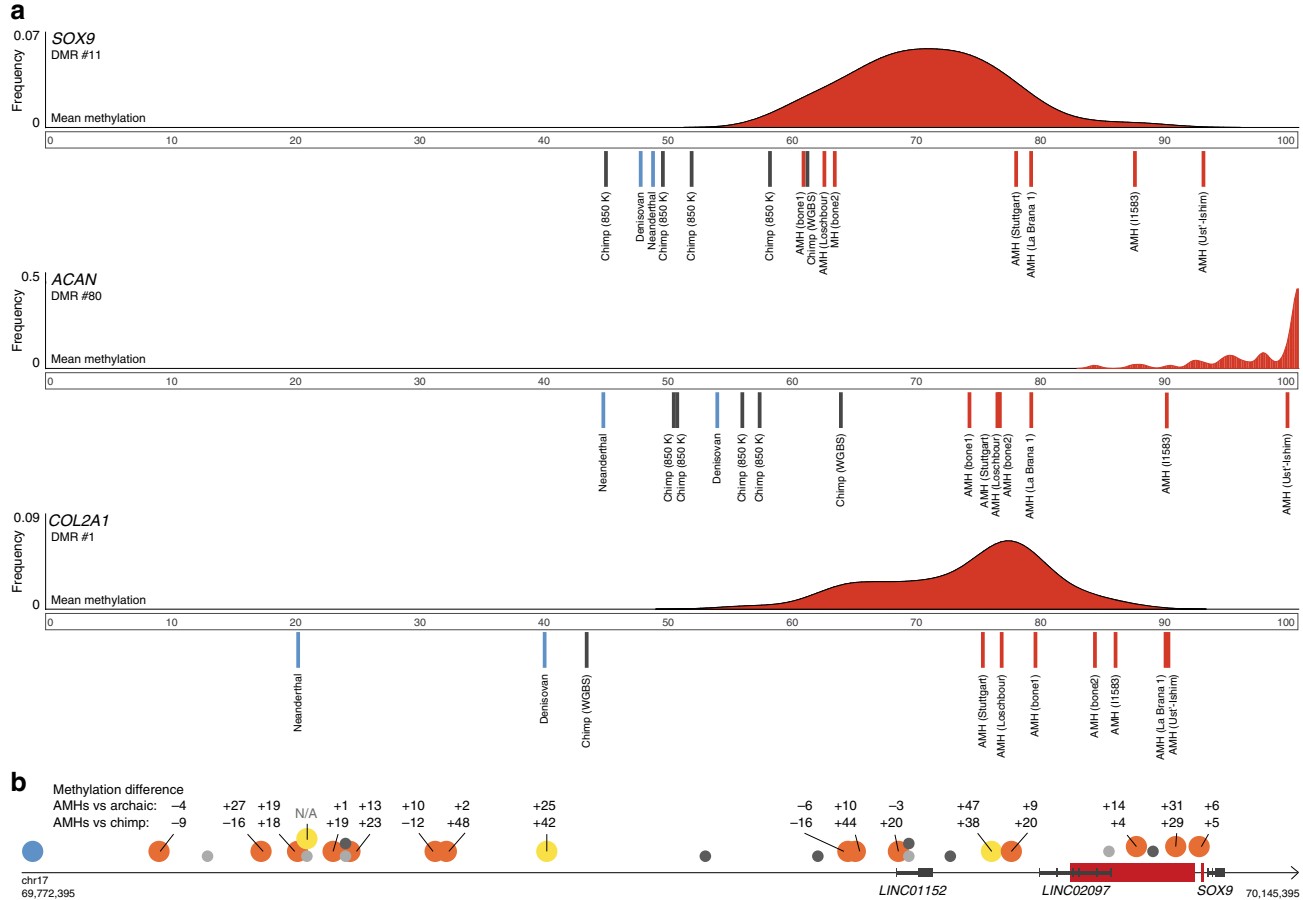

**Fig. 4 Hypermethylation of *SOX9*, *ACAN*, and *COL2A1* in AMHs. a** Methylation levels in the AMH-derived DMRs in *SOX9*, *ACAN*, and *COL2A1*. AMH samples are marked with red lines, archaic human samples are marked with blue lines and chimpanzee samples are marked with gray lines. The distribution of methylation across 52 AMH samples (450K methylation arrays) is presented in red. **b** *SOX9* and its upstream regulatory elements. AMH-derived DMRs are marked with red rectangles. Previously identified putative enhancers are marked with circles: human craniofacial (orange), human craniofacial, chimpanzee-biased (blue), human skeletal (yellow), human non-skeletal (dark gray), and mouse (light gray). Numbers above skeletal enhancers show the difference in mean bone methylation between AMHs and archaic humans (top) and between AMHs and chimpanzee (bottom). Across almost all *SOX9* enhancers, AMHs are hypermethylated compared to archaic humans and the chimpanzee.

technology-specific patterns. Although the vast majority of loci show stable patterns regardless of these factors, some loci are nevertheless affected by them[43]. Perfectly matching two human samples across all of these factors is rarely achievable. Therefore, we used a stricter approach, which results in fewer discoveries, but is able to leverage the variability introduced by tissue, sex, age, disease, and technology to identify loci where these factors are unlikely to underlie the observed methylation differences. For the top candidates, we also matched samples for these factors and showed that the differential methylation is still observed.

A central confounder of reconstructed methylation maps is their resolution. Although their resolution is substantially better than that of methylation arrays, it is still inferior to that of WGBS. Both WGBS and ancient reconstruction rely on the same process

to measure methylation: deamination of cytosines. However, in WGBS, the use of bisulfite results in ~99% deamination rate, whereas in ancient reconstruction the average rate is usually lower than 5%[43]. To account for this, we used a window of 25 CpGs, which increases power at the cost of resolution. Therefore, in this study, we are only able to detect DMRs that are larger than the window used. To detect more subtle changes, genomes with higher deamination rates, or merging several genomes together are required.

Humans are distinguished from other apes in their unique capability to communicate through speech. This capacity has been attributed not only to neural changes, but also to configurational alterations to the vocal tract[44], as humans have a larynx that is located particularly low among primates[44] (Fig. 6a), and

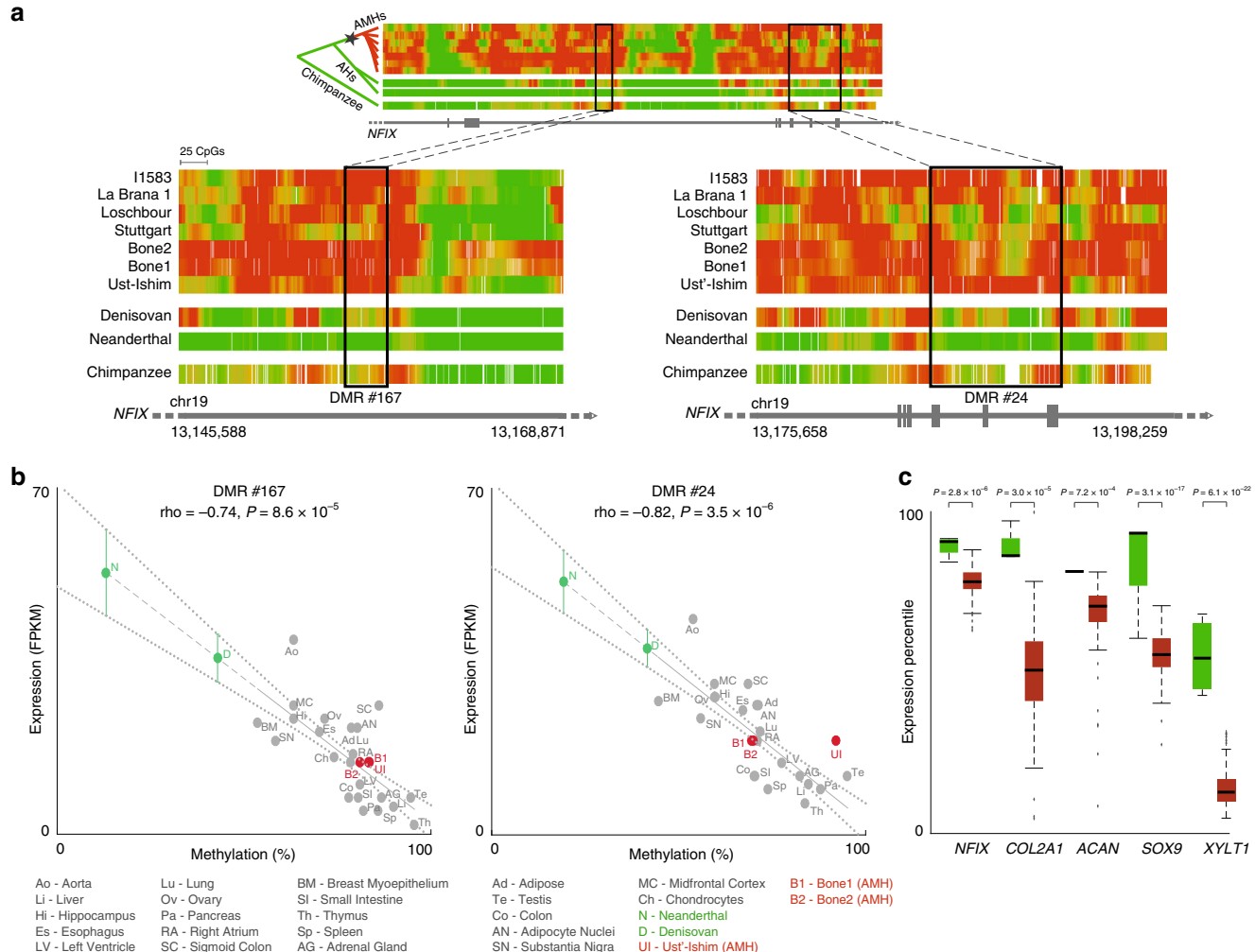

**Fig. 5 DNA methylation and expression pattern of *NFIX* in modern and archaic humans. a** Methylation levels along *NFIX*, color-coded from green (unmethylated) to red (methylated). Methylation levels around the two AMH-derived DMRs (#24 and #167) are shown in the zoomed-in panels. These two DMRs represent the regions where the most significant methylation changes are observed, but hypermethylation of *NFIX* in AMHs can be seen throughout the entire gene body. Chimpanzee and present-day samples were smoothed using the same sliding window as in ancient samples to allow easier comparison. The inferred schematic regulatory evolution of *NFIX* is shown using a phylogenetic tree to the left of the top panel. Star marks the shift in methylation from unmethylated (green) to methylated (red). **b** Methylation levels in DMRs #167 and #24 vs. expression levels of *NFIX* across 22 AMH tissues (gray). In both DMRs, higher methylation is significantly associated with lower expression of *NFIX*. Ust'-Ishim, Bone1 and Bone2 methylation levels (red) are plotted against mean NFIX expression across 13 osteoblast lines. Neanderthal and Denisovan methylation levels (green) are plotted against their predicted expression levels, based on the extrapolated regression line (dashed). Standard errors are marked with dotted lines. The expression levels of *NFIX* in Neanderthal and Denisovan shown in these graphs are extrapolated (green dots). **c** Box plots of expression levels of *NFIX, COL2A1, ACAN, SOX9,* and *XYLT1* in 89 AMH samples (red) and four mouse samples (green) from appendicular bones (limbs and pelvis). Central line shows mean, box borders show 25th and 75th percentiles, whiskers extend to the most extreme data points not considered outliers, crosses show outliers. Expression levels were converted to percentiles based on the level of gene expression compared to the rest of the genome in each sample. *t*-test *P*-values are shown for each gene.

because phonetic range is determined by the different configurations that the vocal tract can produce. The roles of divergent anatomy vs. cognition in our speech skills are still debated[45,46], and some propose that even with a human brain, other apes could not reach the human level of articulation and phonetic range[44,47]. Regardless of its potential role in speech, the process of laryngeal descent is developmentally and evolutionarily associated with facial retraction[48,49]. In this regard, the observation that the top five skeletal DMRs are found in genes that are associated with both facial protrusion and the anatomy of the larynx suggests that these two processes might have been genetically linked, though the interaction between the two is still to be determined, as their exact developmental pathways are beyond the scope of the current study. Thus, it is still to be determined how the positional and structural changes to the larynx emerged and what was the role of the reported methylation changes in this process. For an in-depth review of the anatomy of vocalization and speech, see refs. [44,46].

A longstanding question is whether Neanderthals and AMHs share similar vocal tract anatomy[50,51]. Attempts to answer this question based on morphological differences have proven hard, as the larynx is mostly composed of soft tissues (e.g., cartilage), which do not survive long after death. The only remnant from the Neanderthal laryngeal region is the hyoid bone, which is detached from the rest of the skull[51]. Based on this single bone, or on computer simulations and tentative vocal tract reconstructions, it

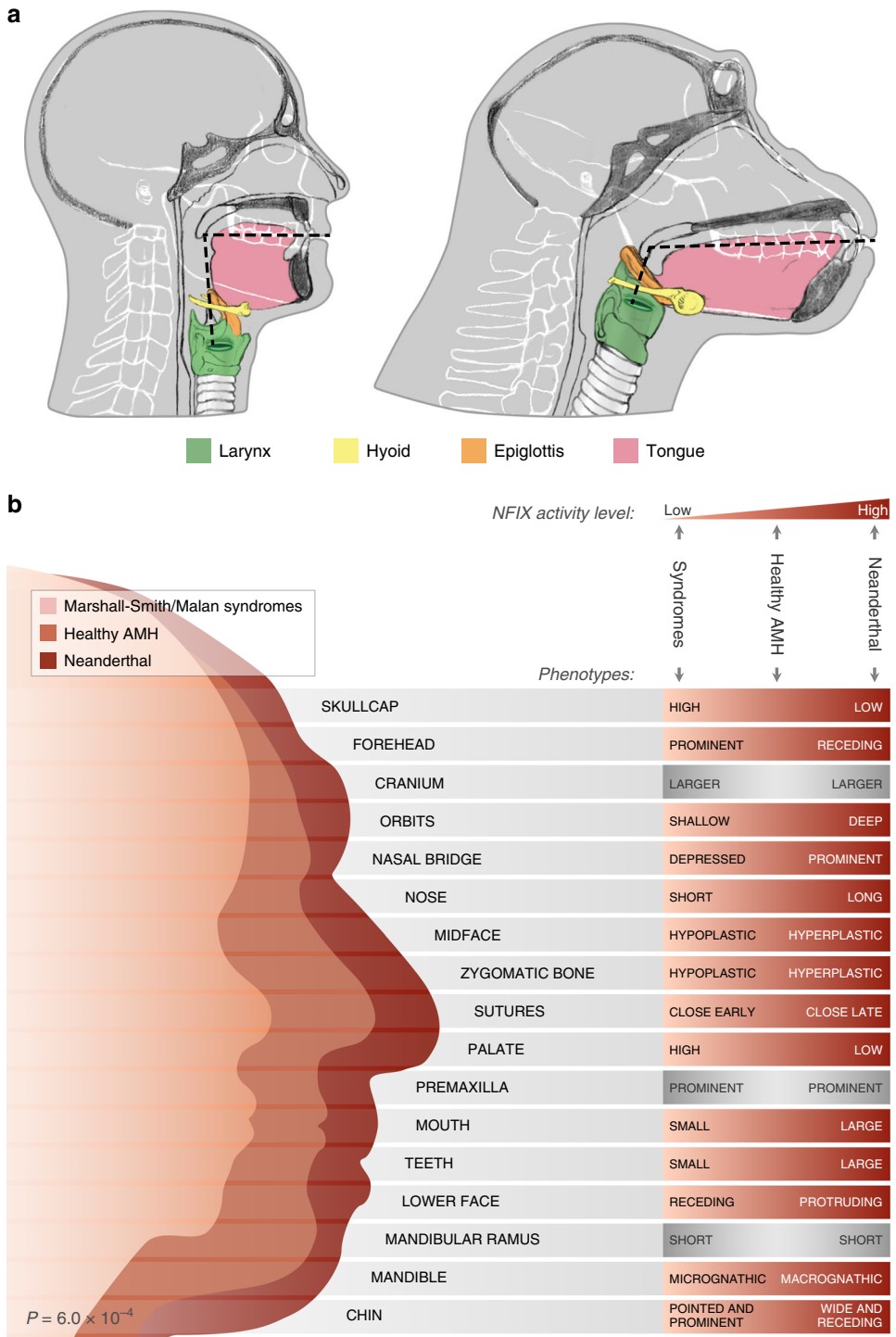

**Fig. 6 *NFIX* down-regulation is associated with modern human-derived traits. a** Vocal anatomy of chimpanzee and AMH. The vocal tract is the cavity from the lips to the larynx (marked by dashed lines). In AMHs, the flattening of the face together with the descent of the larynx led to approximately 1:1 proportions of the horizontal and vertical portions of the vocal tract. **b** Craniofacial features of the Neanderthal, healthy AMH, and AMH with Marshall-Smith or Malan syndromes. Each box shows a phenotype that occurs in the Marshall-Smith/Malan syndromes (i.e., when *NFIX* is partially or completely inactive). The righthand side of each box shows the observed phenotypes of individuals with the syndromes (left), healthy AMHs (middle) and Neanderthals (right). In most phenotypes, the observed phenotypes match the expected phenotypes based on *NFIX* expression. Binomial test *P*-value is shown.

is difficult to characterize the full anatomy of the Neanderthal vocal apparatus, and opinions remain split as to whether it was similar to that of AMHs[50,51].

Most skeletal disease phenotypes that result from *NFIX* dysfunction are craniofacial, as *NFIX* influences the balance between lower and upper projection of the face[52]. In addition, mutations in *NFIX* were shown to impair speech capabilities[42,53]. The exact mechanism is still unknown, but is thought to occur partly through skeletal alterations to the larynx[42]. To investigate if changes in *NFIX* expression could explain morphological changes

in the AMH face and larynx, we examined its clinical skeletal phenotypes. Mutations in *NFIX* were shown to cause the Marshall–Smith and Malan syndromes, whose phenotypes include various skeletal alterations such as hypoplasia of the midface, retracted lower jaw, and depressed nasal bridge[52]. In many patients, the phenotypic alterations are driven by heterozygous loss-of-function mutations that cause haploinsufficiency. This shows that reduced activity of *NFIX*, even if partial, results in skeletal alterations[52]. Because *NFIX* is inferred to have been downregulated in AMHs compared to archaic humans, we hypothesized that similar phenotypes to the ones that are driven by *NFIX* loss-of-function may also exist between modern and archaic humans. For example, because reduced activity of *NFIX* results in a more retracted face, we hypothesized that AMHs would present a more retracted face compared to archaic humans. A similar relationship between facial features and gene dosage has been recently shown in the 16p11.2 locus in humans[54]. We therefore examined the phenotypes of the Marshall–Smith and Malan syndromes and found that not only do most of these phenotypes exist between Neanderthals and modern humans, but their direction matches the direction expected from NFIX downregulation along the AMH lineage (18 out of the 22 Marshall–Smith phenotypes, and 8 out of the 9 Malan phenotypes, $P = 6.0 \times 10^{-4}$, binomial test). In other words, from the Neanderthal, where *NFIX* activity is expected to be highest, through healthy AMHs, to individuals with *NFIX* haploinsufficiency, phenotypic manifestation matches the level of *NFIX* activity (Fig. 6b and Supplementary Data 8).

Notably, many cases of laryngeal malformations in the Marshall–Smith syndrome have been reported. Some of the patients exhibit positional changes to the larynx, changes in its width, and, more rarely, structural alterations to the arytenoid cartilage—the anchor point of the vocal folds, which controls their movement[55]. In fact, these laryngeal and facial changes are thought to underlie some of the limited speech capabilities observed in various patients[42]. This raises the possibility that *NFIX* downregulation in AMHs might be associated with changes in the larynx too.

*SOX9*, *ACAN*, *COL2A1*, *XYLT1*, and *NFIX* are active in early stages of osteochondrogenesis, making the observation of differential methylation in mature bones puzzling at first glance. This could potentially be explained by two factors: (i) The methylome stabilizes as development progresses and remains largely unchanged from late development through adulthood. Thus, adult methylation states often reflect earlier development, and DMRs in adult stages often reflect DMRs in earlier activity levels[3,17,56]. Therefore, these DMRs might reflect early methylation changes in mesenchymal progenitors that are carried over to later stages of osteogenesis. Indeed, the methylation patterns of *NFIX*, *SOX9*, *ACAN*, and *COL2A1* were shown to be established in early stages of human development and remain stable throughout differentiation from mesenchymal stem cells to mature osteocytes[57]. It is further supported by the observation that osteoblasts and chondrocytes show almost identical methylation levels in these DMRs, and are all as hypermethylated as the adult bone methylation levels we report[58]. We have reconfirmed this result by measuring methylation in these DMGs in primary human chondrocytes. Finally, we show that the upstream mesenchymal enhancer of *SOX9* (ref. [23]) is differentially methylated in AMHs (Fig. 4b). (ii) Although expression levels of *SOX9*, *ACAN*, and *COL2A1* gradually decrease with skeletal maturation, these genes were shown to remain active in later developmental stages in the larynx, vertebrae, limbs, and jaws, including in their osteoblasts[21,59]. Interestingly, these are also the organs that are most affected by mutations in these genes, implying that their late stages of activity might still play important roles in morphological

patterning[20,39–41]. It was also shown that facial growth patterns, which shape facial prognathism, differ between archaic and modern humans not only during early development, but also as late as adolescence[60]. Moreover, the main differences between human and chimpanzee vocal tracts are established during post-infant years[61].

Although the DMRs we report most likely exist throughout the skeleton, including the larynx, the evidence we present for the cranium is more direct, as the patterns are observed in modern human and chimpanzee crania. Importantly, it has been suggested that the 1:1 vocal conformation could have been entirely driven by cranial, rather than laryngeal, alterations[49]. Once archaic human cranial samples are sequenced, these observations could be more directly tested.

The results we presented open a window to study the evolution of the human vocal tract and face from genetic and epigenetic perspectives. Our data suggest shared genetic mechanisms that shaped these anatomical regions and point to evolutionary events that separate AMHs from the Neanderthal and Denisovan. The mechanisms leading to such extensive regulatory shifts, as well as if and to what extent these evolutionary changes affected vocalization and speech capabilities are still to be determined.

## Methods

**Skeletal methylation maps.** Previously, our ability to identify differentially methylated regions (DMRs) that discriminate between human groups was confined by three main factors: (i) We had a single DNA methylation map from a present-day human bone, which was produced using a reduced representation bisulfite sequencing (RRBS) protocol, which provides information for only ~10% of CpG positions in the genome. Moreover, the fact that the archaic and present-day methylomes were produced using different technologies—computational reconstruction versus RRBS—potentially introduces a bias. (ii) The analyses included only one bone methylation map from each of the human groups, which limited our ability to identify fixed differences between the groups. Although dozens of maps from additional tissues in present-day humans were included in the analyses, this narrowed the DMRs to represent only human-specific changes that are invariable between tissues. (iii) The work did not include a great ape outgroup. Thus, when a AMH-specific change was identified, it was impossible to determine whether it happened on the AMH lineage, or in the ancestor of Neanderthals and Denisovans[4].

To overcome these obstacles, a major goal of the current study was to significantly extend the span of our skeletal methylome collection, covering as many individuals, sexes, and bone types as we could. This included the generation of many samples, including the high-coverage sequencing of additional ancient genomes, as listed below.

**Present-day human bone DNA methylation maps.** We generated full DNA methylation maps from two femur head bones from present-day humans using WGBS. Femora were chosen because of their abundance in present-day human samples, as well as in ancient DNA samples[5,6,62]. In addition, we collected 53 publicly available partial skeletal methylation maps.

Trabecular bone tissue from femur heads were taken from two patients with osteoarthritis during a total hip replacement surgery, and after filling in a consent form as per Helsinki approval #0178-13-HMO. Importantly, the effects of osteoarthritis processes on trabecular bone are much less substantial than those on the synovium, cartilage, and subchondral bone. Bone1 was a left head of femur taken on August 11, 2014 from a 66 years old female and Bone 2 was a right head of femur taken on September 2, 2014 from a 63-years-old female.

DNA was extracted from bones using QIAamp® DNA Investigator kit (56504, Qiagen). Bones were cut to thin slices (0.2–0.5 mm) and then thoroughly washed (X5) with PBS, to clean samples from blood. Bones were crushed with mortar and pestle in liquid nitrogen, and 100 mg bone powder was taken to extract DNA according to the protocol Isolation of Total DNA from Bones and Teeth of the DNA Investigator kit.

We followed the protocol described in ref. [63] to carry out whole-genome bisulfite sequencing at the Center Nacional d'Anàlisi Genòmica (CNAG). In short, DNA libraries were built with the Illumina TruSeq Sample Preparation kit. Bisulfite treatment was applied in two rounds using the EpiTect Bisulfite kit (QIAGEN) and paired-end sequencing was performed on an Illumina Hi-Seq 2000 instrument. Reads were aligned using the GEM mapper[64], with the reads fully converted in silico. To this end, we generated two versions of each of the reference genomes—human (GRCh37) and viral: in the first version, C's were replaced with T's, and in the second version, G's were replaced with A's. Although methylation state should not depend on read position, positional biases have been previously reported[65]. We observed that the first few bases from each read showed a slightly higher

probability of being called as methylated, and we thus trimmed the first ten bases from each read (M-bias filtering).. Heterozygous positions, positions with a genotype error probability greater than 0.01, and positions with a read depth greater than 250 were filtered out. Only cytosines with six or more reads informative for methylation status were considered. On average, half of the reads from either strand will be informative for methylation status at a given position, so minimum coverage is typically greater than 12. Methylated and unmethylated cytosine conversion rates were determined from spiked-in bacteriophage DNA (fully methylated phage T7 and unmethylated phage lambda). Five samples were excluded based on conversion rates <0.997, supported by visual inspection of CG and non-CG methylation plots. The over-conversion rates for all samples based on methylated phage T7 DNA were ~5%.

Sequence quality was evaluated using FastQC software v0.11.2. TRIMMOMATIC v.0–32 was used to filter low quality bases with the following parameters: -phred33 LEADING:30 TRAILING:30 MAXINFO:70:0.9 MINLEN:70. Paired-end sequencing reads were mapped to bisulfite converted human (hg19) reference genome using Bismark v0.14.3 and bowtie2 v2.2.4 not allowing multiple alignments and using the following parameters: –bowtie2 –non_bs_mm –old_flag -p 4. Potential PCR duplicates were removed using Bismark's deduplicate_bismark_alignment_output.pl Perl program. Bismark's bismark_methylation_extractor script was used to produce methylation calls with the following parameters: -p –no_overlap –comprehensive –merge_non_CpG – no_header –bedGraph –multicore 2 –cytosine_report. Examination of the M-bias plots led us to ignore the first 5 bp of both reads in human samples (Supplementary Fig. 1). Custom scripts were used to summarize methylation levels at CpG sites based on the frequencies of methylated and unmethylated mapped reads on both strands. Methylation data were deposited in NCBI's Gene Expression Omnibus and are accessible through GEO accession number GSE96833.

**Partial skeletal and full non-skeletal DNA methylation maps**. Osteoblast RRBS map, extracted from the femur, tibia, and rib bones of a 6-year-old female (NHOst-Osteoblasts by Lonza Pharma, product code: CC-2538, lot number: 6F4124), was downloaded from GEO accession number GSE27584. Forty-eight 450K methylation array maps, extracted from the femora of adult males and females with osteoarthritis or osteoporosis, were downloaded from GEO accession number GSE64490. Four 450K methylation array maps, extracted from unspecified bones of adult males and females were downloaded from GEO accession number GSE50192. Chimpanzee and human WGBS blood methylation maps were downloaded from NCBI SRA accession number SRP059313. Chimpanzee and human WGBS brain maps were downloaded from GEO accession number GSE37202.

**Bisulfite-PCR of human bone**. A skull of an adult male from India was obtained from the teaching anatomy collection of the Department of Anatomy and Anthropology at the Sackler Faculty of Medicine, Tel Aviv University, Israel (Human 1). Additional two skull specimens (Human 2 and 3) were obtained directly from the operating room of the Department of Neurosurgery, Shaare Zedek Medical Center, Jerusalem, Israel and transferred on dry ice for further analysis. All study participants provided informed consent according to an institutional review board—approved protocol (SZMC 0048-18).

Human 1: Standard precautions to avoid contamination were taken, including wearing disposable coats, masks, hair covers and double gloves. All following steps were performed in a UV cabinet dedicated for the preparation of ancient bone samples and located in a physically separated ancient DNA laboratory at the Faculty of Dental Medicine. The skull was cleaned with an excess of 10% bleach (equal to 0.6% Sodium hypochlorite) and then subjected to UV radiation for 30 min The cortical layer on the temporal surface (*facies temporalis*) of the zygomatic bone (ZB) was removed by low-speed drilling using a Wolf Multitool Combitool Rotary Multi Purpose Tool equipped with a sterile dental burr. Another sterile burr was used to obtain powder of the subcortical trabecular bone within the body of the zygoma. The powder was collected onto a $10 \times 10$ cm aluminum foil sheet pretreated with a 10% bleach solution and then transferred into a sterile 1.5 ml Eppendorf tube for subsequent DNA extraction. Altogether, three samples were obtained: ZB-3 from the right zygoma weighing 20.3 mg, and ZB-3/1 and ZB-3/2 from the left zygoma weighing 29.5 mg and 30.3 mg, respectively. Bone DNA was purified from the three bone powder samples using QIAamp DNA Investigator Kit (QIAgen, 56504) according to manufacturer's instructions.

Human 2 and 3: DNA was extracted from bones using QIAamp® DNA Investigator kit (56504, Qiagen). Bones were thoroughly washed (X5) with PBS, to clean samples from blood. Bones were crushed with mortar and pestle in liquid nitrogen, and 100 mg bone powder was taken to extract DNA according to the protocol Isolation of Total DNA from Bones and Teeth of the DNA Investigator kit.

Genomic DNA was bisulfite converted with the EZ DNA Methylation Lightning Kit (Zymo Research, D5030) according to the manufacturer's instructions. Specifically, each bone sample was bisulfite converted using 500 ng as genomic DNA input for the conversion.

Bisulfite treated DNA were amplified with the FastStart High Fidelity PCR System (Sigma, 03553400001) using the primers listed in Supplementary Table 1. PCR conditions were performed according to manufacturer's instructions and PCR products were visualized on a 1.5% agarose gel. Prior to cloning, PCR products

were purified with Gel/PCR DNA Mini Kit (RBC, YDF100) and quantified with a NanoDrop 2000 spectrophotometer.

CloneJET PCR Cloning Kit (Thermo Scientific, K1231) was used to clone the purified PCR products into a pJET1.2/blunt Cloning Vector following the Blunt-End Cloning Protocol described in the manufacturer's instructions. Five microliter of each cloning reaction product were used for transformation of DH5α Competent Cells (Invitrogen, 18265017). Colonies were grown overnight on LB plates containing 100 μg/ml ampicillin. Positive transformants were picked and grown overnight in liquid LB medium containing 100 μg/ml ampicillin. Subsequently, plasmid minipreps were purified with a RBC Miniprep Kit (YPD100) according to manufacturer's instructions. Purified plasmids were quantified with a NanoDrop 2000 spectrophotometer and sequenced on an Applied Biosystems 3730xl Genetic Analyzer (Supplementary Fig. 5a, b).

**Human primary chondrocyte validation**. Primary chondrocyte cultures were obtained from osteoarthritis (OA) donors in accordance with Hadassah Medical Center Institutional Review Board approval and in accordance with the Helsinki Declaration of ethical principles for medical research involving human subjects. End-stage OA patients, with a Kellgren and Lawrence OA severity score of 3–4 were recruited following receipt of a formal written informed consent ($n = 8$; 75% female, mean age $73 \pm 7.2$ years; mean body mass index $30.1 \pm 5.4$ kg/m$^2$). Hyaline articular cartilage was dissected and human chondrocytes isolated using 3 mg/ml Collagenase Type II (Worthington Cat # LS004177) in DMEM medium (Sigma-Aldrich, St Louis, MI) containing 10% FCS and 1% penicillin-streptomycin (Beit-Haemek Kibutz, Israel), 37 °C, 24 h incubation. Isolated cells were filtered through a nylon cell strainer (40 mm diameter), washed three times with PBS and plated at 1.5 million cells per 14 cm$^2$ tissue culture dish (passage 0, passage 2). Cells were cultured in standard incubation conditions (37 °C, 5% CO$_2$) until confluence. Chondrocyte DNA purification was performed using GenElute™ Mammalian Genomic DNA Miniprep Kit (Sigma, G1N350).

**Chimpanzee bone DNA methylation maps**. Overall, we produced six methylation maps from bones of six common chimpanzee (*Pan troglodytes*) individuals. They include one WGBS of a wild chimpanzee, one RRBS of an infant chimpanzee, and four 850K methylation arrays of captive chimpanzees.

Chimpanzee tissue samples included in this study were opportunistically collected at routine necropsy of these animals. No animals were sacrificed for this study, and no living animals were used in this study.

**WGBS of a chimpanzee bone**. We used a rib bone of a 47-year-old female Chimpanzee provided from the Biobank of the Biomedical Primate Research Center (BPRC), The Netherlands. The postmortem interval was approximately 10–12 h. The bone was collected during the necropsy procedure and immediately frozen and stored at –80 °C.

DNA was extracted in a dedicated ancient DNA laboratory at the Institute of Evolutionary Biology in Barcelona, where no previous work on great apes has ever been conducted. Standard precautions to avoid and monitor exogenous contamination such as frequent cleaning of bench surfaces with bleach, use of sterile coveralls, UV irradiation and blank controls were taken during the process. Two hundred milligram of bone powder were obtained by drilling and the sample was extracted following the Dabney et al.[66] method. A final 25 μl of extract volume was used for genome sequencing.

Analysis was performed similarly to Bone1 and Bone2, with the exception that the BSreads were mapped to bisulfite converted chimpanzee (panTro4) reference genome, and we ignored the first 5bp of read1 and the first 44 bp of read2 in the chimpanzee sample (Supplementary Fig. 1). Methylation data were deposited in NCBI's Gene Expression Omnibus and are accessible through GEO accession number GSE96833.

**RRBS of a chimpanzee bone**. We used two unidentified long bone fragments that belonged to a newborn wild chimpanzee infant who died during a documented infanticide event at Gombe National Park on 9 March 2012. The infant was known to be the offspring of a chimpanzee called Eliza and was partially eaten by an adult female and her family. The sample was collected from the ground about 48 h after the infant's death and stored in RNAlater solution until arrival at Arizona State University (ASU). At ASU the sample was stored at 4 °C until extraction.

Sampling and DNA extractions were conducted at the ASU Ancient DNA Laboratory, a Class 10,000 clean-room facility in a separate building from the Molecular Anthropology Laboratory. Precautions taken to avoid contamination included bleach decontamination and UV irradiation of tools and work area before and between uses, and use of full body coverings for all researchers. The bone samples were pulverized together in December 2012 using a SPEX CertiPrep Freezer Mill. Three DNA extractions were conducted using 50–100 mg of bone powder (Supplementary Table 2) and following the extraction protocol by Rohland and Hofreiter[67]. Two extraction blank controls were included to monitor contamination of the extraction process. One microliter each of the sample extract and the blank control were used for fluorometric quantification with the Qubit 2.0 Broad Range assay. All extracts were combined for a total volume of 345 μl and approximately 0.652 μg of total DNA.

RRBS libraries were generated according to Boyle et al.[68]. 100–200 ng genomic DNA was digested with MspI. Subsequently, the digested DNA fragments were end-repaired and adenylated in the same reaction. After ligation with methylated adapters, samples with different adapters were pooled together and were subjected to bisulfite conversion using the EpiTect Bisulfite kit (QIAGen) per the manufacturer's recommendations with the following modification: after first bisulfite conversion, the converted DNA was treated with sodium bisulfite again to guarantee that conversion rates were no less than 99%. Two third of bisulfite converted DNA was PCR amplified and final RRBS libraries were sequenced in an Illumina HiSeq 2000 sequencer (Supplementary Data 1). Methylation data were deposited in NCBI's Gene Expression Omnibus and are accessible through GEO accession number GSE96833.

**850K DNA methylation arrays**. Four chimpanzee cadavers from captive colonies at the Southwest National Primate Research Center in Texas were used. Femora were opportunistically collected at routine necropsy of these animals and stored in −20 °C freezers at the Texas Biomedical Research Institute after dissection. These preparation and storage conditions ensured the preservation of skeletal DNA methylation patterns.

Samples were then transported to ASU and DNA was extracted from the femoral trabecular bone using a phenol-chloroform protocol optimized for skeletal tissues[69]. From the distal femoral condyles, trabecular bone was collected using coring devices and pulverized into bone dust using a SPEX SamplePrep Freezer/ Mill. Specifically, bone cores were obtained from a transverse plane through the center of the medial condyle on the right distal femur, such that the articular surface remained preserved. Cortical bone was removed from these cores using a Dremel (Supplementary Table 3). Tissue collections were performed at the Texas Biomedical Research Institute, and DNA extractions were conducted at the ASU Molecular Anthropology Laboratory.

Genome-wide DNA methylation was assessed using Illumina Infinium MethylationEPIC microarrays. These arrays analyze the methylation status of over 850,000 sites throughout the genome, covering over 90% of the sites on the Infinium HumanMethylation450 BeadChip as well as an additional 350,000 sites within enhancer regions. For each sample, 400 ng of genomic DNA was bisulfite converted using the EZ DNA Methylation™ Gold Kit according to the manufacturer's instructions (Zymo Research), with modifications described in the Infinium Methylation Assay Protocol. These protocols were conducted at the ASU Molecular Anthropology Laboratory. Following manufacturer guidelines (Illumina), this processed DNA was then whole-genome amplified, enzymatically fragmented, hybridized to the arrays, and imaged using the Illumina iScan system. These protocols were conducted at the Texas Biomedical Research Institute. These array data have been deposited in NCBI's Gene Expression Omnibus and are accessible through GEO Series accession number GSE94677.

Raw fluorescent data were normalized to account for the noise inherent within and between the arrays themselves. Specifically, we performed a normal-exponential out-of-band (Noob) background correction method with dye-bias normalization to adjust for background fluorescence and dye-based biases and followed this with a between-array normalization method (functional normalization) which removes unwanted variation by regressing out variability explained by the control probes present on the array as implemented in the minfi package in R which is part of the Bioconductor project. This method has been found to outperform other existing approaches for studies that compare conditions with known large-scale differences[70], such as those assessed in this study.

After normalization, methylation values ($\beta$ values) for each site were calculated as the ratio of methylated probe signal intensity to the sum of both methylated and unmethylated probe signal intensities. These $\beta$ values range from 0 to 1 and represent the average methylation levels at each site across the entire population of cells from which DNA was extracted (0 = completely unmethylated sites, 1 = fully methylated sites).

$$\beta = \frac{\text{Methylated signal}}{(\text{Methylated signal} + \text{unmethylated signal})}$$

Every $\beta$ value in the Infinium platform is accompanied by a detection P-value, and those with failed detection levels (P-value > 0.05) in greater than 10% of samples were removed from downstream analyses.

The probes on the arrays were designed to specifically hybridize with human DNA, so our use of chimpanzee DNA required that probes non-specific to the chimpanzee genome, which could produce biased methylation measurements, be computationally filtered out and excluded from downstream analyses. This was accomplished using methods modified from Hernando-Herraez et al.[71]. Briefly, we used blastn to map the 866,837 50 bp probes onto the chimpanzee genome (Assembly: Pan_tro_3.0, Accession: GCF_000001515.7) using an e-value threshold of $e^{-10}$. We only retained probes that successfully mapped to the genome, had only 1 unique BLAST hit, targeted CpG sites, had 0 mismatches in 5 bp closest to and including the CpG site, and had 0–2 mismatches in 45 bp not including the CpG site. This filtering retained 622,819 probes.

Additionally, $\beta$ values associated with cross-reactive probes, probes containing SNPs at the CpG site (either human or chimpanzee), probes detecting SNP information, probes detecting methylation at non-CpG sites, and probes targeting

sites within the sex chromosomes were removed using the minfi package in R. This filtering retained a final set of 576,505 probes.

**Bisulfite-PCR of chimpanzee cranial bones**. Postmortem frontal skull bones from two different chimpanzees (chimpanzee 1 and chimpanzee 2) were provided by the Biomedical Primate Research Center (BPRC, The Netherlands). Bones were opportunistically collected during routine necropsy of these animals and stored at −80 °C. Chimpanzee 3 and chimpanzee 4 samples were obtained from the chimpanzee cranial collection in the Department of Paleoanthropology in the Senckenberg Research Institute Frankfurt (DPSF) and Natural History Museum Frankfurt. These two chimpanzee specimens are owned by the Justus Liebig University Gießen.

Chimpanzee 1 and chimpanzee 2: For each sample, bone powder was obtained by crushing the bones with mortar and pestle. Approximately 100 mg bone powder were used to extract DNA using the QIAamp DNA Investigator Kit (Qiagen) following manufacturer's instructions.

Chimpanzee 3 and chimpanzee 4: Cochlear bone powder was obtained by accessing the petrous bone from the cranial base[72]. DNA was extracted from about 50 mg of powder according to the protocol described by Dabney et al.[66], but adapted for the use of High Pure Nucleic Acid Large Volume columns (Roche) instead of the Zymo-Spin V column (Zymo Research) MinElute silica spin column (Qiagen) combination.

Genomic DNA was bisulfite converted with the EZ DNA Methylation— Lightning Kit (Zymo Research, D5030) according to the manufacturer's instructions. Specifically, each bone sample was bisulfite converted two times in parallel using 500ng as genomic DNA input for the conversion.

Three microliter of bisulfite treated DNA were amplified with the FastStart High Fidelity PCR System (Sigma, 03553400001) using the primers listed in Supplementary Table 1. PCR conditions were performed according to manufacturer's instructions and PCR products were visualized on a 1.5% agarose gel. Prior to cloning, PCR products were purified with homemade SPRI beads (chimpanzee 1 and 2) and Gel/PCR DNA Mini Kit (RBC, YDF100, chimpanzee 3 and 4), and quantified with a NanoDrop 2000 spectrophotometer.

CloneJET PCR Cloning Kit (Thermo Scientific, K1231) was used to clone the purified PCR products into a pJET1.2/blunt Cloning Vector following the Blunt-End Cloning Protocol described in the manufacturer's instructions. Three microliter (chimpanzee 1 and 2) and 3 μl (chimpanzee 3 and 4) of each cloning reaction product were used for transformation of DH5α Competent Cells (Invitrogen, 18265017). Colonies were grown overnight on LB plates containing 100 μg/ml ampicillin. Positive transformants were picked and grown overnight in liquid LB medium containing 100 μg/ml ampicillin. Subsequently, plasmid minipreps were purified with a QIAprep Miniprep Kit (Qiagen, chimpanzee 1 and 2), and RBC Miniprep Kit (YPD100, chimpanzee 3 and 4) according to manufacturer's instructions. Purified plasmids were quantified with a NanoDrop 2000 spectrophotometer and sequenced on an Applied Biosystems 3730xl Genetic Analyzer (Supplementary Fig. 5a, b).

**Reconstructing ancient DNA methylation maps**. In a dedicated clean room at Harvard Medical School, powder was extracted from the root of a lower third molar of the Mesolithic La Braña 1 individual (5983–5747 calBCE (6980 ± 50 BP, Beta-226472)), from which a non-UDG-treated library was previously sequenced to 3.5× coverage[8]. Two UDG-treated libraries from the same individual were later generated and enriched for approximately 1.2 million single targeted polymorphisms and sequenced to an average of 19.5× coverage at these position[9]. In this study, we carried out shotgun sequencing of one of the same UDG-treated libraries from this individual on a NextSeq500 instrument using 2 × 76 bp paired end sequences. Following the mapping protocol described previously[9], we trimmed adapter sequences, only processed read pairs whose ends overlapped by at least 15 bp (allowing for one mismatch) so that we could confidently merge them, and then mapped to the human reference sequence hg19 using the command samse in BWA (v0.6.1). We removed duplicated sequences by identifying sequences with the same start and stop position and orientation in the alignment, and picking the highest quality one. After restricting to sequences with a map quality of MAPQ ≥ 10, and sites with a minimum sequencing quality (≥20), we had an average coverage measured at the same set of approximately 1.2 million single nucleotide polymorphism targets of 23.0×. This data is available under GEO accession number: GSE96833, with raw reads deposited under SRA accession number: SRX3194436.

In a dedicated clean room at the University College Dublin, powder was extracted from the cochlear portion of the petrous bone of individual I1583 (archeological ID L14-200) from the site of Barcın Höyük in the Yenişehir Plain of the Marmara Region of Northwest Turkey. The Neolithic individual came from a community that practiced farming, and was anthropologically determined to be a male aged 6–10 years at the time of death (the sex was confirmed genetically). The direct radiocarbon date was 6426-6236 calBCE (7460 ± 50 BP, Poz-82231). In a dedicated clean room at Harvard Medical School, a UDG-treated library was prepared from this powder, which was previously enriched for about 1.2 million SNP targets, sequenced to 13.5× average coverage, and published in ref. [9]. We shotgun sequenced the same library on nine lanes of a HiSeqX10 sequencing with 100bp paired reads. On data processing, we merged overlapping read pairs,

trimmed Illumina sequencing adapters, and dropped read pairs that did not have sample barcodes (up to one mismatch) or cannot be unambiguously merged. We then aligned merged reads with BWA against human reference genome GRCh37 (hg19) plus decoy sequences, and combined all nine lanes of data and removed duplicate molecules, achieving an average of 24.3× coverage evaluated on the 1.2 million targets. This data is available under GEO accession number: GSE96833, with raw reads deposited under SRA accession number: SRX3194436.

**The reconstruction procedure**. Reconstruction of DNA methylation maps was performed on the genomes of the following individuals: Ust'-Ishim[6], Loschbour[7], Stuttgart[7], La Braña 1, I1583, and the Vindija Neanderthal[5], as well as on the previously published Altai Neanderthal and the Denisovan (Supplementary Data 1). The Vindija Neanderthal reads were downloaded from the Max Planck Institute for Evolutionary Anthropology website: http://cdna.eva.mpg.de/neandertal/Vindija/bam/. Only the UDG-treated portion of the genome (B8744) was used. Additional UDG-treated ancient human full genomes have been published to date; however, these were sequenced to a relatively low coverage (<5×), and thus, only crude methylation maps could be reconstructed from them. C→T ratio was computed for every CpG position along the hg19 (GRCh37) human genome assembly, for each of the samples[4].

In order to exclude from the analyses positions that potentially represent pre-mortem C→T mutations rather than post-mortem deamination, the following filters were applied: (i) Positions where the sum of A and G reads was greater than the sum of C and T reads were excluded. (ii) For genomes that were produced using single-stranded libraries (i.e., Ust'-Ishim, Altai Neanderthal, Denisovan, Vindija Neanderthal and ~1/3 of the Loschbour library) positions, where the $G \rightarrow A$ ratio on the opposite strand was greater than 1/(average single strand coverage) were excluded. This fraction represents a threshold of one sequencing error allowed per position. For Loschbour, this was performed only on the fraction of reads that came from the single stranded library. (iii) For all genomes, positions with a C→T ratio >0.25 were discarded. For the Vindija Neanderthal, this threshold was raised to 0.5, due to its relatively low coverage (~7×). (iv) Finally, a maximum coverage threshold of 100 reads was used to filter out regions that are suspected to be PCR duplicates.

In all genomes, excluding Vindija, a fixed sliding window of 25 CpGs was used for smoothing of the $C \rightarrow T$ ratio. This allowed for an unbiased scanning of differentially methylated regions (DMRs) that is not affected by the size of the window. Due to its relatively low coverage, we extended the sliding window used on the Vindija genome to 50 CpGs. This extended window is not expected to introduce a bias, as this genome was not used for DMR detection, but only for subsequent filtering that was applied equally to all genomes (see later).

$C \rightarrow T$ ratio was translated to methylation percentage using linear transformation determined from two points: zero $C \rightarrow T$ ratio was set to the value 0% methylation, and mean $C \rightarrow T$ ratio in completely methylated (100% methylation) CpG positions in modern human bone reference (hereinafter $\mu_{100}$) was set to the value 100% methylation. Positions where $C \rightarrow T$ ratio > $\mu_{100}$ were set to 100% methylation. For genomes that were extracted from bones, the modern Bone 2 WGBS map, which is the one with the higher coverage between the two WGBS modern bone maps, was used to determine $\mu_{100}$. For genomes that were extracted from teeth, there was no available modern reference methylation map, and therefore, we transformed the $C \rightarrow T$ ratio into methylation percentage based on the assumption that the genome-wide mean methylation is similar to bone tissue. Thus, the genome-wide mean $C \rightarrow T$ ratio represents 75% methylation, which is the genome-wide mean of measured methylation in the Bone 2 reference map. This was accomplished by setting $\mu_{100}$ to 1.33× mean genome-wide $C \rightarrow T$ ratio.

**DMR detection**. The DMR detection algorithm is comprised of five main steps. We hereby provide an overview of the algorithm followed by a detailed description of each step. The overall goal of this pipeline is to detect differential methylation, assign it to the lineage on which it arose and filter out within-lineage variation.

Step 1: Two-way comparisons. To avoid artifacts that could potentially be introduced by comparing DNA methylation maps that were produced using different technologies, our core analysis relied on the comparison of the three reconstructed maps of the Altai Neanderthal, Denisovan, and Ust'-Ishim. Each of the samples was compared to the other two in a pair-wise manner, as a raw $C \rightarrow T$ ratio map against a reconstructed methylation map, and vice versa. This reciprocal comparison insured that the reconstruction process does not introduce biases to one of the groups. The minimum methylation difference threshold was set to 50%, spanning >50 CpGs.

Step 2: Three-way comparisons. This step classifies to which of the three hominins the DMR should be attributed. This step is done by overlapping the three lists of DMRs found in Step 1. For example, a DMR that is detected between the Neanderthal and Ust'-Ishim and also between the Denisovan and Ust'-Ishim is considered specific to Ust'-Ishim.

Step 3: FDR filtering. Various factors could introduce noise to the reconstruction process, including the stochasticity of the deamination process, the use of a sliding window, and variations in read depth within a sample. We ran simulations that mimic the post-mortem degradation processes of ancient DNA, then reconstructed methylation maps from the simulated deamination maps and

finally compared them to the original map and identified DMRs. Any differences in methylation levels between the simulated map and the original reference map stem from noise. Thus, running the same DMR-detection algorithm on the simulated map vs. the reference map, enables an estimation of the false discovery rate. We set the DMR-detection thresholds so that FDR < 0.05.

Step 4: Lineage assignment. The chimpanzee methylation maps were used to polarize the DMRs. For each DMR, methylation levels in the chimpanzee were compared to those of the three hominin groups. For example, if methylation levels in the chimpanzee samples clustered with the archaic humans, the DMR was assigned to the AMH lineage.

Step 5: Within-lineage variability filtering. To determine whether a DMR represents an individual within a group, or is shared by the entire group, we used a total of 67 AMH, archaic and chimpanzee methylation maps. We used a conservative approach, where DMRs in which methylation levels in one group overlap (even partially) the methylation levels in another group were discarded. As 59 out of the 67 maps belong to AMHs, our ability to filter out variation within this group was better, resulting in fewer DMRs along this lineage. Several various measures were used to ascertain that a DMR along a lineage does not represent a sex-specific, bone-specific, age-specific, technology, or disease-specific DMR.

**DMR-detection algorithm**. We developed an algorithm specifically designed to identify DMRs between a deamination map and a full methylome reference. Let $i$ enumerate the CpG positions in the genome. In the deamination map, let $t_i$ be the number of T's at the C position + the number of A's in the opposite strand at the G position, i.e., it counts the total number of T's that appear in a position that is originally C, in the context of a CpG dinucleotide. We similarly use $c_i$ to count the total number of C's that appear in a position that is originally C, in the context of a CpG dinucleotide. The $C \rightarrow T$ ratio is defined as $t_i/n_i$, where $n_i = c_i + t_i$. Let $\varphi_i$ and $\psi_i$ (both between zero and one) be the methylation of this position in the reference genome and in the reconstructed one, respectively. If we denote by $\pi$ the deamination rate, assumed to be constant throughout the genome, and if we assume that deamination of C into T is a binomial process with probability of success $\pi\psi_i$, we get

$$t_i \sim B(n_i, \pi\psi_i). \tag{1}$$

Our null hypothesis is that the $i$th CpG is not part of a DMR, namely that $\psi_i = \varphi_i$. The alternative hypothesis states that this CpG is part of a DMR. The definition of this statement is that $|\psi_i - \varphi_i| \geq \Delta$, where $\Delta$ is some pre-specified threshold. In other words, under the alternative hypothesis we get that $\psi_i \geq \varphi_i + \Delta$ if the site has low methylation in the reference genome, and $\psi_i \leq \varphi_i - \Delta$ if it has high methylation in the reference genome.

**Per-site statistic**. Let us start with the first option, testing whether $\psi_i \geq \varphi_i + \Delta$ when $\varphi_i$ is low. A log-likelihood-ratio statistic would be

$$\ell_i^+ = \ln \frac{\Pr(t_i|n_i, \pi(\varphi_i + \Delta))}{\Pr(t_i|n_i, \pi\phi_i)} = t_i \left[ \ln\left(1 + \frac{\Delta}{\varphi_i}\right) - \ln\frac{1 - \pi(\varphi_i + \Delta)}{1 - \pi\varphi_i} \right] + n_i \ln\frac{1 - \pi(\varphi_i + \Delta)}{1 - \pi\varphi_i}.$$

Similarly, we can test whether $\psi_i \leq \varphi_i - \Delta$ when $\varphi_i$ is high using the log-likelihood-ratio statistic

$$\ell_i^- = \ln \frac{\Pr(t_i|n_i, \pi(\varphi_i - \Delta))}{\Pr(t_i|n_i, \pi\phi_i)} = t_i \left[ \ln\left(1 - \frac{\Delta}{\varphi_i}\right) - \ln\frac{1 - \pi(\varphi_i - \Delta)}{1 - \pi\varphi_i} \right] + n_i \ln\frac{1 - \pi(\varphi_i - \Delta)}{1 - \pi\varphi_i}.$$

We used the value $\Delta = 0.5$ for all samples. The value of $\pi$, the deamination rate, was estimated using the overall $C \rightarrow T$ ratio in CpG positions whose methylation level is 1 in the modern human Bone 2 WGBS methylation map, after exclusion of putative pre-mortem substitutions, as described in the reconstruction procedure section (Supplementary Data 1).

**Detecting DMRs**. The statistics $\ell_i^+$ and $\ell_i^-$ quantify how strongly the estimated methylation in position $i$ deviates from $\varphi_i$. Next, we use these values to identify DMRs using the cumulative-sum procedure explained below. The process is repeated twice: on the statistic $\ell_i^+$ to identify DMRs where the sample has elevated methylation with respect to the reference, and on the statistic $\ell_i^-$ to identify DMRs where the sample has reduced methylation with respect to the reference.

For convenience, we explain the cumulative-sum procedure in the context of $\ell_i^+$, but an essentially identical procedure is used for $\ell_i^-$. We define a new vector $Q^+$ by the recursion

$$Q_0^+ = 0, \quad Q_i^+ = \max(Q_{i-1}^+ + \ell_i^+, 0).$$

Under the null hypothesis, $\ell_i^+$ has a negative expectation which produces a negative drift that keeps $Q^+$ at zero, or close to zero, levels. Under the alternative hypothesis the expectation is positive, hence the drift over a DMR is positive, leading to an elevation in the values of $Q^+$. Therefore, our next step is to find all intervals $[a, b]$ such that $Q_{a-1}^+ = 0$, $Q_{b+1}^+ = 0$, and $Q_i^+ > 0$ for $a \leq i \leq b$. Let $Q_m^+$ be

the maximum value of $Q^+$ in this interval, where $m$ is the position of the maximum. Then, the interval $[a, m]$ would be called a putative DMR.

The statistics $\ell_i^+$ and $\ell_i^-$ are affected by the number of observed cytosine reads, and thus have higher power to detect hypermethylation (i.e., larger number of cytosine reads) vs. hypomethylation (Supplementary Fig. 3).

**Filtering DMRs**. Of course, $Q^+$ may increase locally due to randomness, and thus a putative DMR may not reflect a true DMR. To filter out such intervals, we used two strategies. First, we applied a set of filters to assure that the putative DMRs have reasonable biological properties. Second, we cleaned the remaining putative DMRs by applying a false discovery rate (FDR) procedure. In the first strategy, we applied two filters: (i) Putative DMRs that harbor less than 50 CpG positions, thus are shorter than twice the smoothing window size, were removed. (ii) To avoid situations where two consecutive CpG sites whose genomic locations are remote appear on the same DMR, we modify the vector $Q_i^+$ as follows. Let $d_{i,j}$ be the distance along the genome (in nucleotides) between CpG sites $i$ and $j$. Then, for every site $i$ such that $d_{i,i-1} > \delta$ we set $Q_i^+ = 0$. We used $\delta = 1000$ nt for all samples.

To further remove putative DMRs that are unlikely to reflect true DMRs, we eliminated all DMRs where $Q_m^+ < Q_T^+$. Here, $Q_m^+$ is the maximum value of $Q^+$ in the interval as defined earlier, and $Q_T^+$ is a threshold determined using a false discovery rate (FDR) procedure, see the Filtering out noise section below.

**Testing the algorithm**. To verify that the approach above results in a low number of false positives, we applied the procedure to deamination maps, when compared to themselves in the form of reconstructed methylomes. As expected, we obtained a negligible number of DMRs, ranging between 0.4 and 1% of the number of DMRs detected between the humans.

**Two-way DMR detection**. In order to avoid artifacts that could potentially be introduced by comparing DNA methylation maps that were produced using different technologies, our core analysis relied on the comparison of the three reconstructed maps of the Altai Neanderthal, Denisovan, and Ust'-Ishim. These are all high-resolution maps that were derived from genomes sequenced to high coverage (Supplementary Data 1). In particular, the Ust'-Ishim methylome is of exceptional quality due to its high coverage and deamination rate (Supplementary Data 1). Also, going through the same post-mortem degradation processes, the Ust'-Ishim cellular composition is likely to be similar to that of the Neanderthal and Denisovan.

In order for a deamination map to serve as a reference in the comparison, we have transformed its $C \rightarrow T$ ratio values into methylation values (see The reconstruction procedure section above). To remove potential bias that could be introduced through the comparison of a reconstructed methylation map to a deamination map, we ran each two-way comparison twice: once with the methylation map of sample 1 against the deamination map of sample 2, and once with the deamination map of sample 1 against the methylation map of sample 2 (Supplementary Fig. 3). Therefore, the comparison of three genomes required a total of six two-way comparisons: Ust'-Ishim versus an Altai Neanderthal reference, Ust'-Ishim versus a Denisovan reference, Altai Neanderthal versus an Ust'-Ishim reference, Altai Neanderthal versus a Denisovan reference, Denisovan versus Ust'-Ishim reference, and Denisovan versus Altai Neanderthal reference. Because the DNA of these three individuals was extracted from both sexes, the DMR-detection algorithm was only applied to autosomes.

**Three-way DMR detection**. In order to identify DMRs where one group of humans (hereinafter, hominin 1) differs from the other two human groups (hereinafter, hominin 2, and hominin 3), we set out to find those DMRs that were detected both between hominin 1 and 2, and between hominin 1 and 3. To this end, we compare the two lists (hominin 1 vs. hominin 2 and hominin 1 vs. hominin 3) and look for overlapping DMRs[4]. An overlapping DMR exists when a DMR from one list partially (or fully) overlaps a DMR from the second list. Only the overlapping portion of the two DMRs from the two-lists was taken.

**Filtering out noise**. There are different factors that potentially introduce noise into the reconstruction process. These include the stochasticity of the deamination process, the use of a sliding window to smooth the $C \rightarrow T$ signal, and variations in read depth. In order to account for these factors and estimate noise levels, we ran simulations that mimic the post-mortem degradation processes of ancient DNA, then reconstructed methylation maps from the simulated deamination maps and finally compared them to the original map and identified DMRs.

The simulation process starts with a methylation map, where the measured or reconstructed methylation at position $i$ is $\psi_i$ and is assumed the true methylation. Given that $n_i$ is the coverage at this position, we use the binomial distribution (1) to randomly draw $t_i$—the number of C's that had become T's through deamination. The resulting $t_i$'s were then used to compute the $C \rightarrow T$ ratios for each position, smoothed and filtered using the same sliding window and thresholds used in the original analysis, and linearly transformed to methylation percentages as explained above (hereinafter, simulated methylation map, Supplementary Fig. 2a). Any differences in methylation levels between the simulated map and the original

reference map stem from noise. Thus, running the same DMR-detection algorithm described above on the simulated map vs. the reference map, enables an estimation of the false discovery rate. We ran these simulations 100 times for each of the three genomes (Altai Neanderthal, Denisovan, Ust'-Ishim) and determined the values of the $Q_T^+$ and $Q_T^-$ thresholds (see the Filtering DMRs section above) such that the mean number of DMRs that are detected in the simulations is < 0.05 the number of real DMRs detected (i.e., FDR < 0.05).

**DMRs separating chimpanzees and humans**. To identify DMRs that separate chimpanzees from all human groups (both modern and archaic), we first compared the chimpanzee WGBS bone methylome to each of two present-day WGBS maps (those of Bone1 and Bone2). This was done by scanning the chimpanzee map using a sliding window of 25 CpGs, in intervals of one CpG position. In each window, we counted the number of methylated and unmethylated reads in each sample, and computed a $P$-value using Fisher's Exact test. We then computed FDR-adjusted $P$-values for each window, and discarded windows with FDR > 0.05 or where the mean methylation difference ($\Delta$) was below 0.5. We then merged overlapping windows. This left 8040 DMRs between the chimpanzee and Bone1, and 12,666 DMRs between the chimpanzee and Bone2. Next, we intersected the two lists to identify DMRs where the chimpanzee differs from the both present-day samples. This left 6417 DMRs. Lastly, we compared the chimpanzee methylation levels to all other human samples (modern and archaic) and filtered out DMRs where the chimpanzee is found within the range of methylation levels observed in humans. To do so, we followed the procedure described in the removing DMRs with high within-group variability section below. This resulted in 2031 DMRs that separate chimpanzees and humans.

**Determining the lineages where DMRs originated**. DMRs where Ust'-Ishim differs from the Neanderthal and the Denisovan could either arose on the AMH branch, or in the ancestor of Neanderthals and Denisovans. In order to allocate the DMRs to the branch in which the change occurred, we used the chimpanzee DNA methylation data.

First, we used the chimpanzee bone WGBS map. We defined the distance of a DMR in hominin $H$ to chimpanzee as the mean absolute difference in methylation, $d_{H,C} = \sum_{i \in DMR} |\psi_i^H - \psi_i^C|$. Here, $\psi_i^H$ is the reconstructed methylation at the $i$'th CpG in hominin $H$, and $\psi_i^C$ is the measured methylation in the same site in the chimpanzee. For Ust'-Ishim-specific DMRs, we used the following procedure: (i) If both archaic humans were closer to the chimpanzee, the DMR was placed on the AMH branch. (ii) If Ust'-Ishim was closer than both archaic humans to the chimpanzee, the DMR was placed on the branch of the ancestor of Neanderthals and Denisovans. (iii) Otherwise, the DMR was discarded. Out of 5111 Ust'-Ishim-specific DMRs, we could place 1729 DMRs on the AMH branch and 1255 on the branch of the ancestor of Neanderthals and Denisovans. 1807 Ust'-Ishim-specific DMRs were discarded due to inconclusive lineage assignment, and 320 had no data in the chimpanzee WGBS map. For Neanderthal-specific DMRs, we discarded all DMRs where Ust'-Ishim and the Denisovan were not found to be closer to the chimpanzee than the Neanderthal. Out of 3107 Neanderthal-specific DMRs, 693 were placed on the Neanderthal branch, 2202 were deemed inconclusive and were discarded, and 212 had no data in the chimpanzee WGBS map. Similarly, we discarded Denisovan-specific DMRs where Ust'-Ishim and Altai Neanderthal were not found to be closer to the chimpanzee than the Denisovan. Out of 1461 Denisovan-specific DMRs, 499 were placed on the Denisovan branch, 855 were deemed inconclusive, and for 107 we had no data in the chimpanzee WGBS map.

We next developed a second, stricter, scheme by also using the chimpanzee 850K DNA methylation arrays datasets. As the probes cover just part of the CpGs in a DMR, we need to adjust the DMR methylation level in order to allow a meaningful comparison of 850K methylation data to full methylation maps. If we mark by $j$ the CpGs in a DMR that are covered by 850K methylation array (which is a subset of all the CpGs in this DMR), and mark their total number by $J = \sum_{j \in DMR} 1$, then the methylation in the DMR as measured by the array is $m = 1/J \cdot \sum_{j \in DMR} \psi_j^{array}$, where $\psi_j^{array}$ is the methylation level measured at position $j$ in the array. Let $m_I = \sum_{i \in DMR} \psi_i^{WGBS}$ be the methylation of this DMR as computed from the full methylation map, where $\psi_i^{WGBS}$ is the methylation level measured at position $i$ in the full map. Let $m_J = \sum_{j \in DMR} \psi_j^{WGBS}$ be the methylation as computed from the full methylation map when limited only to positions $j$. Then, we correct the array methylation value $m$ to:

$$m' = \min\left(m \cdot \frac{m_I}{m_J}, 1\right). \qquad (2)$$

This procedure was applied to DMRs covered by at least one probe (~65% of DMRs). For the remaining ~35% of DMRs, we only used the WGBS chimpanzee methylome. This approach was used in parallel with filtering DMRs using the modern human 450K arrays (Supplementary Fig. 3, see next section).

There are pros and cons to each of these approaches. Using more chimpanzee datasets allow for more informative process. However, 850K methylation array probes are distributed unevenly across the genome. Although most DMRs are covered by at least one probe (mean number of probes per DMR: 1.7, median: 1, maximum: 64), many are nonetheless not covered. AMH

On one hand, lineage assignment of DMRs for which we have array data is more robust and less prone to misclassification. On the other hand, DMRs with array data are more likely to be filtered out, as there is more power to detect variability. This could potentially alter the genomic distribution of DMRs. Therefore, we use both approaches throughout the paper. In analyses where it is important to maintain an unbiased distribution of DMRs we only use the chimpanzee WGBS map for polarization, and AMH bone WGBS maps for filtering (see next chapter), whereas in analyses where it is more important to minimize variability, or where we look at specific DMRs, we use the stricter approach. The chimpanzee RRBS data was adjusted using the same technique. However, it was not used for lineage assignment, but rather only as a source for additional information on DMRs. This is because this protocol particularly targets unmethylated CpGs, and is therefore too biased for lineage assignment.

**Removing DMRs with high within-group variability**. Our three-way DMR detection algorithm above produces a list of DMRs where one of the three hominins (Ust'-Ishim, Altai Neanderthal, or Denisovan) is significantly different from the other two. However, such DMRs could stem from variability within any of the groups, and in such cases cannot be regarded as truly differentiating between the human groups. Some variability may be removed during the process described above (see the Determining the lineages where DMRs originate section), but even DMRs whose origin can be assigned to a particular lineage do not necessarily represent fixed methylation changes. To filter out regions that are variable within any of the human groups, or across all of them, we used two approaches. First, we used the two modern human WGBS maps, and the I1583 reconstructed skull methylation map. DMRs where the Neanderthal or Denisovan methylation levels were found within the range of modern human methylation (i.e., Ust'-Ishim, the two WGBS maps and I1583) were discarded. This left 1530 out of 1729 Ust'-Ishim-derived DMRs (hereinafter, full AMH-derived DMRs), 1230 out of 1255 DMRs where the Neanderthal and Denisovan are both derived, 692 out of 693 full Neanderthal-derived DMRs, and 496 out of 499 Denisovan-derived DMRs.

The second approach adds to this the 52 450K methylation array samples, as well as the three reconstructed methylation maps from teeth (i.e., Loschbour, Stuttgart, and La Braña 1). As described above, using also methylation probes for filtering DMRs provides more power, but can also introduce biases. Thus, this filtering was used for most analyses, except those where unbiased genomic distribution of DMRs is critical. Probe methylation data was corrected as described in Eq. (2). Within AMH- and archaic-derived DMRs, a DMR was deemed fixed if the Neanderthal and the Denisovan methylation levels both fell outside the range of methylation across all modern human samples (reconstructed, WGBS and 450K maps). Similarly, within Neanderthal-derived and Denisovan-derived DMRs, a DMR was deemed fixed if the respective hominin fell outside the range of methylation across all modern human samples and the other archaic hominin. This approach yielded 873 AMH-derived DMRs (hereinafter referred to as AMH-derived DMRs), 939 archaic-derived DMRs, 570 Neanderthal-derived DMRs, and 443 Denisovan-derived DMRs.

The limited number of archaic human methylation maps introduces asymmetry in our ability to determine the level of fixation of DMRs along different lineages. Whereas we used dozens of AMH skeletal samples, we have just a few archaic samples. This provides us with the ability to better estimate the distribution of methylation values within each DMR in AMH, and thus to determine how significantly methylation values in other samples deviate from it. To enhance our ability to estimate variability within archaic human lineages, we added to the analysis the reconstructed methylation map of the Vindija Neanderthal. The USER-treated portion of this genome (the portion amenable for methylation reconstruction) was sequenced[5] to a depth of 7×. Therefore, the methylation map that could be reconstructed from this individual has a considerably lower resolution compared to the other reconstructed maps used in this study (coverage 19× to 52×). Nevertheless, due to the reduced ability to detect variability along the archaic human linages, we employed this map for additional variability filtering along these lineages. DMRs where the Vindija Neanderthal clustered with the other hominins, and not with the Altai Neanderthal (or not with either of the archaic humans in the archaic-derived DMRs) were discarded. The number of DMRs mentioned throughout this chapter already includes this filtering.

A general concern in working with DNA methylation data is that DMRs that are specific to one group do not necessarily represent an evolutionary change, but rather reflect a characteristic such as technology used to measure methylation, tissue, sex, disease or age that is shared by individuals in this group and not by others. We take two complementary approaches to ascertain that the DMRs we report are not driven by these factors: (a) for the top DMGs, we match the samples for the above factors and test whether the hypermethylation of AMHs is still observed. To this end, we compared Ust'-Ishim (adult femur with no known diseases, methylation map produced using our reconstruction method) to the Vindija Neanderthal (adult femur with no known diseases, methylation map produced using our reconstruction method), and we also compared 52 modern human samples (adult femora, methylation array maps) to four chimpanzee samples (adult femora, methylation array maps). In all cases, AMHs show significant hypermethylation compared to the matched samples (Supplementary Fig. 3c, d, see the Methylation in AMH, chimpanzee and Neanderthal femora chapter for additional information). (b) throughout the pipeline, we take only

DMRs where one human group clusters completely outside the other groups regardless of tissue, sex, disease, age or technology. Thus, these factors are unlikely to drive the reported methylation changes. This approach is particularly useful in AMH-derived DMRs, where each group of samples (i.e., AMH samples vs. archaic and chimpanzee samples) include both males and females, juveniles and adults, and they come from femora, ribs, tibia, skulls, and teeth. Thus, it is unlikely that the DMRs that differentiate these groups reflect variability that stems from these parameters[43] (Fig. 1a–c). Archaic-derived DMRs and Neanderthal-derived DMRs are also unlikely to reflect differences in the above parameters, as in these DMRs, the Vindija Neanderthal sample (adult, femur bone) is clustered with the Altai Neanderthal sample (juvenile/adult, phalanx), and not with AMHs, where most samples are from femora of adult females. Denisovan-derived DMRs, on the other hand, are more likely to stem from age or bone type differences than other types of DMRs. This is because the Denisovan sample is the only finger bone, and it comes from a child (6–13.5 years) (Supplementary Data 1). Thus, we cannot rule out the possibility that some of the Denisovan-derived DMRs reflect finger-specific, rather than lineage-specific methylation patterns. These DMRs could also possibly reflect age-specific differences, but this is less likely, as the AMH I1583 sample[9] and the chimpanzee 850K samples are the same age group as the Denisovan (Denisovan: 6–13.5 years old, I1583: 6–10, chimpanzees: 10–13) but show different methylation patterns than the Denisovan (Supplementary Data 1, Fig. 1).

Note that we do not generally expect the number of DMRs along a lineage to be proportional to the length of the lineage, as this number is determined by several factors. First, the statistical power to detect DMRs depends on coverage and deamination levels. Thus, our ability to detect DMRs was lowest in the Denisovan, and highest in Ust'-Ishim. Second, the ability to filter out within-population variability was substantially higher along the AMH lineage, to which most samples belong. While filtering out such variability, we also exclude variability that exists across both AMH and archaic populations. This filtering also discards genomic regions that are variable between sexes, bone types and regions where methylation patterns tend to be more stochastic. Variability that exists exclusively along the Neanderthal lineage was partially removed using the Vindija Neanderthal sample, which comes from a different bone (femur vs. phalanx) and age (adult vs. juvenile/adult). Along the Denisovan-lineage, on the other hand, such variability could not be filtered out using our array of samples (Fig. 1).

We also repeated the Gene ORGANizer analyses (see the Gene ORGANizer analysis section) after removal of 20 DMRs that overlap regions which were shown to change methylation during osteogenic differentiation[57]. We show that the enrichment of voice-affecting genes holds, and thus, the differentiation state of cells in the samples is unlikely to explain the results we report.

**Comparison to previous reports**. We have previously reported that compared to present-day humans, the HOXD cluster of genes is significantly hyper-methylated in the Neanderthal and Denisovan samples[4]. Using the current methylation maps, we show that this observation holds (Supplementary Fig. 2b). Adding chimpanzee data, we see that similarly to AMHs, chimpanzee samples are also hypomethylated compared to archaic humans. This suggests that the hypermethylation arose along the archaic-human lineage. However, we find that the Ust'-Ishim individual is an outlier among modern humans, and that his methylation levels are closer to the Neanderthal than to modern humans, as was also shown by Hanghøj et al.[73]. The Neanderthal and Ust'-Ishim individuals are found >2 standard deviations from the mean observed methylation in modern humans. This suggests that although the Neanderthal is hypermethylated compared to almost all modern humans, she is not found completely outside modern human variation. The Denisovan, on the other hand, is found even further away, and significantly outside the other populations. Given this, the HOXD DMR was classified as Denisovan-derived (Supplementary Data 2). The Ust'-Ishim remains include a single femur, and to our knowledge, it was not compared morphologically to other humans. Thus, further analysis is needed in order to determine whether the hypermethylation of the Ust'-Ishim individual compared to other AMHs is manifested in morphological changes as well. Moreover, as this DMR is classified as Denisovan-derived, we cannot rule out the possibility that it is driven to some extent by age or bone type differences.

Compared to the previously reported DMRs[4], in this study we found four times as many AMH-derived and archaic-derived DMRs (2805 full bone DMRs compared to 891) and roughly twice as many Neanderthal- and Denisovan-derived DMRs (440 and 598 compared to 295 and 307 in the Denisovan and Neanderthal, respectively). The list of DMRs reported here cannot be directly compared to our previous list of DMRs because of several key differences in the analysis: (i) The previous study focused on DMRs that are invariable across tissues, whereas here we focused on DMRs in skeletal tissues. In the previous study, we were therefore able to extrapolate and find trends that extend beyond the skeletal system, such as neurological diseases. In this paper we focus on the skeletal system, hence the different appearance of the body map (Fig. 2b, c). (ii) the current study used stricter thresholds for DMR detection, including a minimum of 50 CpGs in each DMR (compared to ten CpGs previously), and a requirement for physical overlap in the three-way DMR detection procedure. (iii) In this study, the AMH reference is a reconstructed ancient map, whereas in

the previous study the AMH reference, as well as the other tissues used for filtering out noise, were mainly cultured cell lines with RRBS methylation maps.

When filtering DMRs along the lines of the previous study by taking only DMRs with low inter-tissue variability in humans (STD < 10%), we indeed observe similar trends. For example, when taking AMH-derived DMGs and analyzing their expression patterns using DAVID's tissue expression tools, we found that the brain is the most represented organ, with 51.5% of DMGs expressed in this organ (×1.28, FDR = $2.6 \times 10^{-4}$), and glial cells are the most over-represented cell type (×20.6, but FDR > 0.05, UP_TISSUE DB, Supplementary Data 3). In fact, the brain is the only significantly enriched organ in this analysis. Similarly, when analyzing the GNF DB, we found that the subthalamic nucleus is the most enriched body part (×1.60, FDR = $9.2 \times 10^{-4}$), followed by additional brain regions, such as the olfactory bulb (×1.54, FDR = 0.01), globus pallidus (×1.41, FDR = 0.04), and more (Supplementary Data 3). Similar enrichment patterns of the brain can be observed when analyzing expression patterns of all AMH-derived DMGs (Supplementary Data 3). Finally, we also find that similarly to the previous report, these DMRs are linked to diseases more often (23.1% compared to the genome average of 10.8%, DAVID OMIM_DISEASE DB).

**Computing correlation between methylation and expression.** In order to identify regions where DNA methylation is tightly linked with expression levels, we scanned each DMR in overlapping windows of 25 CpGs (the window used for smoothing the deamination signal). In each window we computed Pearson's correlation between DNA methylation and expression levels of overlapping genes as well as the closest genes upstream and downstream genes, across 21 tissues[33]. For each DMR, we picked the window with the best correlation (in absolute value) and computed regression FDR-adjusted $P$-value. DMRs that overlap windows with FDR < 0.05 were considered to be regions where methylation levels are significantly correlated with expression levels. Ninety such DMRs were found among the skeletal AMH-derived DMRs, 93 among the archaic human-derived DMRs, 40 among Neanderthal-derived DMRs, and 19 among Denisovan-derived DMRs.

As no expression data were available for Ust'-Ishim, Bone1 and Bone2, we approximated their *NFIX* expression level by taking the average of *NFIX* expression from three osteoblast RNA-seq datasets that were downloaded from GEO accession numbers GSE55282, GSE85761, and GSE78608. RNA-seq data for chondrocytes was downloaded from the ENCODE project, GEO accession number GSE78607 and plotted against measured methylation levels in primary chondrocytes (see the Human primary chondrocyte validation chapter). Notably, even though the expression and methylation data come from different individuals, plotting them against one another positions them only ~one standard deviation from the expression value predicted by the regression line (Fig. 5b). Future studies providing RNA expression levels for the laryngeal skeleton and vocal folds might provide further information on the methylation-expression links of these genes.

**Studying the function of DMGs.** Gene ontology and expression analyses were conducted using Biological Process and UNIGENE expression tools in DAVID, using an FDR threshold of 0.05.

**Gene ORGANIzer analysis.** Similarly to sequence mutations, changes in regulation are likely to be unequally distributed across different body systems, owing to negative and positive selection, as well as inherent traits of the genes affecting each organ. Thus, we turned to investigate which body parts are affected by the DMGs. To this end, we ran the lists of DMGs in Gene ORGANIzer[11], which is a tool that links genes to the organs they affect, through known disease and normal phenotypes. Thus, it allows us to investigate directly the phenotypic function of genes, to identify their shared targets and to statistically test the significance of such enrichments. We ran the lists of DMGs in the ORGANIze option using the default parameters (i.e., based on confident and typical + non-typical gene-phenotype associations).

When we ran the list of skeletal AMH-derived DMRs, we found 11 significantly enriched body parts, with the vocal folds and the larynx being the most enriched parts (×2.11 and ×1.68, FDR = 0.017 and FDR = 0.046, respectively). Most other parts belonged to the face (teeth, forehead, lips, eyelid, maxilla, face, and jaws), as well as the pelvis and nails (Fig. 2c, d and Supplementary Data 4). For archaic-derived DMGs, the lips, limbs, jaws, scapula, and spinal column were enriched (Supplementary Fig. 4e, Supplementary Data 4). The Neanderthal-derived and Denisovan-derived DMG lists did not produce any significantly enriched organs, but the immune system was significantly depleted within Neanderthal-derived DMRs (×0.67, FDR = 0.040).

In order to examine whether such trends could arise randomly from the reconstruction method, we repeated the analysis on the previously described 100 simulations. We ran all simulated DMGs (4153) in Gene ORGANIzer and found that no enrichment was detected, neither for voice-related organs (vocal folds: ×0.99, FDR = 0.731, larynx: ×1.02, FDR = 0.966, FDR = 0.966), nor for any other organ.

**Validation of face and larynx enrichment in Gene ORGANIzer.** To test whether the enrichment of the face and larynx could be attributed to the fact that the

analyses are based on skeletal tissues, we tested whether the proportion of genes related to the face, larynx, vocal folds, and pelvis within AMH-derived skeleton-related DMGs is higher than expected by chance. Out of 100 skeleton-related DMGs, 31 genes are known to affect the voice, 34 affect the larynx, 87 affect the face, and 65 affect the pelvis, whereas genome-wide these proportions are significantly lower (14.2%, 20.2%, 70.0%, 52.4%, $P = 1.0 \times 10^{-5}$, $P = 1.3 \times 10^{-3}$, $P = 2.1 \times 10^{-3}$, $P = 0.03$, for vocal folds, larynx, face, and pelvis, respectively, hypergeometric test). For additional validation tests, see main text.

Genes associated with craniofacial features were taken from the GWAS-catalog (version 2019-04-21), using a threshold of $P < 10^{-8}$. The following features were used: dental caries, cleft palate, facial morphology, intracranial volume, cleft palate (environmental tobacco smoke interaction), cranial base width, craniofacial macrosomia, facial morphology (factor 1, breadth of lateral portion of upper face), facial morphology (factor 10, width of nasal floor), facial morphology (factor 11, projection of the nose), facial morphology (factor 12, vertical position of sublabial sulcus relative to central midface), facial morphology (factor 14, intercanthal width), lower facial height, nose morphology, nose size, tooth agenesis (maxillary third molar), tooth agenesis (third molar), facial morphology traits (multivariate analysis), lower facial morphology traits (ordinal measurement), lower facial morphology traits (quantitative measurement), middle facial morphology traits (quantitative measurement), and upper facial morphology traits (ordinal measurement). We then tested their overlap with DMGs. Genes associated with craniofacial features in the GWAS catalog significantly overlapped DMGs compared to the fraction expected by chance (5.17×, $P = 3.4 \times 10^{-4}$, hypergeometric test). As a control, we then tested how this 5-fold enrichment compares to non-craniofacial features. We used blood-related GWAS as a representative of general non-craniofacial GWAS. We extracted from the GWAS catalog 22 blood-related traits (the same number as extracted for craniofacial features), by taking the first 22 traits that appear in a search for the term blood and applying a threshold of $P < 10^{-8}$. We then used these genes as a background control for the craniofacial enrichment. We observed a 3.86× enrichment of DMGs with regard to craniofacial-associated vs. non-craniofacial-associated genes ($P = 0.01$, chi-square test).

Additionally, we conducted a permutation test on the list of 129 AMH-derived DMGs that are linked to organs on Gene ORGANIzer, replacing those that are linked to the skeleton with randomly selected skeleton-related genes. We then ran the list in Gene ORGANIzer and computed the enrichment. We repeated the process 100,000 times and found that the enrichment levels we observed within AMH-derived DMGs are significantly higher than expected by chance for the laryngeal and facial regions, but not for the pelvis ($P = 8.0 \times 10^{-5}$, $P = 3.6 \times 10^{-3}$, $P = 8.2 \times 10^{-4}$, and $P = 0.115$, for vocal folds, larynx, face and pelvis, respectively, permutation test, Supplementary Fig. 4b–e).

Potentially, longer genes have higher probability to overlap DMRs. Indeed, DMGs tend to be longer (148 vs. 39 kb, $P = 9.9 \times 10^{-145}$, t-test). We thus checked the possibility that genes affecting the larynx and face tend to be longer than other genes, and are thus more likely to contain DMRs. We found that length of genes could not be a factor explaining the enrichment within genes affecting the larynx, as these genes tend to be shorter than other genes in the genome (mean: 62.5 vs. 73.2 kb, $P = 0.001$, t-test). Genes affecting the face, on the other hand, tend to be longer than other genes (mean: 77.1 vs. 65.6 kb, $P = 4.6 \times 10^{-5}$, t-test). To examine if this factor may underlie the enrichment we observe, we repeated the analysis using only DMRs that are found within promoter regions (5 kb upstream to 1 kb downstream of TSS), thus eliminating the gene length factor. We found that the genes where such DMRs occur are still significantly associated with the face ($P = 0.036$, Fisher's exact test). We next repeated the promoter DMR analysis for all genes and compared the Gene ORGANIzer enrichment levels in this analysis to the genome-wide analysis. We observed very similar levels of enrichment (2.02×, 1.67×, and 1.24×, for vocal folds, larynx, and face, respectively, albeit FDR values >0.05 due to low statistical power). Importantly, AMH-derived DMGs also do not tend to be longer than DMGs on the other branches (148 vs. 147 kb, $P = 0.93$, t-test). Together, these analyses suggest that gene length does not affect the observed enrichment in genes affecting the face and larynx.

Additionally, to test whether cellular composition or differentiation state could bias the results, we ran Gene ORGANIzer on the list of DMGs, following the removal of 20 DMRs that are found <10 kb from loci where methylation was shown to change during osteogenic differentiation[57]. We found that genes affecting the voice and face are still the most over-represented (2.13×, 1.71×, and 1.27×, FDR = 0.032, FDR = 0.049, and FDR = 0.040, for vocal folds, larynx, and face, respectively, Supplementary Data 4).

We also investigated the possibility that (for an unknown reason) the DMR-detection algorithm introduces positional biases that preferentially identify DMRs within genes affecting the voice or face. To this end, we simulated stochastic deamination processes along the Ust'-Ishim, Altai Neandertal, and Denisovan genomes, reconstructed methylation maps, and ran the DMR-detection algorithm on these maps. We repeated this process 100 times for each hominin and found no enrichment of any body part, including the face, vocal folds, or larynx (1.07×, 1.07×, and 1.04×, respectively, FDR = 0.88 for vocal folds, larynx, and face, permutation test). Perhaps most importantly, none of the other archaic branches shows enrichment of the larynx or vocal folds. However, archaic-derived DMGs show over-representation of the jaws, as well as the lips, limbs, scapulae, and spinal column (Supplementary Fig. 4e, Supplementary Data 4). In addition,

DMRs that separate chimpanzees from all humans (archaic and modern, Supplementary Data 2) do not show over-representation of genes that affect the voice, larynx, or face, compatible with the notion that this trend emerged along the AMH lineage. We also sought to test whether the larynx and vocal folds, which we found to be significantly enriched only along the AMH lineage, are also enriched when compared to the other lineages. We ran a chi-squared test on the fraction of vocal folds-affecting and larynx-affecting AMH-derived DMGs (25 and 29, respectively, out of a total of 120 organ-associated DMGs), compared to the corresponding fraction in the DMGs along all the other lineages (42 for vocal folds, 49 for larynx, out of a total of 275 organ-associated DMGs). We found that both the larynx and vocal folds are significantly enriched in AMHs by over 50% compared to the other lineages ($1.57\times$ for both, $P = 0.0248$ and $P = 0.0169$ for vocal folds and larynx, respectively, chi-squared test).

Furthermore, we added a human bone reduced representation bisulfite sequencing (RRBS) map, and produced a RRBS map from a chimpanzee infant unspecified long bone (Supplementary Data 1). RRBS methylation maps include information on only ~10% of CpG sites, and are biased towards unmethylated sites. Therefore, they were not included in the previous analyses. However, we added them in this part as they originate from a chimpanzee infant and a present-day human that is of similar age to the Denisovan (Supplementary Data 1), allowing sampling from individuals that are younger than the rest. Repeating the Gene ORGANizer analysis after including these samples in the filtering process, we found that the face and larynx are the only significantly enriched skeletal regions, and the enrichment within voice-affecting genes becomes even more pronounced ($2.33\times$, FDR = $7.9 \times 10^{-3}$, Supplementary Data 4).

We also examined if pleiotropy could underlie the observed enrichments. To a large extent, the statistical tests behind Gene ORGANizer inherently account for pleiotropy[11], hence the conclusion that the most significant shared effect of the AMH-derived DMGs is in shaping vocal and facial anatomy is valid regardless of pleiotropy. Nevertheless, we tested this possibility more directly, estimating the pleiotropy of each gene by counting the number of different Human Phenotype Ontology (HPO) terms that are associated with it across the entire body[19]. We found that DMGs do not tend to be more pleiotropic than the rest of the genome ($P = 0.17$, t-test), nor do differentially methylated voice-affecting and face-affecting genes tend to be more pleiotropic than other DMGs ($P = 0.19$ and $P = 0.27$, respectively, t-test).

Next, we tested whether the process of within-lineage removal of variable DMRs and the differential number of samples along each lineage biases the Gene ORGANizer enrichment analysis. To do so, we analyzed the pre-filtering DMRs along each lineage. We detect very similar trends to the post-filtering analysis, with the laryngeal and facial regions being the most significantly enriched within AMH-derived DMRs ($1.58\times$, $1.44\times$, and $1.21\times$ to $1.31\times$ for the vocal folds, larynx and different facial regions, respectively, FDR < 0.05), and for archaic-derived DMRs, we detect no enrichment of the laryngeal region (FDR = 0.16 and FDR = 0.43 for the vocal folds and larynx, respectively), and the most enriched regions are the face, limbs, and urethra. With the exception of the urethra, these results are very similar to the results reported for the filtered DMRs, suggesting that the process of within-lineage removal of variable DMRs and the differential number of samples along each lineage does not bias the enrichment results.

Overall, we observe that AMH-derived DMGs across all 60 AMH samples are found outside archaic human variability, regardless of bone type, disease state, age, or sex, and that chimpanzee methylation levels in these DMGs cluster closer to archaic humans than to AMHs, suggesting that these factors are unlikely to underlie the observed trends. Finally, we tested whether the filtering process in itself might underlie the observed trends. To this end, we re-ran the entire pipeline on Neanderthal-derived and Denisovan-derived DMGs, while applying to them all the filters as if they were Ust'-Ishim DMGs. This resulted in substantially fewer loci (89 for the Neanderthal and 50 for the Denisovan), which limits statistical power, but can still be used to examine whether there are any trends of enrichment similar to those observed in AMHs. We found no evidence that the filtering process could drive the enrichment of the vocal or facial areas: within Neanderthal-derived loci, filtered as if they were Ust'-Ishim-derived, we found that the vocal folds were ranked only 18th, with a non-significant enrichment of $1.27\times$ (FDR = 0.815, compared to an enrichment of $2.11\times$ within AMH-derived DMGs). The larynx was ranked 76th and showed a non-significant depletion of $0.87\times$ (FDR = 0.783), and the face was ranked 31st, with a non-significant enrichment of $1.09\times$ (FDR = 0.815). Within Denisovan-derived loci, filtered as if they were Ust'-Ishim-derived, none of the loci were linked to the vocal folds nor to the larynx (FDR = 0.535 and FDR = 0.834, respectively), and the face was ranked 30th ($1.29\times$, FDR = 0.535, Supplementary Data 4). This test suggests that the filtering process in itself is very unlikely to underlie the enrichment of the vocal and facial parts within AMH-derived DMGs.

Next, we applied the Neanderthal/Denisovan filters to the Ust'-Ishim-derived loci. This resulted in 792 loci. We found that the vocal folds remained the most enriched body part ($1.76\times$, FDR = 0.032), the larynx was marginally significant ($1.53\times$, FDR = 0.0502), and the facial region was significantly enriched too (e.g., cheek and chin ranked 2nd, 3rd within significantly enriched body parts, $1.66\times$ and $1.63\times$, FDR = 0.031 and FDR = 0.013, respectively, Supplementary Data 4). Importantly, we do not rule out the option that extensive regulatory changes in genes related to vocal and facial anatomy might have occurred along the Neanderthal and Denisovan lineages as well. Indeed, as we report in

Supplementary Fig. 4e, parts of the face are enriched within Archaic-derived DMGs. However, we currently see no substantial evidence supporting this.

Importantly, the link between genetic alterations and phenotypes related to the voice is complex. Some brain-related disorders (i.e., clinical disorders that affect the brain) result in alterations to the voice, the mechanism in which is very difficult to pin down. Although the mechanism leading to voice alterations (either in its pitch, timbre, volume, or range) in some of the genes we report is unknown, many of the disorders are skeletal, suggesting the mechanism is related to anatomical changes to the vocal tract. Such changes could also affect more primary functions of the larynx, such as swallowing and breathing. However, the enrichment we observe in Gene ORGANizer shows these genes were also shown to drive vocal alterations in the disorders they underlie[11,19]. Voice and speech alterations were also shown to be driven by cultural, dietary and behavioral changes affecting bite configuration[74]. Here too, these factors are unlikely to underlie the vocal alterations in the genes we report, as individuals from the same family as the individual with the disorder, who do not carry the dysfunctional allele, were not reported to present any vocal phenotypes.

The larynx is an organ which is primarily involved in breathing and swallowing in mammals. In humans, the larynx is also used to produce complex speech, but not every change to the larynx necessarily affects speech. Despite these additional functions, the genes reported by Gene ORGANizer and HPO were specifically associated with voice alterations, directly or indirectly, suggesting that although they could have additional effects, their effect on the voice is their most shared function.

**Overlap with enhancer regions**. To further test whether the AMH-derived DMRs overlap skeletal regulatory regions, we examined the previously reported 403,968 human loci, where an enrichment of the active enhancer mark H3K27ac was detected in developing human limbs (E33, E41, E44, and E47)[75]. Each DMR was allocated a random genomic position in its original chromosome, while keeping its original length and matching the distribution of GC-content and CpG density between the original and permutated lists. GC-content and CpG density matching was done by matching a 10-bin histogram of the original and permutated lists. This was repeated for 10,000 iterations. We found that AMH-derived DMRs overlap limb H3K27ac-enriched regions ~$2\times$ more often than expected by chance (610 overlapping DMRs, compared to $312.4 \pm 21.7$, $P < 10^{-4}$, permutation test).

SOX9 upstream putative enhancer coordinates used in Fig. 4b were taken from[13,14,23,76,77].

**Computing the density of changes along the genome**. We computed the density of derived CpG positions along the genome in two ways. First, we used a 100 kb window centered in the middle of each DMR and computed the fraction of CpGs in that window which are differentially methylated (i.e., are found within a DMR). Second, for the chromosome density plots, we did not center the window around each DMR, but rather used a non-overlapping sliding 100 kb window starting at position 1 and running the length of the chromosome.

**NFIX, COL2A1, SOX9, ACAN, and XYLT1 phenotypes**. The vocal tract and larynx affecting genes presented in this paper show involvement in laryngeal cartilage and soft tissue phenotypic variation. Clinical phenotypes can be of high severity, with substantial impacts on normal breathing functions, to the point where the cause of death is due to respiratory distress. SOX9 and NFIX are often associated with laryngomalacia[11,19] (Supplementary Data 5), a collapse of the larynx due to malformation of the laryngeal cartilaginous framework and/or malformed connective tissues, particularly during inhalation. Patients with mutations in COL2A1 often show backwards displacement of the tongue base[11,19]. Less severe phenotypes of the reported genes include variation of voice quality in the form of pitch variation (high in patients suffering from XYLT1 mutations) and sometimes hoarseness of the voice (reported for some patients with mutations of ACAN, Supplementary Data 5)[11,19]. Whether this is due to variation of the vocal tract and laryngeal anatomy influenced by the ACAN mutation or due to a scaled down vocal tract size in the case of the XYLT1 mutation which also causes primordial dwarfism is not yet clear.

**NFIX phenotypes**. Skeletal phenotypes that are associated with the Marshall–Smith syndrome were extracted from the Human Phenotype Ontology (HPO)[19]. Non-directional phenotypes (e.g., irregular dentition) and phenotypes that are expressed in both directions (e.g., tall stature and short stature) were removed.

Mutations in NFIX have also been linked to the Sotos syndrome. However, NFIX is not the only gene that was linked to this syndrome; mutations in NSD1 were also shown to drive similar phenotypes[52]. Therefore, it is less relevant in assessing the functional consequences of general shifts in the activity levels of NFIX. Nevertheless, it is noteworthy that in the Sotos syndrome too, most symptoms are a mirror image of the Neanderthal phenotype (e.g., prominent chin and high forehead).

**Comparing gene expression between AMH and mouse**. Ninety-three appendicular skeleton samples were used to compare expression levels of NFIX, SOX9,

*ACAN*, and *COL2A1* in human and mouse: 1. Five Human expression array data of iliac bones[78], downloaded from ArrayExpress accession number E-MEXP-2219. 2. Eighty-four human expression array of iliac bones, downloaded from ArrayExpress accession number E-MEXP-1618. 3. Three mouse expression array data of femur and tibia bones, downloaded from ArrayExpress accession number E-GEOD-61146. 4. One mouse RNA-seq of a tibia bone, downloaded from supplementary data. Expression values were converted to percentiles, according to each gene expression level compared to the rest of the genome across each sample (Fig. 5c).

**Methylation in AMH, chimpanzee, and Neanderthal femora**. To check whether the AMH hypermethylation of *SOX9*, *ACAN*, *COL2A1*, *XYLT1*, and *NFIX* could be a result of variability between bone types, we compared the four chimpanzee femur 850K methylation arrays to the 52 present-day femur 450K methylation arrays. We took probes within AMH-derived DMRs that appear on both arrays. We found that these genes are consistently hypermethylated in AMHs ($P = 1.6 \times 10^{-7}$, *t*-test), with 38 probes showing >5% hypermethylation in AMH, whereas only eight probes show such hypermethylation in chimpanzees (Supplementary Fig. 5d). Therefore, even when comparing methylation from the same bone, same sex, same developmental stage, measured by the same technology, and across the same positions, AMH show consistent hypermethylation across all of these DMGs.

Similarly, when comparing the DMRs in *SOX9*, *ACAN*, *COL2A1*, *XYLT1*, and *NFIX* between the Ust'-Ishim and Vindija Neanderthal samples, the Vindija Neanderthal sample is consistently hypomethylated compared to the Ust'-Ishim individual ($P = 1.2 \times 10^{-5}$, Supplementary Fig. 5c, *t*-test). Both of these samples were extracted from femora of adult individuals, and methylation was reconstructed using the same technology. This suggests that the hypermethylation of AMHs compared to Neanderthals is unlikely to be driven by age or bone type, and rather reflects evolutionary shifts.

**Scanning *SOX9* for mutations altering NFI binding motifs**. To examine whether the changes in regulation of *SOX9* could possibly be explained by changes in the binding sites of NFI proteins, we searched for the NFI motif along the gene body and the 350 kb upstream region of *SOX9*. We looked for NFI motifs that exist in the genomes of the Altai and Vindija Neanderthal, as well as in the Denisovan, but were abolished in AMHs. We did not find any evidence of such substitutions.

**Comparison to divergent traits between Neanderthals and AMHs**. To further investigate potential phenotypic consequences of the DMGs we report, we probed the HPO database[19] and compared these HPO phenotypes to known morphological differences between Neanderthals and modern humans[32]. To compile a list of traits in which Neanderthals and AMHs differ, we reviewed key sources that surveyed Neanderthal morphology summarized in Aiello, L. and Dean[12]. We identified traits in which Neanderthals are found completely outside AMH variation, as well as traits where one group is significantly different from the other, but the distribution of observed measurements partially overlap. Non-directional traits (i.e., traits that could not be described on scales such as higher/lower, accelerated/delayed etc.) were not included, as could not be paralleled with HPO phenotypes. The compiled list included 107 phenotypes, 75 of which have at least one equivalent HPO phenotype (4.8 on average). For example, the HPO phenotype Taurodontia (HP:0000679) was linked to the trait Taurodontia, and the following HPO phenotypes were linked to the trait Rounded and robust rib shafts: broad ribs (HP:0000885), hypoplasia of first ribs (HP:0006657), short ribs (HP:0000773), thickened cortex of long bones (HP:0000935), thickened ribs (HP:0000900), thin ribs (HP:0000883), thoracic hypoplasia (HP:0005257). For each skeleton-affecting phenotype, we determined whether it matches a known morphological difference between Neanderthals and AMHs[32]. For example, *Hypoplastic ilia* (HPO ID: HP:0000946) was marked as divergent because in the Neanderthal the iliac bones are considerably enlarged compared to AMHs[12]. We then counted for each gene (whether DMG or not) the fraction of its associated HPO phenotypes that are divergent between Neanderthals and AMHs. We found that four of the top five most differentially methylated skeletal genes (*XYLT1*, *NFIX*, *ACAN*, and *COL2A1*) are in the top 100 genes with the highest fraction of divergent traits between Neanderthals and AMHs (out of a total of 1,789 skeleton-related genes). In fact, *COL2A1*, which is the top ranked DMR (Supplementary Data 2), is also the gene that is overall associated with the highest number of derived traits (63) (Supplementary Data 7). This suggests that these extensive methylation changes are possibly linked to phenotypic divergence between archaic and AMHs.

**Reporting summary**. Further information on research design is available in the Nature Research Reporting Summary linked to this article.

## Data availability
All sequencing and methylation data generated in this work have been deposited in NCBI's Gene Expression Omnibus under GEO accession number GSE96833. All other data and materials are contained in the paper and its supplementary information or available upon request.

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

## Acknowledgements

We would like to thank Sagiv Shifman, Yoel Rak, Rodrigo Lacruz, Erella Hovers, Anna Belfer-Cohen, Achinoam Blau, Iain Mathieson, Philip Lieberman, Daniel Lieberman, and Terry Capellini for their useful advice, Svante Pääbo, Janet Kelso, Kay Prüfer, Johannes Krause, and Anne Pusey for providing data, Sjaak Kaandorp and Christine Kaandorp-Huber from Safari park Beekse Bergen in Netherlands for their cooperation in animal conservation and use for research, and Maayan Harel for illustrations. D.G. is supported by the Clore Israel Foundation. TMB is supported by BFU2017-86471-P (MINECO/ FEDER, UE), U01 MH106874 grant, Howard Hughes International Early Career, Obra Social "La Caixa" and Secretaria d'Universitats i Recerca and CERCA Program del Departament d'Economia i Coneixement de la Generalitat de Catalunya. D.R. is an Investigator of the Howard Hughes Medical Institute and is also supported by an Allen Discovery Center for the Study of Human Brain Evolution funded the Paul G. Allen Family Foundation. C.L.-F. is supported by FEDER and BFU2015-64699-P grant from the Spanish government. R.P. was supported by ERC starting grant ADNABIOARC (263441). R.M.G. and J.M.O. are supported by NYSTEM contract C030133. Funding for the collection and processing of the 850K chimpanzee data was provided by the Leakey Foundation Research Grant for Doctoral Students, Wenner-Gren Foundation Dissertation Fieldwork Grant (Gr. 9310), James F. Nacey Fellowship from the Nacey Maggion-calda Foundation, International Primatological Society Research Grant, Sigma Xi Grant-in-Aid of Research, Center for Evolution and Medicine Venture Fund (ASU), Graduate Research and Support Program Grant (GPSA, ASU), and Graduate Student Research Grant (SHESC, ASU) to G.H. Collection of the chimpanzee bone from Tanzania was funded by the Jane Goodall Institute, and grants from the US National Institutes of Health (AI 058715) and National Science Foundation (IOS-1052693), and facilitated by Elizabeth Lonsdorf and Beatrice Hahn.

## Author contributions

D.G. planned and conducted analyses. L.C. supervised the computational and experimental work. E.M. supervised experiments. L.A.T., B.Y., D.G., and L.C. conceived statistical analyses. D.G., L.C., and E.M. wrote the manuscript. All other authors contributed to the production of data and wrote their respective parts of the manuscript.

## Competing interests

The authors declare no competing interests.

## Additional information

David Gokhman[1][✉], Malka Nissim-Rafinia[1], Lily Agranat-Tamir[1,2], Genevieve Housman[3,4], Raquel García-Pérez[5], Esther Lizano[5], Olivia Cheronet[6], Swapan Mallick[7,8,9], Maria A. Nieves-Colón[3,4], Heng Li[7], Songül Alpaslan-Roodenberg[8], Mario Novak[10,11], Hongcang Gu[7], Jason M. Osinski[12], Manuel Ferrando-Bernal[5], Pere Gelabert[5], Iddi Lipende[13], Deus Mjungu[13], Ivanela Kondova[14], Ronald Bontrop[14], Ottmar Kullmer[15], Gerhard Weber[6], Tal Shahar[16], Mona Dvir-Ginzberg[17], Marina Faerman[18], Ellen E. Quillen[19], Alexander Meissner[7,20,21], Yonatan Lahav[22,23], Leonid Kandel[24], Meir Liebergall[24], María E. Prada[25], Julio M. Vidal[26], Richard M. Gronostajski[12,27], Anne C. Stone[3,4,28], Benjamin Yakir[2], Carles Lalueza-Fox[5], Ron Pinhasi[6], David Reich[7,8,9], Tomas Marques-Bonet[5,29,30,31], Eran Meshorer[1,32][✉] & Liran Carmel[1][✉]

[1]Department of Genetics, The Alexander Silberman Institute of Life Sciences, Faculty of Science, The Hebrew University of Jerusalem, 91904 Jerusalem, Israel. [2]Department of Statistics, The Hebrew University of Jerusalem, 91905 Jerusalem, Israel. [3]School of Human Evolution and Social Change, Arizona State University, Tempe, AZ 85281, USA. [4]Center for Evolution and Medicine, Arizona State University, Tempe, AZ 85287, USA. [5]Institute of Evolutionary Biology (UPF-CSIC), PRBB, 08003 Barcelona, Spain. [6]Department of Evolutionary Anthropology, University of Vienna, 1090 Vienna, Austria. [7]Broad Institute, Cambridge, MA 02138, USA. [8]Department of Genetics, Harvard Medical School, Boston, MA 02115, USA. [9]Howard Hughes Medical Institute, Harvard Medical School, Boston, MA 02115, USA. [10]Institute for Anthropological Research, 10000 Zagreb, Croatia. [11]Earth Institute and School of Archaeology, University College Dublin, Dublin 4, Ireland. [12]Department of Biochemistry, Jacobs School of Medicine and Biomedical Sciences, University at Buffalo, Buffalo, NY 14203, USA. [13]Gombe Stream Research Center, Jane Goodall Institute, Kigoma, Tanzania. [14]Biomedical Primate Research Centre (BPRC), Rijswijk, Netherlands. [15]Department of Palaeoanthropology and Messel Research, Senckenberg Center of Human Evolution and Paleoecology, Frankfurt am Main, Germany. [16]Department of Neurosurgery, Shaare Zedek Medical Center, Jerusalem, Israel. [17]Laboratory of Cartilage Biology, Institute of Dental Sciences, Faculty of Dental Medicine, Hebrew University of Jerusalem, 91120 Jerusalem, Israel. [18]Laboratory of Bioanthropology and Ancient DNA, Institute of Dental Sciences, Faculty of Dental Medicine, The Hebrew University of Jerusalem, 91120 Jerusalem, Israel. [19]Department of Genetics, Texas Biomedical Research Institute, San Antonio, TX 85287, USA. [20]Harvard Stem Cell Institute, Cambridge, MA 02138, USA. [21]Department of Stem Cell and Regenerative Biology, Harvard University, Cambridge, MA 02138, USA. [22]Otolaryngology - Head & Neck Surgery Department, Laryngeal Surgery Unit, Kaplan Medical Center, Rehovot, Israel. [23]The Hebrew University Medical School, Jerusalem, Israel. [24]Orthopaedic Department, Hadassah – Hebrew University Medical Center, Jerusalem, Israel. [25]I.E.S.O. 'Los Salados'. Junta de Castilla y León, León, Spain. [26]Junta de Castilla y León, Servicio de Cultura de León, León, Spain. [27]Genetics, Genomics and Bioinformatics Program, New York State Center of Excellence in Bioinformatics and Life Sciences, Jacobs School of Medicine and Biomedical Sciences, University at Buffalo, Buffalo, NY 14203, USA. [28]Institute of Human Origins, Arizona State University, Tempe, AZ 85287, USA. [29]Catalan Institution of Research and Advanced Studies (ICREA), 08010 Barcelona, Spain. [30]CNAG-CRG, Centre for Genomic Regulation (CRG), Barcelona Institute of Science and Technology (BIST), 08028 Barcelona, Spain. [31]Institut Català de Paleontologia Miquel Crusafont, Universitat Autònoma de Barcelona, Edifici ICTA-ICP, c/ Columnes s/n, Barcelona, Spain. [32]The Edmond and Lily Safra Center for Brain Sciences (ELSC), The Hebrew University of Jerusalem, 91904 Jerusalem, Israel. [✉]email: david.gokhman@mail.huji.ac.il; eran.meshorer@mail.huji.ac.il; liran.carmel@huji.ac.il

