## [Peer Review File · Nature Communications]

Reviewers' Comments:

Reviewer #1:

Remarks to the Author:

Gokhman et al. generated and/or used methylation maps from modern and ancient AMHs, archaic humans (Neanderthals and Denisovans), as well as chimpanzees to detect DMRs that arose after the split between humans and Neanderthals/Denisovans. Methylation patterns are associated with repression of genes involved in cranial and facial morphology in the human lineage, suggesting a role for methylation in shaping the human face and potentially the vocal tract.

The question of whether archaic humans such as Neanderthals and Denisovans and AMHs share the same vocal anatomy is important to test the hypothesis that speech is a unique AMH feature. Morphological studies are limited by the fact that the larynx is mostly composed of cartilage and consequently is not preserved in skeletal remains of archaic humans. Instead, Gokhman et al. propose to use a molecular approach to infer the shaping of craniofacial anatomy and vocal tract in AMH and archaic human lineages. Their approach relies heavily on the method they have developed that leverages patterns of ancient DNA damage to reconstruct methylation maps.

As I was reading the manuscript, I could tick off many of my a priori concerns about tissue- and sex-specificity of DNA methylation, impact of DNA damage on methylated cytosines, DMR detection using different methylome reconstruction methods, correlations between DNA methylation and gene expression, and validation using a cell model and/or existing syndromes. Really, either the authors have been extremely careful with the validation of their empirical and analytical results, or the manuscript has already gone through at least one round of external reviews and the authors have addressed all issues in earnest. As a result, there is not much to critique from a data generation and analysis point of view.

I had a thorough look at the statistical analyses detailed in the methods section because the detection of DMRs from deamination maps and empirical methylomes is likely to be the most problematic analysis. The authors have adopted a rather conservative approach, even though they rely heavily on the use of a "methylome reference" that may itself be slightly biased. Nevertheless, the DMR detection algorithm seems to perform reliably since the comparison between deamination maps and reconstructed methylomes from the same maps lead to a low number of false positives. In addition, the two- and three-way DMR detection strategies are additional precautions that account for the different technologies/methodologies as well as the incompleteness of methylation maps at the genomic level.

In my humble opinion, this study was carefully planned (at least according to how it is presented in the manuscript) and the interpretation of the results is cautious. The NIFX results summarised in Figure 6 are definitely the most striking results because they nicely tie together molecular evolution and pathogenetic mechanisms behind some AMH craniofacial syndromes.

My only minor comments are:

– I suggest to use Anatomically Modern Human (AMH) instead of just Modern Human (MH) to comply with the description used by the large majority of anthropologists.

– Line 175: The authors report "44 of the MH-derived DMRs" out of 873 MH-derived DMRs. It does not seem like much but the permutation test is significant and shows 60.4x enrichment. However, how many DMGs does it represent, i.e., 44 or less?

– Lines 737–738: "The first few bases from each read have been shown to have a slightly higher probability of being called as methylated¹⁸, so we trimmed the first ten bases from each read": This sentence is misleading because reference 18 (Schultz et al. 2015 Nature) does not address the issue of methylation bias at the ends of sequencing reads as far as I know. However, Hansen

et al 2012 Genome Biology shows M-bias plots where methylation bias can occur at the end and/or at the start of reads depending on the bisulfite method used. I suggest to rewrite the sentence to be clearer.

Reviewer #2:

Remarks to the Author:

In a previous paper the authors developed a method to infer CpG methylation based on the DNA damage pattern from ancient DNA. As the methylation profile stems from ancient bones, it is crucial to compare a sufficient number of ancient methylation profiles with modern profiles of humans and outgroups (Chimpanzee) to interpret the ancient profiles. This is what they do in this paper, by inferring methylation profiles from additional published ancient genomes, ancient genomes sequenced for this paper and chimpanzee methylation profiles. Their main conclusion from the analysis of this data, as prominently laid out in the title is "regulatory divergence of genes affecting vocal and facial anatomy in modern humans". While the generated data and method is clearly of interest I have serious doubts about this bold conclusion. It seems too good to be true and it probably is.

A problem is that the experimental design is suboptimal largely (but not solely) due to practical constraints. Different number of samples with different coverage are assessed with different methods from different bones. The authors claim that their analysis circumvents this, but the confusing and convoluted filtering and combination of analyses makes it impossible to judge how a bias might come about. That the end result is likely biased is evident from the fact that the 70% of the DMR are on the archaic lineages compared to modern lineage. While I see that many of these ad hoc and biased decisions are unavoidable, it should be noted at this point that a transparent and logic pipeline that allows to analyse the evolution of ancient DMRs is not a selling point of this paper. More importantly, any conclusion drawn from this set of DMRs is difficult to judge.

But even assuming that it would be an unbiased set of DMRs that is assigned to the modern and archaic human lineage. The following enrichment analysis is also very confusing and makes many non-transparent claims. A major principle of any enrichment analysis is that the expectation against with this enrichment is found needs to be clear (just like the controls in a biochemical experiment need to be clearly defined to judge its conclusiveness). What needs to be tested here is that the DMGs on the MH lineage are strongly enriched for vocal and facial anatomy compared to the archaic lineage (It's not enough to say that they are not enriched on the archaic lineage as absence of evidence is not evidence for absence). Furthermore, in order to make the bold claim, this would need to be tested against complete permutations (i.e. including all the convoluted preprocessing and analysis) and ideally it would need to be shown that this is specific, i.e. not found in any trio of primates. Unless one is willing to set the scientific standards for bold claims much lower for human evolution (which I am not).

I would rather prefer a much more solid and transparent paper than one where a "cool" finding was sought that in the end is likely to good to be true. So while the generated data and approach are clearly worthwhile one needs to restructure the logic and approach of the analysis completely to make it transparent and conclusive.

Reviewer #3:

Remarks to the Author:

This study builds on the authors' previous study by contributing additional datasets. Even though these new datasets are heterogeneous, the authors were able to identify several interesting patterns. However, I have some concerns regarding the validity of the null sets used in their statistical tests. I also felt that some claims were overstated without direct support. Finally, the paper lacked clarity in certain arguments.

Main statistical concerns:

p. 7, line 143: The authors performed a permutation test to conclude that MH-derived DMRs were significantly associated with MH-derived sequence changes and CTCF binding sites. This test appears to have been conducted against the entire genomic background, which is an inappropriate null. DMRs are generally GC-rich and CpG-enriched relative to the genomic background and contain a certain number of CpG sites by definition. A more appropriate background should be regions matched for sequence composition to the DMRs, not those randomly distributed given the unusual sequence composition of DMRs. The authors could also consider chromosomal distributions in addition to sequence composition, as chromosomes differ substantially in terms of sequence composition and thus can potentially bias the results. Similarly, the tests reported on lines 177, 191, and 247 (also on p. 81) should all be revised using appropriate null sets.

Another issue: is the reported trend regarding sequence changes and CTCF binding sites exclusively in MH-derived DMRs? The authors may perform the same analysis using Denisovan-derived, Neanderthal-derived, and undetermined DMRs to check if the MH-derived DMRs are significantly closer to MH-derived sequence changes or CTCF binding sites.

In addition, the GWAS enrichment test was performed only on genes associated with craniofacial features. Additional non-craniofacial feature related traits should be also compared as a control, to confirm that the observed enrichment is not due to DMRs being generally enriched for disease traits.

Overstatement or lacking support:

In the last sentence of the summary, the authors state that "... these genes became downregulated after the split from Neanderthals and Denisovans..". This claim is not substantiated by comparing gene expression. As the authors are aware, hyper-methylation does not necessarily lead to repression. I feel the authors can at least check the expression in chimpanzee tissues to substantiate their claim of downregulation.

Another statement that I felt was an overstatement appears on p. 9, line 197: "Finally, we ruled out the options that our DMR-detection algorithm, or biological factors such as gene length, cellular composition, pleiotropy or developmental stage might underlie the enrichment of these organs (see Methods)."

I am not convinced that the authors can rule out pleiotropic effects. A clear cautionary example is SOX9, which is well known to be extremely pleiotropic. The analyses performed (showing that voice and face-affecting genes are not more pleiotropic than other genes) do not imply those genes are not pleiotropic. Were the DMR genes they identified more or less pleiotropic than other voice- and face-affecting genes? This direct test is missing. The authors' claim on pleiotropy should, at a minimum, be toned down. If anything, the authors should include it as a potential limitation.

Similarly, regarding the gene length test on p. 77, do DMGs tend to be longer than other genes? This specific statistic is missing in the associated discussion.

Some discussion/results regarding hypo- versus hyper-methylation and different categories of gene annotation would add considerably to the paper. Most DMRs related to facial/vocal functions were reported to be hyper-methylated, but I could not tell whether this was the overall trend in MH DMRs, and how they varied in other branches. For example, hypo versus hyper proportions could be indicated in Figure 2A.

Minor comments

On p.6, line 122-124 and elsewhere, "we identified 9,679 regions that showed methylation differences between the high-quality representative methylomes of each human group"; it wasn't clear to me if this number is a sum of DMRs that differ in any two out of the three groups?

p. 9, line 181. 'top quartile'; of what exactly?

3) In Figure 3b (and Extended Data Figure 2g), the authors used the fraction of MH-derived CpG to present the plot for Archaic-derived DMRs. I feel it is more appropriate to consider the fraction of Archaic-derived CpGs in the plot for the Archaic-derived. Also, the simulated DMRs should have appropriate sequence composition as discussed above.

p. 46, line 763-764: The fact that several bone samples came from the same 6-year old female suggests that this individual was not well. Any other information on this matter? Was the disease unrelated to bones?

p.55, line 951: Chimpanzee- specific SNPs should be also considered, and it is not clear whether this was the case.

p. 59, line 997-p. 60 line 1069: This section indicates that new data was generated, but whether it is deposited in a public database and will become available is not reported.

p.85: I could not find GSE96833. The authors may check if this contains an error.

A paragraph in methods is duplicated (page 67 lines 1208-1218 and page 68-69 lines 1237-1248)

Why did they choose to test closeness between DMRs and CTCF binding sites (page 7, line 145)? I found it puzzling initially. Is it because they were interested in repressive marks, after realizing the potential significance of hyper-methylation? Some reasoning based on the data (such as the trend of hypo- versus hyper-methylation) needs to be offered here.

Reviewer #4:

Remarks to the Author:

Dear Editor, dear authors,

I have been invited to comment on the anatomy/phenotypic evolution of the vocal tract aspect of the study "Regulatory Divergence of Genes Affecting Vocal and Facial Anatomy in Modern Humans" and I would like to thank the editors for the opportunity to read a very interesting study with a novel approach to understanding differences in human and Neanderthal anatomy based on genetic information. In the following I am providing my feedback and comments on the parts of the study which concern themselves with the vocal tract and the face.

General considerations regarding the evolution of the anatomy of the vocal tract in this study
The summary of the evolution of vocal tract anatomy in modern humans (p. 15-16, lines 334-358) is adequate in reporting the various ongoing debates and considerations on a difficult anatomical region and reasons therefore (little bone/cartilage tissue left to fossilise). The summary could benefit from some more details on similarities and differences observed between the human and chimpanzee vocal tract anatomy since both species have been genetically studied and we can study their vocal tract anatomy. It might allow for understanding/linking some of the anatomical differences observed to the genetics reported here. I would recommend the work of Takeshi Nishimura (2005) in this matter, who investigated the similarities and differences in the postnatal vocal tract development of humans and chimpanzees (literature provided at the end of this text).

With regards what the study means for understanding the evolution of the human vocal tract, I find the core message of the study a bit confused and thus a bit difficult to understand. The text reads to me that the authors think that genetic variation drives face and vocal tract variations

towards a modern human configuration. However, due to lack of precise reporting on some topics (see below), it is not clear to me what the message of the authors is here - is the variation of the face driving changes of the vocal tract or is the changes to the vocal tract driving the changes to the face?

This is not a trivial question as it touches heavily on ongoing debates on how different "functional" regions of the head -i.e. neurocranium = brain, visceral cranium = feeding and breathing and other aspects such as head carriage (i.e. linked to locomotion) interact with each other. The face and the vocal tract are not isolated cranial regions but link tightly with the rest of the cranium.

I am fully aware that this is absolutely beyond the scope of the current study but in this case, the text should be clear that it is not the goal of the study to contribute to that discussion. If this was not the intended reading of the text, I do apologise but in that case, it might be useful to consider clarifying the text more. This particularly the case as interactions between face and vocal tract development is indicated. All in all, I think the authors study is much stronger on gene effects on facial development than it is on the voice altering effects. I would suggest that this is being considered.

An option would be to lead with the face development and to indicate that substantial alterations of the facial development, the vocal tract anatomy would be affected as well. Particularly, in the discussion, the authors already approach this (p. 16, lines 343-345) to some degree. I would recommend to discuss your findings in relation to Nishimura (2006) on the differences in face flattening and vocal tract development.

I understand that the authors are investigating regulatory genes, which in turn influence/cascade other genes and processes but in that case I would say that I am not comfortable with the statements on how genes influence the voice. There are two reasons for this. First, it is very difficult to state that genes have a direct influence on a feature like the voice. As recently indicated, modern speech sounds might have undergone substantial evolution since the palaeolithic based on environmental factors only and if that is indeed the case as Blasi et al. might be able to demonstrate, then it would be very difficult to link genes directly to voice. Furthermore, larynx anatomy likely is far more affected in relation to its primary functions (see below).

In some parts of the text (e.g. p.10, lines 207-212, p. 10, lines 213-220, p. 11, lines 236-238, p. 12, lines 249-254 and p. 15, lines 329-333) it is indicated that there is a direct affect of the genes on the voice. These text regions are to some degree also confusing as they argue aspects of larynx anatomy, larynx development, larynx mechanisms and voice quality as support of the genetic results. This per se is not as problem but all these topics are highly complex and not adequately introduced. Perhaps the recent publication by Gunz et al. (2019) might serve as a template on how to present a relatively good chain of evidence of the effects of regulatory genes of phenotype? My main concerns are thus with some lack of clear definitions of vocal tract anatomy and functions and particularly from linking these with the genetics. I would like to present some them to the authors in order to assist them in strengthening the clarity of their findings

1. Anatomy of the larynx
2. Function of the larynx
3. Definition of speech and voice
4. Development of the larynx vs. phenotypic presentation of the larynx

1. Anatomy of the larynx

First, there is need for some corrections of the anatomy of the vocal tract reported.

On page 10, lines 220-221, concerning the attachments of the vocal folds within the larynx: the vocal folds are posteriorly anchored at the vocal processes of the arytenoid cartilages. Anteriorly, the vocal folds attach to the thyroid cartilage (lamina), not the cricoid cartilage (any anatomy textbook can clarify that). The cricoid cartilage is the link element between the trachea and the

rest of the laryngeal cartilages and it has synovial joints with both the thyroid and the arytenoid cartilages.

On page 16, lines 338-339 - it should be different vocal tract configurations not conformations. Vocal tract configurations indicate shape variation of the vocal tract due to adjustments of vocal tract filters such as the position of the lips, teeth, soft palate, tongue shape and position and overall length of the vocal tract and degree of opening of the mouth. This in turn leads to variation in the vocal tract frequency output (e.g. vowels a, i, u or consonants).

The authors indicate that they consider the size? shape? position? (this is not clear from the text e.g. p. 10, lines 218-222, p. 17 lines 376-378) of the arytenoid cartilages essential for voice alteration. However, no literature is cited on what the arytenoids are actually doing in breathing, feeding and speech and how altered anatomy of these will alter the voice. I would think the authors will have quite a difficult time to find exactly what happens to the voice due to altered arytenoid anatomy. The literature is not clear what changes consistently even when they are paralysed other than that swallowing becomes difficult. I would invite the authors to find support for their arytenoid statements or alter the text.

2. Function of the larynx

Whilst humans take the mammalian use of the larynx for acoustic signalling to new levels of complexity, one always needs to keep in mind that like all other mammalian larynges (and even bird and reptile ones) it is first and foremost part of the respiratory system. I am aware that the authors are interested in the impact of genetic variation on the variation of speech production but since these are regulatory genes, I do not think it is justified to completely ignore the role these genes might play in those main functions as well.

The main function of the larynx is to regulate airflow to and from the lungs to guarantee adequate oxygen levels at all times and to safekeep the latter from invasion of foreign objects during feeding and swallowing processes. The way the larynx can safely close the airway is due to the adduction of the vocal folds and the vestibular folds and the lowering of the epiglottis. Any anatomical changes to the larynx will have to guarantee these main functions prior to any speech-production specific anatomical variations and this should be expressed in the text.

This is particularly the case for literature cited to support the role genes have in affecting the voice. I have read in detail all your citations supporting your statements regarding genes affecting the voice (i.e. 20, 21, 22, 23, 28, 33 and 43) and studied the extended data table 5. Nowhere in table 5 are voice alterations linked to your genes and in all these citations, there are 2 lines of text in totals referring to a hoarse voice and a high voice. This is not evidence that these genes affect the voice. However, what is striking to me is that all these papers (and table 5) describe severe and clear malformations of the larynx and respiratory system in relation to breathing function (e.g. narrowed larynx inlet, collapsed cartilage framework, narrowing of the trachea, tongue obstruction etc.). You might state that due to the severe breathing issues, the airflow for speech production is not given but I would think that this does not present a strong case for functional voice affection by a gene. I would focus on the primary functional aspects of these larynx malfunctions and leave the voice affecting statements out of the text. This leads to point

3 Definition of speech and voice

Speech-production specific anatomical variations of the larynx are difficult to understand and have not been studied much comparatively between humans and other living primates. The exception from this might be the presence or absence of vestibular air sacs and what that means for the production of single formant frequencies in speech. However, little is known about how the difference in e.g. thyroid or cricoid cartilage (if reported at all) influences the production of speech or the quality of voice. The role of the larynx in speech is not clearly defined in the text but this is crucial for the way the study reports on what the larynx does or does not in speech and what this

means for the quality of voice, which is again something different.

In speech-production, the larynx regulates the airflow from the lungs in speech and makes sure that the vibrating vocal folds create steady formant frequencies which are recognised as acoustic signals. The airflow is modified in the larynx so that a succession of varying speech elements can be produced. This is achieved via the speed and duration of the vibration of the vocal folds and the size and shape of the opening of the glottis (rima glottidis). The position of the larynx and the tension of the vocal folds also play a role in colouring/altering the quality of the voice, such as volume, pitch etc.

It is to me not clear in the text if the authors consider quality of voice, which varies substantially between individual humans and also at different stages of life within a single individual is evolutionary important or if they think larynx anatomy variation is important for speech production per se? I would have thought the latter but I would be much more precise on that issue throughout the text. If the latter is the case, I would be precise on which differences in larynx anatomy you consider evolutionary relevant. In context of this, I also found the reports on the developing larynx difficult to track through the text.

4. Development of the larynx vs. phenotypic presentation of the larynx

The laryngeal cartilages without any doubt are mesoderm derived and develop from the branchial arches 4-6. The tracheal cartilaginous rings are also mesodermally derived and I would like some clarification if the authors have accounted for the expression of their investigated genes in these structures as well.

Importantly, the larynx is located at the junction with the endodermally derived lungs and trachea. Tissues originating from the developing trachea immigrate into the larynx to form parts of the vocal folds, including the vocalis muscle (visceral muscle) and all soft tissues of the larynx. There are several inferences made to vocal folds in the text and in relation to the genes investigated but there is no specification of the varying developmental tissue origins between the larynx cartilages and the vocal folds. I would be grateful if there is clarification if these genes have an expression in the development of the vocal folds as well. Furthermore, the formation of the larynx is complex with a temporary closure of the glottic opening during the prenatal development (around week 10). I would therefore be very careful in making statements about the influence chondrossification regulating genes have on the vocal folds.

In summary, it should be possible to improve some of the definitions of anatomy vocal tract and its functions and be more defined on the affects the genes have on the indicated structures. I would recommend considering discussing the findings in relation to what is already known about facial and vocal tract development but do not think these changes are difficult or extensive to undertake.

Literature cited:

Blasi, D.E. et al. (2019). Human sound systems are shaped by post-Neolithic changes in bite configuration. *Science* 363 (15th March)

Gunz, P. et al. (2019). Neanderthal Introgression sheds light on modern human endocranial globularity. *Current Biology* 29: 120-127

Nishimura, T (2005). Developmental changes in the shape of the supralaryngeal vocal tract in chimpanzees. *American Journal of Physical Anthropology* 126(2): 193-204

Nishimura, T., Mikami, A., Suzuki, J. Matsuzawa, T. (2006) Descent of the hyoid in chimpanzees: evolution of face flattening and speech. *Journal of Human Evolution* 51(3):244-254

Reviewer #1 (Remarks to the Author):

Gokhman et al. generated and/or used methylation maps from modern and ancient AMHs, archaic humans (Neanderthals and Denisovans), as well as chimpanzees to detect DMRs that arose after the split between humans and Neanderthals/Denisovans. Methylation patterns are associated with repression of genes involved in cranial and facial morphology in the human lineage, suggesting a role for methylation in shaping the human face and potentially the vocal tract.

The question of whether archaic humans such as Neanderthals and Denisovans and AMHs share the same vocal anatomy is important to test the hypothesis that speech is a unique AMH feature. Morphological studies are limited by the fact that the larynx is mostly composed of cartilage and consequently is not preserved in skeletal remains of archaic humans. Instead, Gokhman et al. propose to use a molecular approach to infer the shaping of craniofacial anatomy and vocal tract in AMH and archaic human lineages. Their approach relies heavily on the method they have developed that leverages patterns of ancient DNA damage to reconstruct methylation maps.

As I was reading the manuscript, I could tick off many of my a priori concerns about tissue- and sex-specificity of DNA methylation, impact of DNA damage on methylated cytosines, DMR detection using different methylome reconstruction methods, correlations between DNA methylation and gene expression, and validation using a cell model and/or existing syndromes. Really, either the authors have been extremely careful with the validation of their empirical and analytical results, or the manuscript has already gone through at least one round of external reviews and the authors have addressed all issues in earnest. As a result, there is not much to critique from a data generation and analysis point of view.

We appreciate the reviewer's comments, kind words and careful examination of this work. Indeed, these aspects are central in such analyses and we tried to be as methodical and thorough as possible in addressing them. This work includes over 50 validation tests and it was critical for us to support each of its claims by multiple orthogonal evidence. We are glad to see that the reviewer sees the extensive work that was put into this project in the past 5 years.

I had a thorough look at the statistical analyses detailed in the methods section because the detection of DMRs from deamination maps and empirical methylomes is likely to be the most problematic analysis. The authors have adopted a rather conservative approach, even though they rely heavily on the use of a "methylome reference" that may itself be slightly biased. Nevertheless, the DMR detection algorithm seems to perform reliably since the comparison between deamination maps and reconstructed methylomes from the same maps lead to a low number of false positives. In addition, the two- and three-way DMR detection strategies are additional precautions that account for the different technologies/methodologies as well as the incompleteness of methylation maps at the genomic level.

We thank the reviewer for their comment. Indeed, the DMR-detection method used in this study is particularly conservative and incorporates several different measures to make sure that the DMRs found are as robust as possible.

In my humble opinion, this study was carefully planned (at least according to how it is presented in the manuscript) and the interpretation of the results is cautious. The NIFX results summarized in Figure 6 are definitely the most striking results because they nicely tie together molecular evolution and pathogenetic mechanisms behind some AMH craniofacial syndromes.

We thank the reviewer for the kind words.

My only minor comments are:

– I suggest to use Anatomically Modern Human (AMH) instead of just Modern Human (MH) to comply with the description used by the large majority of anthropologists.

Following the reviewer's comment, we changed MH to AMH throughout the manuscript.

– Line 175: The authors report “44 of the MH-derived DMRs” out of 873 MH-derived DMRs. It does not seem like much but the permutation test is significant and shows 60.4x enrichment. However, how many DMGs does it represent, i.e., 44 or less?

28 of these DMRs are found within genes, the rest are intergenic. The enrichment test was based on DMRs rather than DMGs because of the relatively low reliability of algorithms linking enhancers to specific genes. The enrichment is due to the relatively small number of putative enhancers related to human craniofacial development.

– Lines 737–738: “The first few bases from each read have been shown to have a slightly higher probability of being called as methylated¹⁸, so we trimmed the first ten bases from each read”: This sentence is misleading because reference 18 (Schultz et al. 2015 Nature) does not address the issue of methylation bias at the ends of sequencing reads as far as I know. However, Hansen et al 2012 Genome Biology shows M-bias plots where methylation bias can occur at the end and/or at the start of reads depending on the bisulfite method used. I suggest to rewrite the sentence to be clearer.

The reviewer is correct. We have thus changed this paragraph to: “Although methylation state should not depend on read position, positional biases have been previously reported [Hansen et al 2012 Genome Biology]. We observed that the first few bases from each read showed a slightly higher probability of being called as methylated, so we trimmed the first ten bases from each read (M-bias filtering).”

Reviewer #2 (Remarks to the Author):

In a previous paper the authors developed a method to infer CpG methylation based on the DNA damage pattern from ancient DNA. As the methylation profile stems from ancient bones, it is crucial to compare a sufficient number of ancient methylation profiles with modern profiles of humans and outgroups (Chimpanzee) to interpret the ancient profiles. This is what they do in this paper, by inferring methylation profiles from additional published ancient genomes, ancient genomes sequenced for this paper and chimpanzee methylation profiles. Their main conclusion from the analysis of this data, as prominently laid out in the title is “regulatory divergence of genes affecting vocal and facial anatomy in modern humans”. While the generated data and method is clearly of interest I have serious doubts about this bold conclusion. It seems too good to be true and it probably is.

Even though the reviewer probably did not mean it this way, we agree that the results are important and illuminating. Due to their importance, we were very cautious in reporting them and have tested them using 54 orthogonal tests, controls and validations, which is far beyond the standard for comparative genomics studies.

Any specific concerns regarding the validity of the results are addressed below.

A problem is that the experimental design is suboptimal largely (but not solely) due to practical constraints. Different number of samples with different coverage are assessed with different

methods from different bones. The authors claim that their analysis circumvents this, but the confusing and convoluted filtering and combination of analyses makes it impossible to judge how a bias might come about. That the end result is likely biased is evident from the fact that the 70% of the DMR are on the archaic lineages compared to modern lineage. While I see that many of these ad hoc and biased decisions are unavoidable, it should be noted at this point that a transparent and logic pipeline that allows to analyse the evolution of ancient DMRs is not a selling point of this paper. More importantly, any conclusion drawn from this set of DMRs is difficult to judge.

The reviewer raises two main concerns: the clarity of the DMR-detection pipeline, and potential biases.

With regard to clarity, the original version of the manuscript contained 14 pages of supplementary information describing the algorithm in detail. Following this comment we realized that while this might provide an in-depth description of the process, it might also be difficult for readers to grasp the general picture. We have therefore added an introductory section to the DMR-detection chapter, providing an overview of the entire process. In short, the pipeline is based on three common steps in methylation analyses: DMR detection, lineage assignment and filtering out of within-lineage variability. We believe that the new summary section, together with Extended Data Fig. 1 (which also summarizes the pipeline) should provide the reader with a clear and transparent overview of the DMR detection process.

The reviewer also raises four potential sources of bias: number of samples, coverage, the different methods and the different types of bones. These are sensible concerns, and these factors could potentially affect some analyses. However, they were either addressed in the study or are irrelevant to it:

Coverage: there are three samples which were used for the initial DMR-detection algorithm: the Altai Neanderthal (52x), Ust'-Ishim (43x) and Denisovan (30x). As the coverage of the Ust'-Ishim sample is intermediate, it is unlikely to cause it to be an outlier (i.e., systematically classified as MH- or archaic-derived DMRs). More importantly, our DMR-detection method directly accounts for the sample coverage and incorporates it into the detection statistics. Finally, the subsequent DMR-filtering step is using methylation values rather than read counts and is therefore also unlikely to be affected by coverage. In summary, the variation in coverage is not a factor that is expected to create biases in our study. This issue is also discussed in the chapter "Removing DMRs with high within-group variability".

Number of samples: all DMRs in this study are detected by first comparing only the Altai Neanderthal, Denisovan and Ust'-Ishim maps. Additional samples are used solely to discard DMRs that are variable within a lineage. As 59 out of the 67 maps used in the study belong to MHs, our power to detect and discard such DMRs is higher compared to the other lineages. This results in relatively fewer DMRs being left in this group (and more DMRs along the archaic lineage compared to the MH lineage, as mentioned by the reviewer). See "Removing DMRs with high within-group variability" chapter for further information. Given the improved ability to detect fixed changes along the MH lineage, MH-derived DMRs are the focus of our study. Therefore, we do not directly compare between the lineages, but rather focus on describing the findings that relate to the MH lineage. Nevertheless, we would like to emphasize that the differential power to find variability does not seem to result in any biases in the enrichment tests we report: when we analyze DMRs without within-lineage filtering, the trends we reported are clearly repeated, suggesting they are not driven by the differences in the number of samples; in MH-derived DMRs, the laryngeal and facial regions are the most significantly enriched regions (1.58x, 1.44x and 1.21x-1.31x for the vocal cords, larynx and different facial regions, respectively, FDR < 0.05). For archaic-derived DMRs, on the other hand, the laryngeal region is

not significantly enriched (FDR = 0.16 and FDR = 0.43 for the vocal cords and larynx, respectively), whereas the most enriched regions are the face, limbs, and urethra, which (with the exception of the urethra) were reported as most enriched in the analysis of non-variable archaic-derived DMRs too (Extended Data Table 4). These results suggest that the enrichment patterns detected are not affected by the variable number of samples in each lineage. These results now appear in the “Validation of face and larynx enrichment in Gene ORGANizer” chapter.

Different methods: the use of different methods actually makes the analysis more conservative, as we require that the methylation values of one human group be completely outside the range of methylation in the other group, regardless of the method used for measuring methylation (Fig. 1). In other words, if one of the methods causes methylation levels to shift in any direction, this would increase the range of methylation observed for a group. By requiring that regardless of method, all samples in that group be completely outside the other group, we account for this potential bias and are more conservative. Importantly, the paper focuses on MH-derived DMRs, where each of the two groups (a. MHs and b. archaic humans and chimps) include all three methods (WGBS, reconstruction and arrays). Finally, we show that for our top 5 candidate genes, hypermethylation is clearly observed even when the same method is compared between archaic and modern human samples (Vindija reconstructed map vs. Ust’-Ishim reconstructed map, Extended Data Fig. 3c. Both are also from the same bone (femur)). A similar pattern is observed when comparing chimp and human methylation arrays (Extended Data Fig. 3d), suggesting that the observed hypermethylation is not driven by method biases.

Bone type: differential sources of tissues are a main concern in many comparative analyses. We have taken several different measures to minimize or eliminate altogether any potential biases due to this factor: (1) First, the methylation levels we report are skeleton-wide and are very unlikely to be specific to one bone-type. This is achieved by the conservative way in which we define a DMR. In our analyses, a locus is considered a DMR if all samples in one group cluster completely outside all samples from the other groups. MH samples include the following bones: femora, crania, rib, tibia and teeth, and the other groups (chimp and archaic humans) include the following bones: femora, crania, rib and phalanges. The fact that in MH-derived DMRs, all MH samples are found completely outside the archaic and chimp samples, regardless of bone type, suggests that the patterns observed are not bone-specific, but rather extend to all the various bones and teeth in the study and probably the entire skeleton. For example, if a pattern we observe in a locus is specific to teeth, rather than the MH lineage, we would not expect the MH femora to cluster with the MH teeth. Instead, if the region is specific to teeth, we would expect MH femora to cluster with chimp femora. The exception to this is Denisovan-derived DMRs, where some of the patterns could be attributed to finger-specific methylation, rather than lineage-specific methylation. However, this is irrelevant to our study as we do not analyze candidate Denisovan-derived DMRs. (2) For our top five most differentially methylated genes, we further validate their hypermethylation in MHs by comparing the same bone type between the groups (MH femur vs. Neanderthal femur, and MH femur vs. chimp femur). In all cases, the five genes show significant hypermethylation in the MH samples, despite being from the same bone. (3) For our top candidate gene, NFIX, we further validate its derived hypermethylation in MHs by comparing its methylation levels in MH vs chimp crania. Here too, NFIX shows significant hypermethylation in MHs compared to chimps, despite all samples being cranial. (4) The vast majority of samples within each group are limb bones, further narrowing the distance between their anatomical origin. (5) We removed regions which have previously been shown to change their methylation levels during osteogenic differentiation. This suggests that the regions we report, where methylation levels remain consistent during osteogenic differentiation, establish their patterns at early stages (before the differentiation to different bone types is complete, probably much earlier, and are thus even more likely to represent skeleton-

wide patterns). An in-depth description of these factors and considerations can be found in the “Removing DMRs with high within-group variability” chapter.

Overall, despite being very sensible concerns, we believe that our conservative approach in detecting DMRs has minimized or eliminated altogether the contribution of these factors, and that the validation tests on matched and unfiltered DMRs show that they are unlikely to bias this study.

But even assuming that it would be an unbiased set of DMRs that is assigned to the modern and archaic human lineage. The following enrichment analysis is also very confusing and makes many non-transparent claims. A major principle of any enrichment analysis is that the expectation against which this enrichment is found needs to be clear (just like the controls in a biochemical experiment need to be clearly defined to judge its conclusiveness). What needs to be tested here is that the DMGs on the MH lineage are strongly enriched for vocal and facial anatomy compared to the archaic lineage (It’s not enough to say that they are not enriched on the archaic lineage as absence of evidence is not evidence for absence).

We thank the reviewer for his/her suggestion. We agree that a direct comparison to the other lineages is an interesting and important analysis and we have thus conducted this test based on the reviewer’s outline. Our results re-confirm our previous findings and show that the extent of regulatory changes in genes affecting the vocal tract in MHs is significantly higher than that along the other lineages. These results have been added to the paper under the chapter:

“Validation of face and larynx enrichment in Gene ORGANizer”. Before delving into the results themselves, we would like to emphasize that the analysis the reviewer suggested asks a slightly different question than the one we sought to study in our paper. The question we have investigated was “what are the organs most affected by the methylation changes along each of the lineages?”. The control or background for such a question is not the other lineages, but rather the other organs and the genetic distribution of gene-organ links. Thus, in each lineage, and especially along the MH lineage, we sought to rank the organs from most to least affected and compare these results to the genetic background based on the number of genes expected to affect each organ (the default background in Gene ORGANizer). Importantly, in addition to general trend analyses, unique changes to specific genes should be investigated regardless to whether other genes in other lineages affect similar organs. Therefore, we believe the methylation changes we report, particularly in the top 5 candidates, are interesting in themselves, regardless of the trends along the other lineages.

The reviewer offers to investigate another aspect of the enrichment we report in the paper, i.e., test whether the changes in genes affecting the vocal tract and face along the MH lineage are not only enriched compared to expected, but also compared to what is observed in the other lineages. This is a slightly different question, as an organ could go through rapid evolution on two lineages, and still be the most divergent organ in each. In fact, the facial region, which we reported to be significantly enriched along both the archaic and modern lineages, has been shown to have gone through particularly extensive morphological changes along both these lineages [Aiello, L. & Dean, C. *An Introduction to Human Evolutionary Anatomy*. (Elsevier, 2002)]. Therefore, the former analysis would identify such changes, even if they are shared with other lineages, whereas the latter, where the two lineages are compared against one another, would probably not. This is a particularly important distinction, as almost every bone in the modern human body has been shown to differ in one or more aspects from its archaic counterparts [Aiello, L. & Dean, C. *An Introduction to Human Evolutionary Anatomy*. (Elsevier, 2002)]. Because regulatory changes are expected to be the driving force behind most morphological changes in closely related groups, it would not be surprising to find regulatory changes affecting almost every bone along these lineages. Nevertheless, we believe both

questions are very interesting, and none has been addressed in human evolution genetic studies so far. We therefore thank the reviewer for suggesting another important analysis to add to the paper.

We therefore sought to test whether the larynx and vocal cords, which we found to be significantly enriched only along the MH lineage, are enriched when compared to the other lineages. We ran a chi-squared test on the fraction of vocal cords- and larynx-affecting MH-derived DMGs, compared to the corresponding fraction in the DMGs along all the other lineages. We found that both the larynx and vocal cords are significantly enriched in MHs by over 50% (1.57x for both, $P = 0.0248$ and $P = 0.0169$ for vocal cords and larynx, respectively).

Furthermore, in order to make the bold claim, this would need to be tested against complete permutations (i.e. including all the convoluted preprocessing and analysis) and ideally it would need to be shown that this is specific, i.e. not found in any trio of primates.

Unless one is willing to set the scientific standards for bold claims much lower for human evolution (which I am not).

I would rather prefer a much more solid and transparent paper than one where a “cool” finding was sought that in the end is likely to good to be true. So while the generated data and approach are clearly worthwhile one needs to restructure the logic and approach of the analysis completely to make it transparent and conclusive.

As a control, the reviewer suggests to test whether permutations where all preprocessing steps and analyses are conducted, show any enrichment patterns which could suggest that the trends we see are an artifact of the analysis, rather than true biologically meaningful results. This analysis has been done in the original version of the paper and is described in the “Filtering out noise” and “Validation of face and larynx enrichment in Gene ORGANizer” chapters.

The reviewer also suggests testing whether these results are observed when comparing a trio of primates. Unfortunately, our study includes all the primate bone methylation maps published to date that we are aware of. Nevertheless, we have shown that such enrichments are observed only in one out of the five lineages analyzed in this study (MHs, archaic, Neanderthal, Denisovan and chimps), already suggesting that this is unlikely to be the result of a systematic bias. Finally, regulatory changes to an organ can occur on more than one lineage, and therefore, it is unclear to us how such a test would support/refute our results. It is, nevertheless, interesting in itself as an evolutionary question to study which organs went through extensive changes in various primate lineages, but we believe that this is outside the scope of this study.

Overall, the current study includes 54 controls and validation tests, with 55 pages of supplementary information describing every step of the process in detail and transparency. This detailed description might have caused the overall logic of some steps to be vague at times, and we have thus added summary paragraphs where deemed necessary. We have also conducted additional analyses which the reviewer suggested to further test the results we report. We believe the current version of the manuscript addresses potential concerns thoroughly, or in Reviewer 1’s words: “In my humble opinion, this study was carefully planned... and the interpretation of the results is cautious.”

Reviewer #3 (Remarks to the Author):

This study builds on the authors’ previous study by contributing additional datasets. Even though these new datasets are heterogeneous, the authors were able to identify several interesting patterns. However, I have some concerns regarding the validity of the null sets used

in their statistical tests. I also felt that some claims were overstated without direct support. Finally, the paper lacked clarity in certain arguments.

We thank the reviewer for his/her comments. We followed the reviewer's suggestions regarding the null sets, toned down the relevant arguments and rephrased any unclear arguments.

Main statistical concerns:

p. 7, line 143: The authors performed a permutation test to conclude that MH-derived DMRs were significantly associated with MH-derived sequence changes and CTCF binding sites. This test appears to have been conducted against the entire genomic background, which is an inappropriate null. DMRs are generally GC-rich and CpG-enriched relative to the genomic background and contain a certain number of CpG sites by definition. A more appropriate background should be regions matched for sequence composition to the DMRs, not those randomly distributed given the unusual sequence composition of DMRs. The authors could also consider chromosomal distributions in addition to sequence composition, as chromosomes differ substantially in terms of sequence composition and thus can potentially bias the results. Similarly, the tests reported on lines 177, 191, and 247 (also on p. 81) should all be revised using appropriate null sets.

We agree with the reviewer that the appropriate null background for this test should be sequence matched and that chromosomes should be matched as well. Indeed, the original analysis used chromosome matching of the original and permuted locus, but this was not mentioned in the text. As the reviewer mentions, GC-content and CpG density matching were not done originally and we agree with the reviewer that this is a more accurate randomization. Therefore, we re-ran the permutation tests while matching the length, chromosome, GC distribution and CpG density of regions in the original and permuted lists. GC-content and CpG density matching was done by matching a 10-bin histogram of the original and permuted lists.

We found that the lists overlap significantly more often than expected by chance. For the overlap of AMH-derived DMRs with putative craniofacial enhancers we found that the two lists overlap 5.1x more often than expected by chance ($P < 10^{-4}$). For the overlap of AMH-derived DMRs with candidate selective sweep regions we found that voice- and face-affecting genes are found 15% closer on average to candidate selective sweep regions ($P < 10^{-4}$), and 40% closer to these regions than other DMRs. As the effect size of the former result is somewhat restricted, we now only mention that voice- and face-affecting genes are found 40% closer to candidate selective sweep regions than other DMGs ($P < 10^{-4}$, line 253).

With regard to the permutation test in line 191 (currently line 195), the permutations are based on randomization of gene IDs rather than genomic localization, a setup that likely controls for the factors mentioned above.

For the overlap of AMH-derived DMRs with putative limb enhancers (currently p.83), we found that AMH-derived DMRs tend to overlap putative limb enhancers ~2x more often than expected by chance ($P < 10^{-4}$).

With regard to the CTCF results, following Reviewer 3's comment, suggesting this analysis is confusing to the reader and is unnecessary, we decided to remove this analysis from the current version of the manuscript.

Another issue: is the reported trend regarding sequence changes and CTCF binding sites exclusively in MH-derived DMRs? The authors may perform the same analysis using Denisovan-derived, Neanderthal-derived, and undetermined DMRs to check if the MH-derived DMRs are significantly closer to MH-derived sequence changes or CTCF binding sites.

Please see previous comment.

In addition, the GWAS enrichment test was performed only on genes associated with craniofacial features. Additional non-craniofacial feature related traits should be also compared as a control, to confirm that the observed enrichment is not due to DMRs being generally enriched for disease traits.

We thank the reviewer for his/her practical and helpful suggestion. We agree that this is a useful control. Following the reviewer's comment, we added a test where genes associated with non-craniofacial features were used as a control for the craniofacial enrichment test. We used GWAS catalog blood-associated genes as our control group and found that DMGs significantly overlap craniofacial-associated genes compared to blood-associated genes (3.86-fold, $P = 0.011$, chi-square test, see "Validation of face and larynx enrichment in Gene ORGANizer" chapter for further information).

Overstatement or lacking support:

In the last sentence of the summary, the authors state that "... these genes became downregulated after the split from Neanderthals and Denisovans..". This claim is not substantiated by comparing gene expression. As the authors are aware, hyper-methylation does not necessarily lead to repression. I feel the authors can at least check the expression in chimpanzee tissues to substantiate their claim of downregulation.

We thank the reviewer. Following the reviewer's comment, we rephrased this sentence accordingly to better reflect our findings. Given that hypermethylation was previously shown to be a repression mark of this gene network in several studies, we rephrased the summary to "we identify widespread hypermethylation..." and "We propose that these repression patterns appeared after the split from Neanderthals and Denisovans"

We also agree with the reviewer that direct evidence from chimpanzee skeletal tissues would be helpful. This would probably reduce the need to use phrasings like "we suggest/propose" with regard to gene expression and would be the "final nail in the coffin". Unfortunately, we have been trying to obtain fresh chimpanzee bones for more than 6 years without success. Many of the authors on this paper have direct access to chimpanzee samples, including Tomas Marques-Bonet, who supervises all EU zoos, but none has been able to obtain a chimpanzee bone that is fresh enough to extract high-quality RNA from. In fact, to date, no such datasets have been published. We are continuing our efforts to obtain such samples, but unfortunately, we are unlikely to obtain them for this manuscript.

Another statement that I felt was an overstatement appears on p. 9, line 197: "Finally, we ruled out the options that our DMR-detection algorithm, or biological factors such as gene length, cellular composition, pleiotropy or developmental stage might underlie the enrichment of these organs (see Methods)."

I am not convinced that the authors can rule out pleiotropic effects. A clear cautionary example is SOX9, which is well known to be extremely pleiotropic. The analyses performed (showing that voice and face-affecting genes are not more pleiotropic than other genes) do not imply those genes are not pleiotropic. Were the DMR genes they identified more or less pleiotropic than other voice- and face-affecting genes? This direct test is missing. The authors' claim on pleiotropy should, at a minimum, be toned down. If anything, the authors should include it as a potential limitation.

We thank the reviewer for his/her comment. Indeed, SOX9 is a pleiotropic gene and it was cardinal for us to examine whether pleiotropy biases any of our results. The reviewer suggests a direct test to examine whether larynx- and face-affecting DMGs are more/less pleiotropic than

other DMGs. In this test, which appears in the supplementary information under the chapter “Validation of face and larynx enrichment in Gene ORGANizer”, we examined the number of phenotypes associated with each gene and saw no evidence suggesting that larynx- or face-affecting DMGs are more pleiotropic than other DMGs (t-test, $P = 0.19$ and $P = 0.27$, respectively). Importantly, given the nature of Gene ORGANizer, even if a subset of genes tends to be pleiotropic, it should not result in over-representation of specific organs such as the vocal cords and the face, unless they are indeed affected to a larger extent. This last point is explained in detail in the chapter “Validation of face and larynx enrichment in Gene ORGANizer”.

Similarly, regarding the gene length test on p. 77, do DMGs tend to be longer than other genes? This specific statistic is missing in the associated discussion.

Indeed, as the reviewer mentions, we tested if larynx- and face-affecting DMGs tend to be longer than other DMGs, but not if DMGs tend to be longer than other genes. As expected, DMGs do tend to be longer than other genes, as longer genes are more likely to overlap a DMR (148 kb vs. 39 kb, $P = 9.9 \times 10^{-145}$, t-test). Importantly, this trend is unlikely to drive the enrichment trends we report, as the trends hold when controlling for gene length. Also, MH-derived DMGs do not tend to be longer than DMGs on the other branches (148 kb vs. 147 kb, $P = 0.93$, t-test). We now present these points in the “Validation of face and larynx enrichment in Gene ORGANizer” chapter.

Some discussion/results regarding hypo- versus hyper-methylation and different categories of gene annotation would add considerably to the paper. Most DMRs related to facial/vocal functions were reported to be hyper-methylated, but I could not tell whether this was the overall trend in MH DMRs, and how they varied in other branches. For example, hypo versus hyper proportions could be indicated in Figure 2A.

We thank the reviewer for his/her comment. Indeed, most DMRs were hypermethylated (see Extended Data Table 2 for a complete description of methylation levels across all DMRs). However, we were cautious not to infer conclusions from this pattern, as our ℓ_i^+ and ℓ_i^- statistics are affected by the number of cytosine reads observed, and thus these statistics have more power to detect hyper vs. hypomethylation. Importantly, this does not affect the reconstructed methylation levels (which were also confirmed independently in Hanghøj et al, Fast, Accurate and Automatic Ancient Nucleosome and Methylation Maps with epiPALEOMIX, MBE, 2016). Rather, it reduces our ability to detect hypomethylation. We realize this point was missing from the paper, and we now mention this in the “Detecting DMRs” chapter.

Minor comments

On p.6, line 122-124 and elsewhere, “we identified 9,679 regions that showed methylation differences between the high-quality representative methylomes of each human group”; it wasn’t clear to me if this number is a sum of DMRs that differ in any two out of the three groups?

Indeed, this sentence was not clear enough. This is indeed the sum of DMRs that differ between any of the two human groups. We rephrased it to: “Using this method, we identified 9,679 regions overall that showed methylation differences between any of the high-quality representative methylomes of the Denisovan, the Altai Neanderthal, and the Ust’-Ishim anatomically modern human”

p. 9, line 181. ‘top quartile’; of what exactly?

We added “ Q statistic”, the statistic we use to rank DMRs, to emphasize that the top quartile refers to the most significant DMRs. The sentence now reads: “To test whether this enrichment remains if we take only the most confident DMRs, we limited the analysis to DMGs where the most significant DMRs are found (top quartile, Q statistic).”

3) In Figure 3b (and Extended Data Figure 2g), the authors used the fraction of MH-derived CpG to present the plot for Archaic-derived DMRs. I feel it is more appropriate to consider the fraction of Archaic-derived CpGs in the plot for the Archaic-derived. Also, the simulated DMRs should have appropriate sequence composition as discussed above.

We completely agree with the reviewer. The data presented is actually for archaic-derived CpGs, but we now understand that the legend was incorrect. To better emphasize that the derived CpGs were computed for the lineage on which the DMRs arose, we changed the legend for Figure 3b to “Within each lineage, the fraction of differentially methylated CpGs was computed as the number of derived CpGs per 100 kb centered around the middle of each DMR”. As Extended Data Figure 2g presents only MH-derived DMRs and MH-derived CpG density, we left its legend unchanged.

p. 46, line 763-764: The fact that several bone samples came from the same 6-year old female suggests that this individual was not well. Any other information on this matter? Was the disease unrelated to bones?

We contacted Lonza (who supplied these cells to the ENCODE project, from which we downloaded the methylation data). Unfortunately, they did not have additional information about the state of this individual, but based on the description of these cells, there is no mentioning of a disease that potentially affects bones and these osteoblasts are described as “normal” (NH_{ost}-Osteoblasts by Lonza Pharma, product code: CC-2538, lot number: 6F4124). We have added the cell ID information to the methods. Please note that as the methylation map from this individual was produced using RRBS technology, it is biased towards unmethylated positions, and therefore we were very cautious in comparing it to other maps in our study and used it only once for an additional Gene ORGANizer validation to test whether some of the patterns we see might be driven by differences in age. See further details in the “Validation of face and larynx enrichment in Gene ORGANizer” chapter.

p.55, line 951: Chimpanzee- specific SNPs should be also considered, and it is not clear whether this was the case.

Indeed, chimpanzee-specific SNPs were also considered and removed from the list. We now clearly state in the text that both human and chimp-specific SNPs were removed.

p. 59, line 997-p. 60 line 1069: This section indicates that new data was generated, but whether it is deposited in a public database and will become available is not reported.

We thank the reviewer for noticing this. Indeed, we forgot to add a reference for this data, which was uploaded to GEO together with the rest of the data we generated. We now refer to the GEO accession in which this data was deposited (GSE96833), as well as its SRA accession (SRX3194436). We also noticed that these data were missing for the second ancient genome we sequenced, and we therefore added this information to the I1583 section too. See next comment for reviewer access to the data.

p.85: I could not find GSE96833. The authors may check if this contains an error.

The reviewer is right. This record exists, but it is still private (it will be publicly released when the paper is accepted). To allow access for the reviewer, we created the following token: sfqhggyigbfbvwp, which can be entered in the "If you are a reviewer, enter secure token here:" box and allow access to all maps and raw sequencing data.

A paragraph in methods is duplicated (page 67 lines 1208-1218 and page 68-69 lines 1237-1248)

We thank the reviewer for this comment. We removed the second paragraph from the text.

Why did they choose to test closeness between DMRs and CTCF binding sites (page 7, line 145)? I found it puzzling initially. Is it because they were interested in repressive marks, after realizing the potential significance of hyper-methylation? Some reasoning based on the data (such as the trend of hypo- versus hyper-methylation) needs to be offered here.

We thank the reviewer. Following Reviewer 2's comment concerning the same analysis, as well as the fact that there are many potential factors that could affect these results, and because this analysis is very peripheral to the main text, we decided to remove it from the current version of the manuscript.

Reviewer #4 (Remarks to the Author):

Dear Editor, dear authors,

I have been invited to comment on the anatomy/phenotypic evolution of the vocal tract aspect of the study "Regulatory Divergence of Genes Affecting Vocal and Facial Anatomy in Modern Humans" and I would like to thank the editors for the opportunity to read a very interesting study with a novel approach to understanding differences in human and Neanderthal anatomy based on genetic information. In the following I am providing my feedback and comments on the parts of the study which concern themselves with the vocal tract and the face.

We thank the reviewer for his/her in depth review, we did our best to incorporate all of these comments into the current version of the manuscript.

General considerations regarding the evolution of the anatomy of the vocal tract in this study The summary of the evolution of vocal tract anatomy in modern humans (p. 15-16, lines 334-358) is adequate in reporting the various ongoing debates and considerations on a difficult anatomical region and reasons therefore (little bone/cartilage tissue left to fossilise).

We thank the reviewer. We have tried to present the different aspects of this debate to inform the reader as accurately as possible about the different opinions regarding vocal tract evolution.

The summary could benefit from some more details on similarities and differences observed between the human and chimpanzee vocal tract anatomy since both species have been genetically studied and we can study their vocal tract anatomy. It might allow for understanding/linking some of the anatomical differences observed to the genetics reported here. I would recommend the work of Takeshi Nishimura (2005) in this matter, who investigated the similarities and differences in the postnatal vocal tract development of humans and chimpanzees (literature provided at the end of this text).

We thank the reviewer for their comment. The Nishimura study is a great addition to the manuscript and its description of vocal tract development is illuminating and clear. Based on this study, we have added the following sentence to the discussion: "Modern humans have a 1:1 proportion between the horizontal and vertical dimensions of the vocal tract, which develops mainly in post-infant years and is unique among primates" with reference to the Nishimura study. This fits with our genetic data showing regulatory differences that extend to adulthood. We have also added the following sentence: "Moreover, the main differences between human and chimp vocal tracts are established during post-infant years" to the discussion about the function of our top five genes in late developmental stages. We thank the reviewer for this reference and the idea to present a clearer link between the function of the genes we report and phenotypic observations in development.

With regards what the study means for understanding the evolution of the human vocal tract, I find the core message of the study a bit confused and thus a bit difficult to understand. The text reads to me that the authors think that genetic variation drives face and vocal tract variations towards a modern human configuration. However, due to lack of precise reporting on some topics (see below), it is not clear to me what the message of the authors is here - is the variation of the face driving changes of the vocal tract or is the changes to the vocal tract driving the changes to the face?

We thank the reviewer for pointing this unclarity. We believe this is unclear because we do not take a stance regarding the interaction between the two as we do not think the current data provides enough information to determine which process is driven by which. However, to clarify this, we now state in the discussion that "though the interaction between the two is still to be determined".

This is not a trivial question as it touches heavily on ongoing debates on how different "functional" regions of the head -i.e. neurocranium = brain, visceral cranium = feeding and breathing and other aspects such as head carriage (i.e. linked to locomotion) interact with each other. The face and the vocal tract are not isolated cranial regions but link tightly with the rest of the cranium.

I am fully aware that this is absolutely beyond the scope of the current study but in this case, the text should be clear that it is not the goal of the study to contribute to that discussion. If this was not the intended reading of the text, I do apologise but in that case, it might be useful to consider clarifying the text more. This particularly the case as interactions between face and vocal tract development is indicated. All in all, I think the authors study is much stronger on gene effects on facial development than it is on the voice altering effects. I would suggest that this is being considered.

An option would be to lead with the face development and to indicate that substantial alterations of the facial development, the vocal tract anatomy would be affected as well. Particularly, in the discussion, the authors already approach this (p. 16, lines 343-345) to some degree. I would recommend to discuss your findings in relation to Nishimura (2006) on the differences in face flattening and vocal tract development.

Following the reviewer's comment, we have made several changes to the manuscript. First, we added the following paragraph to the discussion, emphasizing that the cranial observations are more direct than the vocal tract ones and further emphasizing the link between the two with Nishimura's paper: "Although the DMRs we report most likely exist throughout the skeleton, including the larynx, the evidence we present for the cranium is more direct, as the patterns are observed in modern human and chimpanzee crania. Importantly, it has been suggested that the 1:1 vocal conformation could have been entirely driven by cranial, rather than laryngeal, alterations [Nishimura, 2006]. Once archaic human cranial samples are sequenced, these

observations could be more directly tested". Second, we have added another reference to the Nishimura 2006 paper and further emphasized that the flattening of the face and the descent of the larynx are processes that are tightly linked: "The 1:1 proportion was reached through retraction of the human face, together with the descent of the larynx, pulling the tongue with it, and suggesting that the two process are tightly linked". Third, following the reviewer's suggestion, we clarified that disentangling the exact developmental interaction between facial retraction and laryngeal descent is beyond the scope of the study: "the interaction between the two is still to be determined, as their exact developmental pathways are beyond the scope of the current study".

I understand that the authors are investigating regulatory genes, which in turn influence/cascade other genes and processes but in that case I would say that I am not comfortable with the statements on how genes influence the voice. There are two reasons for this. First, it is very difficult to state that genes have a direct influence on a feature like the voice. As recently indicated, modern speech sounds might have undergone substantial evolution since the palaeolithic based on environmental factors only and if that is indeed the case as Blasi et al. might be able to demonstrate, then it would be very difficult to link genes directly to voice. Furthermore, larynx anatomy likely is far more affected in relation to its primary functions (see below).

We understand the concern of the reviewer and share the view that linking genes to phenotypic effects on the voice is a very challenging task. However, in our study, we examine genes that are already known to directly affect the voice, usually through laryngeal alterations. In addition, we have taken several measures to verify that the link between these genes and voice is established, and there are several points to take into account in this regard:

- (1) All of our data come from monogenic diseases, where one gene was shown to underlie the disease and directly cause its phenotypes. Thus, the link between the gene and the disease is established to a very high degree [Hamosh, A., Scott, A. F., Amberger, J. S., Bocchini, C. A. & McKusick, V. A. Online Mendelian Inheritance in Man (OMIM), a knowledgebase of human genes and genetic disorders. *Nucleic Acids Res.* 33, (2005)].
- (2) For four out of the five top genes, laryngeal alterations were shown to appear in over 50% of patients (based on the *typical* classification of this phenotype in the Human Phenotype Ontology (HPO) database [Köhler, S. et al. The Human Phenotype Ontology project: Linking molecular biology and disease through phenotype data. *Nucleic Acids Res.* 42, (2014)]). We agree with the reviewer that laryngeal alterations are likely to affect other primary functions of the larynx, such as swallowing and breathing, but specifically, the genes we analyze were previously linked to vocal phenotypes.
- (3) Identifying vocal alterations that are driven by neurological changes is substantially harder than the types of phenotypes we focus on, where in many cases, the vocal alterations reported in HPO are driven by alterations to the larynx [Köhler, S. et al. The Human Phenotype Ontology project: Linking molecular biology and disease through phenotype data. *Nucleic Acids Res.* 42, (2014)].
- (4) The Blasi et al. paper illuminates a very interesting angle in the evolution of speech, where some vocal aspects are affected by cultural and environmental rather than anatomical/neurological effects. In our study, given the mechanism that was reported in many of the disorders, the vocal alterations we are focusing on are unlikely to be driven by environmental/cultural differences, as healthy individuals from the same families do not exhibit these phenotypes [Hamosh, A., Scott, A. F., Amberger, J. S., Bocchini, C. A. & McKusick, V. A. Online Mendelian Inheritance in Man (OMIM), a knowledgebase of human genes and genetic disorders. *Nucleic Acids Res.* 33, (2005)].

Following the reviewer's comment, we tried to clarify these points by adding the following paragraph to the "Validation of face and larynx enrichment in Gene ORGANizer" chapter: "The link between genetic alterations and vocal phenotypes is complex. Some brain disorders result in vocal alterations in which the mechanism is very difficult to pin down. In disorders where the vocal phenotype is the result of skeletal alterations to the larynx, however, the mechanism is more easily characterized. Although the mechanism leading to vocal alterations in some of the genes we report is unknown, many of the disorders are skeletal, suggesting the mechanism is related to anatomical changes to the vocal tract. Such changes could also affect more primary functions of the larynx, such as swallowing and breathing. However, the enrichment we observe in Gene ORGANizer shows these genes were also shown to drive vocal alterations in the disorders they underlie. Vocal and speech alterations were also shown to be driven by cultural, dietary and behavioral changes affecting bite configuration [Blasi et al., 2019]. Here too, these factors are unlikely to underlie the vocal alterations in the genes we report, as individuals from the same family, who do not carry the dysfunctional allele, were not reported to present any vocal phenotypes."

In some parts of the text (e.g. p.10, lines 207-212, p. 10, lines 213-220, p. 11, lines 236-238, p. 12, lines 249-254 and p. 15, lines 329-333) it is indicated that there is a direct affect of the genes on the voice. These text regions are to some degree also confusing as they argue aspects of larynx anatomy, larynx development, larynx mechanisms and voice quality as support of the genetic results. This per se is not as problem but all these topics are highly complex and not adequately introduced. Perhaps the recent publication by Gunz et al. (2019) might serve as a template on how to present a relatively good chain of evidence of the effects of regulatory genes of phenotype?

We thank the reviewer for his/her comment and see the reviewer's point. Indeed, while writing the manuscript we encountered a problem: ideally, we would like the reader to become familiar in the introduction with the potential effects of laryngeal and craniofacial alterations to the voice. In the first drafts of the paper, we introduced these subjects in the introduction. However, based on the feedback we received, this caused a very problematic misconception: describing laryngeal anatomy in the introduction caused many readers to think that our work was aimed at finding such differences, whereas the case is the opposite – the enrichment analyses were blind, simply entering the list of DMRs to the enrichment tools, and reporting the most enriched and significant organs linked to these DMRs. As this is probably the most central point to understanding our work, we decided to move the explanations of laryngeal anatomy to the discussion, while shortly describing the link between skeletal alterations to the larynx and vocal phenotypes in the results. We realize that none of the options is optimal, but we think that the latter better describes the logic behind this study. In other words, the reported organs could have been completely different if the enrichment analyses would have resulted in other organs. Nevertheless, we understand now that some of these descriptions were not clear enough, and thus we have tried to integrate the reviewer's comments into the text while keeping in mind the length constraints of the manuscript. We have also added several citations to point readers to further reading (see next).

My main concerns are thus with some lack of clear definitions of vocal tract anatomy and functions and particularly from linking these with the genetics. I would like to present some them to the authors in order to assist them in strengthening the clarity of their findings

1. Anatomy of the larynx
2. Function of the larynx

3. Definition of speech and voice
4. Development of the larynx vs. phenotypic presentation of the larynx

We thank the reviewer for the in-depth description that follows. We would like to stress out that, while the reviewer raises many interesting questions, parts of their descriptions involve levels of detail that are beyond the resolution that can be achieved in our study. Details on how exactly genes affect larynx anatomy and how anatomical changes affect changes in vocalizations are yet to be fully elucidated, even from a pure anatomical aspect. Our study points to substantial regulatory alterations to genes that affect the anatomy of the voice box, but cannot determine the exact mechanism. We have tried to make this point more clear in the text, see our specific responses below.

1. Anatomy of the larynx

First, there is need for some corrections of the anatomy of the vocal tract reported.

On page 10, lines 220-221, concerning the attachments of the vocal folds within the larynx: the vocal folds are posteriorly anchored at the vocal processes of the arytenoid cartilages. Anteriorly, the vocal folds attach to the thyroid cartilage (lamina), not the cricoid cartilage (any anatomy textbook can clarify that). The cricoid cartilage is the link element between the trachea and the rest of the laryngeal cartilages and it has synovial joints with both the thyroid and the arytenoid cartilages.

We thank the reviewer for this correction. We removed the mentioning of the cricoid cartilage and changed the text to: “Importantly, the laryngeal skeleton, and particularly the arytenoid cartilage to which the vocal cords are anchored, are developmentally closest to limb bones, as these are sister skeletal tissues that derive from the somatic layer of the lateral plate mesoderm”.

On page 16, lines 338-339 - it should be different vocal tract configurations not conformations. Vocal tract configurations indicate shape variation of the vocal tract due to adjustments of vocal tract filters such as the position of the lips, teeth, soft palate, tongue shape and position and overall length of the vocal tract and degree of opening of the mouth. This in turn leads to variation in the vocal tract frequency output (e.g. vowels a, i, u or consonants).

We thank the reviewer and corrected the text accordingly to “configurations”.

The authors indicate that they consider the size? shape? position? (this is not clear from the text e.g. p. 10, lines 218-222, p. 17 lines 376-378) of the arytenoid cartilages essential for voice alteration. However, no literature is cited on what the arytenoids are actually doing in breathing, feeding and speech and how altered anatomy of these will alter the voice. I would think the authors will have quite a difficult time to find exactly what happens to the voice due to altered arytenoid anatomy. The literature is not clear what changes consistently even when they are paralysed other than that swallowing becomes difficult. I would invite the authors to find support for their arytenoid statements or alter the text.

Following the reviewer’s comment, we realized that the location of this sentence in the text made its interpretation dual. The reason for mentioning the arytenoid cartilage here was not in order to describe its potential effect on vocalization (which we agree is unclear), but rather to refer to its differentiation lineage, suggesting some DMRs are likely shared between limb and laryngeal tissues. However, as the reviewer mentions, the sentence received an anatomical interpretation based on its proximity to the previous sentence talking about vocal phenotypes driven by skeletal alterations to the larynx. Therefore, we moved this sentence to the next section after the sentence: “This is also evident in the fact that DMRs identified between species

in one tissue often exist in other tissues as well”, which should emphasize that the topic is identification of DMRs in developmentally close tissues.

2. Function of the larynx

Whilst humans take the mammalian use of the larynx for acoustic signalling to new levels of complexity, one always needs to keep in mind that like all other mammalian larynges (and even bird and reptile ones) it is first and foremost part of the respiratory system. I am aware that the authors are interested in the impact of genetic variation on the variation of speech production but since these are regulatory genes, I do not think it is justified to completely ignore the role these genes might play in those main functions as well.

The main function of the larynx is to regulate airflow to and from the lungs to guarantee adequate oxygen levels at all times and to safekeep the latter from invasion of foreign objects during feeding and swallowing processes. The way the larynx can safely close the airway is due to the adduction of the vocal folds and the vestibular folds and the lowering of the epiglottis. Any anatomical changes to the larynx will have to guarantee these main functions prior to any speech-production specific anatomical variations and this should be expressed in the text.

Indeed, the primary functions of the larynx in mammals were missing from the text, and we have thus added the following paragraph to the “Validation of face and larynx enrichment in Gene ORGANizer” chapter:

“Importantly, the larynx is an organ which is primarily involved in breathing and swallowing in mammals. In humans, the larynx is also used to produce complex speech, but not every change to the larynx necessarily affects speech. Despite these additional functions, the genes reported by Gene ORGANizer and HPO were specifically associated with voice alterations, suggesting that although they could have additional effects, their effect on the voice is their most shared function.”

This is particularly the case for literature cited to support the role genes have in affecting the voice. I have read in detail all your citations supporting your statements regarding genes affecting the voice (i.e. 20, 21, 22, 23, 28, 33 and 43) and studied the extended data table 5. Nowhere in table 5 are voice alterations linked to your genes and in all these citations, there are 2 lines of text in totals referring to a hoarse voice and a high voice. This is not evidence that these genes affect the voice. However, what is striking to me is that all these papers (and table 5) describe severe and clear malformations of the larynx and respiratory system in relation to breathing function (e.g narrowed larynx inlet, collapsed cartilage framework, narrowing of the trachea, tongue obstruction etc.). You might state that due to the severe breathing issues, the airflow for speech production is not given but I would think that this does not present a strong case for functional voice affection by a gene. I would focus on the primary functional aspects of these larynx malfunctions and leave the voice affecting statements out of the text.

We thank the reviewer for his/her comment and for checking the literature so thoroughly. The evidence for the link between these genes and vocal phenotypes is often separated into studies that linked the gene to the disorder and studies that described the phenotypes of the disorder, together providing the gene-disorder-phenotype link. An example of that is *POC1A*, which was shown to underlie the SOFT syndrome, a disorder that affects stature, facial phenotypes, the pitch of the voice and more. Before the identification of the underlying gene, reports of this syndrome focused on its phenotypes, describing (among other phenotypes) high-pitched voice in several patients [Shalev et al., *A distinctive autosomal recessive syndrome of severe disproportionate short stature with short long bones, brachydactyly, and hypotrichosis in two consanguineous Arab families*, European Journal of Medical Genetics, 2012]. Later, the

molecular mechanism of this disease was deciphered and *POC1A* was shown to underlie the syndrome [Sarig et al., *Short Stature, Onychodysplasia, Facial Dysmorphism, and Hypotrichosis Syndrome Is Caused by a POC1A Mutation*, *AJHG*, 2012]. In fact, this is the way the Human Phenotype Ontology database is constructed, where disorders are first defined and linked to their symptoms, and then each disorder is linked to the gene which was shown to underlie it. In Gene ORGANizer, HPO phenotypes are simply translated into body parts. As seen in Extended Data Table 5, which simply presents the raw HPO data, many of the genes have been linked to vocal phenotypes (e.g., “abnormality of the voice”, “high-pitched voice”, “hoarse voice”, “dysphonia”, and “difficulties in speaking, chewing and swallowing”). These phenotypes were linked by the HPO team to these genes through the disorders they underlie. Following the reviewer’s comment, we now realize that some of the citations we used were confusing, as they provided just one side of the link (gene-disorder or disorder-phenotype) and thus confused the reader. We have therefore decided to simplify the citations in this part by directly referring to the HPO and Gene ORGANizer databases, and when individual facets of the phenotype are described, we cited the study that discussed it.

This leads to point 3 Definition of speech and voice

Speech-production specific anatomical variations of the larynx are difficult to understand and have not been studied much comparatively between humans and other living primates. The exception from this might be the presence or absence of vestibular air sacs and what that means for the production of single formant frequencies in speech. However, little is known about how the difference in e.g. thyroid or cricoid cartilage (if reported at all) influences the production of speech or the quality of voice. The role of the larynx in speech is not clearly defined in the text but this is crucial for the way the study reports on what the larynx does or does not in speech and what this means for the quality of voice, which is again something different. In speech-production, the larynx regulates the airflow from the lungs in speech and makes sure that the vibrating vocal folds create steady formant frequencies which are recognised as acoustic signals. The airflow is modified in the larynx so that a succession of varying speech elements can be produced. This is achieved via the speed and duration of the vibration of the vocal folds and the size and shape of the opening of the glottis (*rima glottidis*). The position of the larynx and the tension of the vocal folds also play a role in colouring/altering the quality of the voice, such as volume, pitch etc.

We thank the reviewer for their comment and explanation. Indeed, because of limits on the manuscript length, we did not include an in-depth description of the anatomy of speech and vocalization. This is also because there is still much more to study in the genes we report before it is possible to decipher if and *how* their differential regulation affects vocalization and speech. We do not think we can determine that based on current knowledge, and so we focus this study on the regulatory divergence and evolutionary trends observed, rather than deciphering the exact molecular mechanism and its phenotypic outcome. Nevertheless, to allow readers who are interested in a more detailed explanation of the anatomy and physiology of vocalization and speech further information, we now refer the reader to Lieberman’s *The Evolution of the Human Speech* (“For an in-depth review of the anatomy of vocalization and speech, see [Lieberman]).

It is to me not clear in the text if the authors consider quality of voice, which varies substantially between individual humans and also at different stages of life within a single individual is evolutionary important or if they think larynx anatomy variation is important for speech production per se? I would have thought the latter but I would be much more precise on that issue throughout the text. If the latter is the case, I would be precise on which differences in larynx anatomy you consider evolutionary relevant. In context of this, I also found the reports on the developing larynx difficult to track through the text.

We thank the reviewer for his/her comment. Unfortunately, we do not think there is enough information to decide between the options and any precise conclusion regarding it would probably be unsupported. We do not think that at this stage, beyond the relative position of the larynx, we can predict which differences in laryngeal anatomy would be evolutionary relevant. To further clarify this, we added the term “vocalization” to our final statement in the manuscript: “The mechanisms leading to such extensive regulatory shifts, as well as if and to what extent these evolutionary changes affected vocalization and speech capabilities are still to be determined”

4. Development of the larynx vs. phenotypic presentation of the larynx

The laryngeal cartilages without any doubt are mesoderm derived and develop from the branchial arches 4-6. The tracheal cartilaginous rings are also mesodermally derived and I would like some clarification if the authors have accounted for the expression of their investigated genes in these structures as well.

Gene ORGANizer investigates all body parts in its gene enrichment analyses, including the trachea. We did not detect significant enrichment in the trachea, however, we do see non-significant enrichment of trachea-related genes (1.27x, FDR = 0.32) which might be related to the association the reviewer mentions and might have been more significant given more statistical power.

Importantly, the larynx is located at the junction with the endodermally derived lungs and trachea. Tissues originating from the developing trachea immigrate into the larynx to form parts of the vocal folds, including the vocalis muscle (visceral muscle) and all soft tissues of the larynx. There are several inferences made to vocal folds in the text and in relation to the genes investigated but there is no specification of the varying developmental tissue origins between the larynx cartilages and the vocal folds. I would be grateful if there is clarification if these genes have an expression in the development of the vocal folds as well. Furthermore, the formation of the larynx is complex with a temporary closure of the glottic opening during the prenatal development (around week 10). I would therefore be very careful in making statements about the influence chondrossification regulating genes have on the vocal folds.

We thank the reviewer. Unfortunately, the vocal folds are a very under-studied body part and we are unaware of any datasets allowing to map which genes are expressed there during development. We now explain this point in the “Computing correlation between methylation and expression” chapter: “Future studies providing RNA expression levels for the laryngeal skeleton and vocal cords might provide further information on the methylation-expression links of these genes.”

In summary, it should be possible to improve some of the definitions of anatomy vocal tract and its functions and be more defined on the affects the genes have on the indicated structures. I would recommend considering discussing the findings in relation to what is already known about facial and vocal tract development but do not think these changes are difficult or extensive to undertake.

We thank the reviewer for his/her specific comments. We have tried our best to clarify these points in the text and we hope the reviewer finds them satisfactory.

Literature cited:

Blasi, D.E. et al. (2019). Human sound systems are shaped by post-Neolithic changes in bite

configuration. *Science* 363 (15th March)

Gunz, P. et al. (2019). Neanderthal Introgression sheds light on modern human endocranial globularity. *Current Biology* 29: 120-127

Nishimura, T (2005). Developmental changes in the shape of the supralaryngeal vocal tract in chimpanzees. *American Journal of Physical Anthropology* 126(2): 193-204

Nishimura, T., Mikami, A., Suzuki, J. Matsuzawa, T. (2006) Descent of the hyoid in chimpanzees: evolution of face flattening and speech. *Journal of Human Evolution* 51(3):244-254

Reviewers' Comments:

Reviewer #1:

Remarks to the Author:

All my comments suggestions of edits have been thoroughly addressed. I do not have any further concerns.

Reviewer #2:

Remarks to the Author:

The clarification of the authors in the extensive rebuttal and in the new text and also in the new Extended data Figure 1 have helped to understand what was done. However, I am afraid they are not sufficient to convince me that the main claim of this paper is justified. This main claim is in the title: "Regulatory Divergence of Genes Affecting Vocal and Facial Anatomy in Modern Humans". This implies that this is not a general phenomenon seen in all apes or all hominids, but a phenomenon specific for modern humans. If true, this would be a really important finding, but requires evidence appropriate to its importance. Alternatively, one would need to dampen the claim according to the evidence. In any case, it should be made clear whether the authors claim specificity for the MH clade or not. The authors provide a plethora of analyses to support their claim and while the paper has certainly improved, I still find analyses quite intransparent and difficult to judge with respect to the main claim. One important point is that, while the authors say that their analysis is conservative, it might be that their strict filtering for variation in modern samples might cause the observed effect. If, e.g. the genes associated with Vocal and Facial Anatomy would tend to be generally less variable, they would get enriched. So coming back – with the help of Ext. Fig 1 to my suggestion of primate permutations (which wasn't very clearly formulated before, sorry):

If one would exchange the Ust'Ishim sample with the Neanderthal sample and would run the entire analysis as if the Neanderthal sample would be the modern human (and vice versa), and would observe a similar (or not significantly different) enrichment as in the current analysis, this would be problematic for the main claim. Hence, this permutation should be tested. While this would still not get rid of all my worries regarding the correctness of the main claim of the paper, it would certainly be needed.

Reviewer #3:

None

Reviewer #4:

Remarks to the Author:

Dear Editor, dear authors,

Thank you very much for the opportunity to further comment on the manuscript "Regulatory Divergence of Genes Affecting Vocal and Facial Anatomy 1 in Modern Human", I think the exchange with the authors has cleared some of the issues I had regarding the quality and accuracy of the anatomy of the vocal tract and face and I think the manuscript is better for it. Despite the fruitful exchange with the authors, I am sorry to say that I still have some concerns regarding the link between genes and their effect on the voice which I have not found as well addressed and I also do have some concerns about the imprecision on discerning between "voice" and speech evolution. This is an important point and one which I would like to have the authors take another look at.

This is particularly the case because any Nature publications usually have a fairly broad spectrum of readers and this publication will not only interest geneticists but also interested parties from

other fields of expertise and research. It is my conviction that if you want to address a wider public, you also need to make sure that you make an extra effort on communicating your findings in a way that makes them accessible for a wider public and that you maintain high scientific standards in the process.

The main issue here is that "modern human speech" is an extremely complex system, not a single trait. You cannot directly link such a complex system, which is not only depending on anatomy but also on physiology (neuromotor, respiration) and the environment to one or even 5 individual genes and then claim these genes are "voice affecting". Furthermore, the role of the reported "voice quality" aspects which vary in relation to some pathological conditions in speech evolution is not explained either.

This is however a fundamental imprecision: The manuscript tries to make a connection between variations in gene expression over time and within different hominoid species and facial and vocal tract anatomy variation over time, indicating that the differences in facial anatomy likely extend to differences in vocal tract anatomy and hence speech abilities (this is likely so). This does have an important impact on when and within which hominoid species modern human-like speech abilities might become possible. Modern human-like speech abilities in the context of speech evolution should mean the production (vocal tract), reception (hearing anatomy) and processing and interpretation (neuroanatomy and physiology) of modern human-like speech sounds.

The manuscript mostly reports gene expression links between voice quality variations, such as pitch (low/high frequency) and timbre (hoarseness, breathiness of voice) as evidence for this. The tricky issue here is that these are purely variations of the modern human voice quality and have no impact on whether speech ability (production/reception/process) is present or not. In this, the manuscript fails to indicate the role variation in voice quality is to have for the evolutionary potentially meaningful presence or absence of speech abilities. In other words, gene expression might indicate variation in voice quality but this is not linking gene expression to modern human-like speech abilities.

However, the fairly easy way forward would be to be more precise in the reporting of these effects and restrict them to face and vocal tract anatomy-affecting rather than making tenuous claims on single gene impact on speech abilities. In this way, any imprecision will be avoided. At the same time, I find that these changes to improve the manuscript further will consist of minor changes only which should not take much time to implement.

Global changes:

Please replace "vocal cords" throughout the text, keywords and figure legends with "vocal folds". It is the more accurate anatomically consented term recommended by Terminological anatomica.

Please replace "voice box" throughout the text, keywords and figure legends with "larynx". Voice box is not an anatomically correct term.

Please replace voice-affecting gene(s) throughout the text, keywords and figure legends with vocal tract anatomy-affecting or larynx anatomy-affecting genes. That is far more scientifically accurate

Summary section

P 3, summary, line 61

Please rephrase text as follows:

We show that genes affecting facial and vocal tract (i.e. upper respiratory tract) features went through....

P 3, summary, line 63

Please rephrase text as follows:

Widespread hypermethylation in a network of face and vocal tract anatomy affecting genes

Result section

P5, line 98

Please rephrase text as follows:

Observed in genes that regulate facial and vocal tract anatomy

P7, line 151, section title

Please rephrase as:

Face and vocal tract affecting genes are derived in AMHs

P7, line 158

Would it be trabecular morphogenesis rather than trabecula morphogenesis?

P7, line 159

I think it should be skeletal musculature rather than skeletal muscle

P7, Line 163

Please provide some evidence that this is the case for your genes – I do not mean citing Gene ORGANizer but explain to your readers how a regulatory gene like the ones you describe have the same power of phenotypical affect like an organ-building protein producing gene

P8, line 167

Please rephrase text as follows:

Three regions; the face, larynx and pelvis

P8, line 173

Please rephrase text as follows:

Enrichment of the face and larynx

P8, line 174/175

Please rephrase as follows:

That the supralaryngeal vocal tract (nasal and oral cavities and the pharynx) is the most enriched body part we studied

P8, line 178-181

The palate is part of the face – anatomically, that is not separate from the rest of the craniofacial development. If you want to make a point that it is different genetically, please rephrase these two sentences to make that clear, otherwise I would just combine them

P9, line 185

Please rephrase as follows:

Here, the over-representation of genes affecting larynx anatomy is even more pronounced

P9, line 194

Please rephrase as follows

...for face, the larynx, particularly the vocal folds and pelvis, respectively (please ensure you rearrange relevant values as well)

P9, line 198

Please rephrase as follows

...for face, the larynx, particularly the vocal folds and pelvis, respectively (please ensure you rearrange relevant values as well)

P 10, line 213,

Please rephrase as follows:

Enriched in genes affecting the face and vocal tract anatomy

P 10, line 220

I would amend as follows:

The face affecting genes are known to have a clinically significant effect shaping mainly the protrusion of

P 10, line 222-225

This would – in my opinion be the most concise variation to report your findings on the larynx affecting genes:

The vocal tract and larynx affecting genes presented in this paper show relevant involvement in laryngeal cartilage and soft tissue variation. Clinical phenotypes can be of high severity, with high impacts on normal breathing functions, to the point where cause of death is due to respiratory distress. SOX9 and NFIX are often associated with laryngomalacia (Extended Data Table 5), a collapse of the larynx due to malformation of the laryngeal cartilaginous framework and/or malformed connective tissues, particularly during inhalation.

Less severe phenotypical expressions of the reported genes involve individual variation of voice quality in the form of pitch variation (high in patients suffering from XYLT1 mutations) and sometimes hoarseness of the voice (reported for some patients with mutations of ACAN). Whether this is due to a variation of the vocal tract and larynx anatomy influenced by the ACAN mutation or due to a scaled down vocal tract size in the case of the XYLT1 mutation which also causes primordial dwarfism is not yet clear.

Extended table 5 – the title of this table is misleading or incomplete, as not all of these genes listed here were investigated for DMS in the current study as far as I can tell. If that is not the case and I got this wrong, I apologise but in that case, if only the 5 mentioned genes are significantly methylated, then why report all the other genes? In that case you can reduce the table to only the relevant genes.

Please alter the title to indicate that you compare your 5 genes with the rest and please highlight your 5 genes in the table (e.g. bold writing or coloured background).

It might also be worth providing a definition for osteomalacia as I doubt most of your readers would know that this indicates a malformed laryngeal cartilaginous skeleton, particularly affecting the epiglottis and hence severely affecting normal breathing functions.

P 10, line 225

Following the suggested alterations above please delete sentence lines 225-226

P11, lines 229-232

This sentence is convoluted and developmentally not very clear. I would rephrase as follows:

The cartilages of the larynx (thyroid, cricoid, arytenoids and epiglottis) share an origin from the somatic layer of the lateral plate mesoderm with the cartilaginous tissue of the limb bones prior to their ossification. Thus, is likely that...

P 11, line 239

Please rephrase as follows:

...in many face and vocal tract anatomy affecting genes have changed...

P 11, line 247

Please rephrase as follows:

...changes within face and vocal tract anatomy affecting genes is not different than expected

P 12, line 258

Please rephrase as follows:

Strikingly, when ranking DMGs according to the fraction of AMH-derived CpGs, our top five

skeleton-related DMGs (ACAN, COL2A1, NFIX, SOX9 and XYLT1) are evidenced to be involved in clinically relevant variation of the lower and midfacial protrusion and the variation of the vocal tract anatomy. This is very interesting

Changes to discussion section:

I am particularly glad that the authors have changed the manuscript in a way that speculations about the effect of variation in the genes described are put mainly in the discussion section. There are however still some improvements to be made.

P. 16, lines 342-364.

This is very well written and a very good account – at the right level of precision for this paper – what is known and what can be said.

Following this section, I miss how the authors link this excellent text with their speculations about the importance of NFIX in the next section. Why are you choosing NFIX for a good candidate to resolve the issues you just listed in the section before?

P.17, lines 365-386

Line 366-367 NFIX does not impair speech abilities per se but mutations of it result in a multitude of clinical features put together as either the Marshall-Smith or Sotos-like syndromes, depending on where your NFIX mutation is located. Marshall-Smith syndrome (MSS) is characterised by a distinct facial configuration and often severe respiratory distress due to malformations of the upper airways and larynx. Speech in these patients is often, but not always, absent. If it is present, the patients do not seem to have too much problems producing speech sounds but see van Balkom et al. (2011) for a more detailed report on that.

Absence of speech in many MSS patients is due to a combination of a pathological facial and nasopharyngeal configuration, larynx configuration and resulting respiratory deficits, severe cognitive dysfunctions, potential oromotor dysfunctions and aggravated by sensorineural/conductive hearing loss. This is a summary of the literature citation 41 (Shaw et al., 2010) given by the authors who have chosen only to pick out of it the one aspect that suits the interpretation of their data without acknowledging the rest.

Since many MSS patients die very early from severe breathing issues, it would be more accurate to call this a breathing affecting gene. If you want to argue in this manner, then you might as well argue that NFIX is a gene that impairs bipedal walking abilities as many of the patients have a distinct gait due to common hip dysplasia and kyphoscoliosis of the spine. Needless to say that both the larynx pathologies and the hip and spine issues are due to affected bone and connective/cartilaginous tissue rather than directly affecting complex functions such as speech and walking.

At the same time, NFIX is associated with more than one syndrome – and speech issues are reported for Marshall-Smith syndrome only Martinez et al. (2015). The Sotos-like syndrome patients do not have laryngomalacia or respiratory issues. If NFIX really were a speech-impairing gene, then this would have to have that effect on all patients who are affected by NFIX mutations.

In the same vein of arguing, it is worth mentioning the following: ACAN (single locus, single gene) was linked to three different phenotypes, of which one reports a hoarse voice in 3 patients, whereas the rest of the phenotypes linked to 100+ patients for whom there is no mention of any voice quality alterations at all.

The most commonly reported phenotypical traits for ACAN mutations are short stature, face morphology variations, respiratory distress and waddling gait. Hence the same issues of whether ACAN is a bipedal gait affecting gene applies as well whereas it actually is – like NFIX and SOX9 regulating chondrification and ossification processes.

Furthermore, ACAN's association with short stature (mostly of the primordial dwarfism type in contrast to e.g. achondroplasia type dwarfisms) likely results in a scaled down vocal tract size which is generally associated with a high pitched voice. This might mean that you can have a non-altered larynx anatomy and still end up with a variation in voice quality. Yet this is not addressed either.

I just detailed this here as a demonstration on just how difficult it is to actually support your bold statements about what your genes are actually doing or not.

Lines 366-370

Sentence starting on 366-368 please remove it, you cannot support that statement.

Line 368- 370

Sentence starting "To investigate..." at the end, please alter to "... examined its clinical relevant skeletal phenotypes".

Please make it clear here that the following are non-normal phenotypes.

I can accept the statements in lines 375-386.

P. 18, line 387, please loose the interestingly at the beginning of the sentence.

Section lines 388-392

This section is not acceptable in this form.

First, I read the reports on MSS laryngomalacia and the literature on laryngomalacia in detail again. There are many different types of laryngomalacia and most of them involve predominantly the size, shape and placement of the epiglottis and/or the arytenoids in relation to the epiglottis, particularly the soft tissues connecting them with the epiglottis. I did not find any literature which directly stated that the arytenoids are frequently malformed in Marshall-Smith syndrome. So please stay within what is known and report that some of the patients suffering from MSS do have an abnormal laryngeal cartilage framework and soft tissues which create severe respiratory distress and leave the whole speculation about arytenoids and vocal folds out of this.

If you want to say something about NFIX regulation of the larynx, then state that it is likely due to the alteration this causes to the tissues NFIX codes for and that it can have a relationship with the facial structures being altered, that would be acceptable and would match better with the explanations you are putting forward for your adult methylation patterns observed.

The rest of the discussion is ok as it is.

Methods section

Validation of face and larynx enrichment in GeneORGANizer

P 75, line 1470-75

Please remove the text "the DMGs which affect the voice" from this section. Otherwise I will insist on you giving a percentage separately for voice in line 1743 as well, not just for larynx, pelvis ect. I recommend replacing this with:

genes affecting the larynx and vocal tract anatomy and summarising these as 65.

P. 80 lines 1584-1602

First I would like to thank the authors that they acknowledge the importance of the breathing and feeding functions of the supralaryngeal vocal tract. I find it therefore a bit puzzling that this importance is then not presented accordingly in this section of the text. I therefore recommend the following:

Lines 1597-1598, loose the "Importantly" at the beginning of this sentence, then put it in the position of line 1584, at the beginning of this section.

Line 1584, this lead in sentence to the section is misleading, there is no definition given for "vocal

phenotype". Vocal phenotype does not have any meaning in relation to evolution but it has a meaning in clinical studies. This needs to be addressed here. The term "brain disorder" is also not defined at all, neither is vocal alteration.

I would leave this out or be more precise. More precision comes about by talking about modern human voice quality variation in pitch, timbre and volume and to stress that these are very individual.

Lines 1585-1587

I completely disagree with this sentence. Please either make it very clear what you mean by brain disorders, vocal phenotypes and voice alterations or delete it.

Lines 1586-1487

The sentence on links between vocal tract anatomy and voice quality variation – the link is easier to establish – agreed but that concerns the link between the varied vocal tract anatomy and the speech sound output or voice quality, not the genes.

Lines 1590-1592

This sentence is not supported by any statement – please either delete it or provide the evidence for this.

Lines 1593-1596

This sentence reads as a fragment as the second half of it has no context to anything else – the families reported here are not introduced anywhere and nothing is in context.

Lines 1600-1602

This sentence is again not indicating why alteration to voice quality would have an impact on speech evolution. Also still – I am really sorry about this – but I could not follow how a gene reported in Gene ORGANizer does provide the proof that any given gene does have a direct influence on voice quality. Either delete the sentence or explain it again.

Martinez, F. et al. (2015). Novel mutations of NFIX gene causing Marshall-Smith syndrome or Sotos-like syndrome: one gene, two phenotypes. *Translational investigation* 78(5): 533-539

Van Balkom I. et al. (2011) Development and behaviour in Marshall-Smith syndrome: an exploratory study of cognition, phenotype and autism. *Journal of Intellectual Disability Research* 55(10): 973-987

Reviewer #1 (Remarks to the Author):

All my comments suggestions of edits have been thoroughly addressed. I do not have any further concerns.

We appreciate the suggestions raised by the reviewer in previous rounds and are glad they find the current version of the manuscript satisfactory.

Reviewer #2 (Remarks to the Author):

The clarification of the authors in the extensive rebuttal and in the new text and also in the new Extended data Figure 1 have helped to understand what was done. However, I am afraid they are not sufficient to convince me that the main claim of this paper is justified. This main claim is in the title: "Regulatory Divergence of Genes Affecting Vocal and Facial Anatomy in Modern Humans". This implies that this is not a general phenomenon seen in all apes or all hominids, but a phenomenon specific for modern humans. If true, this would be a really important finding, but requires evidence appropriate to its importance. Alternatively, one would need to dampen the claim according to the evidence. In any case, it should be made clear whether the authors claim specificity for the MH clade or not. The authors provide a plethora of analyses to support their claim and while the paper has certainly improved, I still find analyses quite intransparent and difficult to judge with respect to the main claim. One important point is that, while the authors say that their analysis is conservative, it might be that their strict filtering for variation in modern samples might cause the observed effect. If, e.g. the genes associated with Vocal and Facial Anatomy would tend to be generally less variable, they would get enriched. So coming back – with the help of Ext. Fig 1 to my suggestion of primate permutations (which wasn't very clearly formulated before, sorry): If one would exchange the Ust' Ishim sample with the Neanderthal sample and would run the entire analysis as if the Neanderthal sample would be the modern human (and vice versa), and would observe a similar (or not significantly different) enrichment as in the current analysis, this would be problematic for the main claim. Hence, this permutation should be tested. While this would still not get rid of all my worries regarding the correctness of the main claim of the paper, it would certainly be needed.

We thank the reviewer for his/her comment and are pleased that the reviewer finds the current version of the manuscript improved. Following the reviewer's suggestions, we re-ran the entire pipeline on Neanderthal- and Denisovan-derived DMGs, while applying to them all the filters as if they were Ust'-Ishim DMGs. More specifically, all steps up to "FDR filtering" (Extended Data Figure 1) do not depend on the location of the sample in the phylogenetic tree. We then applied the "Variability filtering" and "Lineage assignment" steps to the Neanderthal and Denisova as if they were Ust'-Ishim (and vice versa). This resulted in substantially fewer loci (89 for the Neanderthal and 50 for the Denisovan), which limits statistical power, but can still be used to examine whether there are any trends of enrichment similar to those observed in AMHs. We found no evidence that the filtering process could drive the enrichment of the vocal or facial areas: within Neanderthal-derived loci, filtered as if they were Ust'-Ishim-derived, we found that the vocal cords were ranked only 18th, with a non-significant enrichment of 1.27x (FDR = 0.815, compared to an enrichment of 2.11x within AMH-derived DMGs). The larynx was ranked 76th, and showed a non-significant depletion of 0.87x (FDR = 0.783), and the face was ranked 31st, with a non-significant enrichment of 1.09x (FDR = 0.815). Within Denisovan-derived loci, filtered as if they were Ust'-Ishim-derived, none of the loci were linked to the vocal cords nor to the larynx (FDR = 0.535 and FDR = 0.834, respectively), and the face was ranked 30th (1.29x, FDR = 0.535). This test suggests that the filtering process in itself is very unlikely to underlie the enrichment of the vocal and facial parts within AMH-derived DMGs.

Next, we applied the Neanderthal/Denisovan filters to the Ust'-Ishim-derived loci. This resulted in 792 loci. We found that the vocal cords remained the most enriched body part (1.76x, FDR = 0.032), the larynx was marginally significant (1.53x, FDR = 0.0502), and the facial region was significantly enriched too (e.g., cheek and chin ranked 2nd, 3rd within significantly-enriched body parts, 1.66x and 1.63x, FDR = 0.031 and FDR = 0.013, respectively).

These results are now described in the Supplementary Methods under the section "Validation of face and larynx enrichment in Gene ORGANizer", and are presented in Extended Data Table 4 and referred to in the main text.

Importantly, we do not rule out the option that extensive regulatory changes in genes related to vocal and facial anatomy might have occurred along the Neanderthal and Denisovan lineages as well. Indeed, as we report in Extended Data Fig. 2, parts of the face are enriched within Archaic-derived DMGs. The title of the manuscript does not rule this option either. However, we currently see no substantial evidence supporting this. In light of this, and following this comment, we have rephrased the title to capture more accurately our claim. The title now reads: "Genes Affecting Vocal and Facial Anatomy Went Through Extensive Regulatory Divergence in Modern Humans".

Reviewer #3 (Remarks to the Author):

**Editorial notes: please note that although this reviewer doesn't have remarks to the author, in her/his remarks to the editor, Reviewer 3 feels this study ultimately is a computational study without experimental validation and all the results are about inference and associations.

Putting aside the fact that association and inference are instrumental in scientific work, this study includes a large body of experiments, from the sequencing of novel ancient genomes (La Braña and I1583), through the first methyl-seq experiments on ape bones (chimpanzee WGBS, RRBS and methyl arrays) and the first full human bone methylation maps (WGBS), to validation of hypermethylation across four additional chimpanzee and three additional human bone samples, as well as in chondrocytes. This work is also based on very extensive clinical data, RNA-seq, enhancer mapping, mouse knockout studies, and experiments on the relationship between methylation and expression (both genome-wide and for specific genes, e.g. NFIX). Overall, this study contains 55 orthogonal tests, controls and validations. These include all of the tests the reviewers suggested and cover every step of the analysis. While some of the experimental data described above was produced by other groups, it does not make the data inferior. In fact, it only makes the data more credible and objective, as the experiments were not designed to fit our hypotheses.

Reviewer #4 (Remarks to the Author):

Dear Editor, dear authors,

Thank you very much for the opportunity to further comment on the manuscript "Regulatory Divergence of Genes Affecting Vocal and Facial Anatomy 1 in Modern Human", I think the exchange with the authors has cleared some of the issues I had regarding the quality and accuracy of the anatomy of the vocal tract and face and I think the manuscript is better for it. Despite the fruitful exchange with the authors, I am sorry to say that I still have some concerns regarding the link between genes and their effect on the voice which I have not found as well addressed and I also do have some concerns about the imprecision on discerning between "voice" and speech evolution. This is an important point and one which I would like to have the

authors take another look at.

We thank the reviewer for his/her comments and hope they find the current version of the manuscript to their satisfaction.

This is particularly the case because any Nature publications usually have a fairly broad spectrum of readers and this publication will not only interest geneticists but also interested parties from other fields of expertise and research. It is my conviction that if you want to address a wider public, you also need to make sure that you make an extra effort on communicating your findings in a way that makes them accessible for a wider public and that you maintain high scientific standards in the process.

We agree with the reviewer.

The main issue here is that “modern human speech” is an extremely complex system, not a single trait. You cannot directly link such a complex system, which is not only depending on anatomy but also on physiology (neuromotor, respiration) and the environment to one or even 5 individual genes and then claim these genes are “voice affecting”.

We agree with the reviewer that speech is a complex system, which depends on many different aspects of anatomy, physiology and the environment. With regard to the top 5 genes we report, we do not claim that they are the *only* genes known to affect the voice, nor do we claim that this is the only system they affect. Moreover, their function is not necessarily direct and the hypermethylation we observe did not necessarily result in alterations to the voice. The main result we report is that many genes, which have been previously shown to affect the voice, show signatures of differential regulation in AMHs. In other words, the most significantly shared characteristic of differentially methylated genes in AMHs is that they are known to affect the voice when they lose some/all of their function in humans. This trend possibly (but not necessarily) affected vocal aspects in human evolution, but this should be further tested in future studies. To clarify this point, we rephrased L221 to: “The larynx-affecting genes have been shown to underlie various phenotypes in patients, ranging from slight changes to the pitch and hoarseness of the voice, to a complete loss of speech ability”.

Furthermore, the role of the reported “voice quality” aspects which vary in relation to some pathological conditions in speech evolution is not explained either. This is however a fundamental imprecision: The manuscript tries to make a connection between variations in gene expression over time and within different hominoid species and facial and vocal tract anatomy variation over time, indicating that the differences in facial anatomy likely extend to differences in vocal tract anatomy and hence speech abilities (this is likely so). This does have an important impact on when and within which hominoid species modern human-like speech abilities might become possible. Modern human-like speech abilities in the context of speech evolution should mean the production (vocal tract), reception (hearing anatomy) and processing and interpretation (neuroanatomy and physiology) of modern human-like speech sounds. The manuscript mostly reports gene expression links between voice quality variations, such as pitch (low/high frequency) and timbre (hoarseness, breathiness of voice) as evidence for this. The tricky issue here is that these are purely variations of the modern human voice quality and have no impact on whether speech ability (production/reception/process) is present or not. In this, the manuscript fails to indicate the role variation in voice quality is to have for the evolutionary potentially meaningful presence or absence of speech abilities. In other words, gene expression might indicate variation in voice quality but this is not linking gene expression to modern human-like speech abilities.

Indeed, the relationship between voice quality and speech is an intriguing direction of research. While many of the genes which show differential methylation patterns have been shown to affect voice quality, many others have been shown to affect other aspects of voice and speech (Extended Data Table 5). Based on the results we report in the paper, it would be premature to determine how potential vocal alterations have affected speech, if at all. This is why the results section refers only to aspects of the voice and not speech. In the discussion, we elaborate extensively on the link between vocal anatomy and speech in humans and chimps, and focus on genes affecting the anatomy of the vocal tract.

However, the fairly easy way forward would be to be more precise in the reporting of these effects and restrict them to face and vocal tract anatomy-affecting rather than making tenuous claims on single gene impact on speech abilities. In this way, any imprecision will be avoided. At the same time, I find that these changes to improve the manuscript further will consist of minor changes only which should not take much time to implement.

We agree with the reviewer that the results should focus on the anatomical effects of the genes we report. We were very careful throughout the results section to talk about anatomical effects, and do not mention speech abilities at all. Moreover, we do not claim (or think) that a single gene shaped human speech. Rather, we use the classic way of connecting genes to their function through their phenotypes in monogenic diseases and knockout studies. We conclude that the potential anatomical effects on the face and vocal folds are the result of the combined effect of many genes, from which we highlight five where we see the strongest signals.

Global changes:

Please replace “vocal cords” throughout the text, keywords and figure legends with “vocal folds”. It is the more accurate anatomically consented term recommended by Terminological anatomica.

Following the reviewer’s comment, we replaced “vocal cords” with “vocal folds” throughout the text, figures and tables. We kept “vocal cords” in the keywords to match searches for this term in search engines.

Please replace “voice box” throughout the text, keywords and figure legends with “larynx”. Voice box is not an anatomically correct term.

“Voice box” was replaced with “larynx” throughout the manuscript.

Please replace voice-affecting gene(s) throughout the text, keywords and figure legends with vocal tract anatomy-affecting or larynx anatomy-affecting genes. That is far more scientifically accurate

We thank the reviewer for their suggestion. In this particular case, we think voice-affecting is a more accurate term. This is because in some patients, the source of the voice alteration is unclear and could be unrelated to the laryngeal anatomy (but rather to its physiology, for example, as the reviewer mentioned in their previous comment). This is why using “anatomy” to describe these genes would be inaccurate. Also, we think it is better to be as specific as possible, and the most specific phenotype described for these genes is their effect on the voice. Therefore, we prefer using “voice-affecting” rather than the more general “vocal tract-affecting”.

Summary section

P 3, summary, line 61

Please rephrase text as follows: We show that genes affecting facial and vocal tract (i.e. upper respiratory tract) features went through....

Following the reviewer's suggestion, we changed this sentence to "We show that genes affecting the face and vocal tract went through...". Due to length considerations, we did not add the "(i.e. upper respiratory tract)" as the term "vocal tract" is explained in the main text when it first appears.

P 3, summary, line 63

Please rephrase text as follows: Widespread hypermethylation in a network of face and vocal tract anatomy affecting genes

See comment above for the use of voice vs. vocal tract anatomy

Result section

P5, line 98

Please rephrase text as follows: Observed in genes that regulate facial and vocal tract anatomy

We thank the reviewer for their comment. We think that "affect" is a more accurate term here, as these genes were shown to phenotypically affect these body parts. This also suggests that these genes regulate (directly or indirectly) these anatomical parts, but as the underlying regulatory pathway is unclear, we think it is better to focus on their known phenotypic effect.

P7, line 151, section title

Please rephrase as: Face and vocal tract affecting genes are derived in AMHs

See comment above

P7, line 158

Would it be trabecular morphogenesis rather than trabecula morphogenesis?

We agree with the reviewer that "trabecular morphogenesis" might also be correct, but "trabecula morphogenesis" is how this GO term appears in the database (GO:0061383)

P7, line 159

I think it should be skeletal musculature rather than skeletal muscle

Following the reviewer's comment, we changed the term to "skeletal muscular"

P7, Line 163

Please provide some evidence that this is the case for your genes – I do not mean citing Gene ORGANizer but explain to your readers how a regulatory gene like the ones you describe have the same power of phenotypical affect like an organ-building protein producing gene

We are not sure we understand the comment by the reviewer. We do not claim that the phenotypic effect of structural genes is different than that of regulatory genes, nor do we refer to these types of genes in the text

P8, line 167

Please rephrase text as follows: Three regions; the face, larynx and pelvis

Following the reviewer's comment, we switched the order of face and larynx.

P8, line 173

Please rephrase text as follows: Enrichment of the face and larynx

Following a previous suggestion of the reviewer, "voice box" was changed to "larynx" throughout the text.

P8, line 174/175

Please rephrase as follows: That the supralaryngeal vocal tract (nasal and oral cavities and the pharynx) is the most enriched body part we studied

Following the reviewer's comment, we added "supralaryngeal" to the "vocal tract". We think it is important to keep the explanation of what these body parts are, as many readers are probably unfamiliar with these terms.

P8, line 178-181

The palate is part of the face – anatomically, that is not separate from the rest of the craniofacial development. If you want to make a point that it is different genetically, please rephrase these two sentences to make that clear, otherwise I would just combine them

Indeed, the palate is part of the face. The observation that the GO term "palate development" is enriched too serves as an orthogonal validation to the overall enrichment of the face in Gene ORGANizer (as these are separate databases and draw their information from different sources). Please note that GO terms stand for Gene Ontology terms, and therefore cannot be combined with Gene ORGANizer terms.

P9, line 185

Please rephrase as follows: Here, the over-representation of genes affecting larynx anatomy is even more pronounced

See comment above regarding voice-affecting vs larynx anatomy-affecting

P9, line 194

Please rephrase as follows ...for face, the larynx, particularly the vocal folds and pelvis, respectively (please ensure you rearrange relevant values as well)

We thank the reviewer for their comment. The order of terms is according to their level of enrichment. Additionally, the terms are as they appear on Gene ORGANizer, and therefore "larynx, particularly the vocal folds" is misleading and we think it is better to keep them separate.

P9, line 198

Please rephrase as follows ...for face, the larynx, particularly the vocal folds and pelvis, respectively (please ensure you rearrange relevant values as well)

See comment above

P 10, line 213,

Please rephrase as follows: Enriched in genes affecting the face and vocal tract anatomy

See comment on voice-affecting vs vocal tract anatomy-affecting

P 10, line 220

I would amend as follows: The face affecting genes are known to have a clinically significant effect shaping mainly the protrusion of

As some readers might read “significant” as relating to statistics, we think this change might be confusing

P 10, line 222-225

This would – in my opinion be the most concise variation to report your findings on the larynx affecting genes:

The vocal tract and larynx affecting genes presented in this paper show relevant involvement in laryngeal cartilage and soft tissue variation. Clinical phenotypes can be of high severity, with high impacts on normal breathing functions, to the point where cause of death is due to respiratory distress. SOX9 and NFIX are often associated with laryngomalacia (Extended Data Table 5), a collapse of the larynx due to malformation of the laryngeal cartilaginous framework and/or malformed connective tissues, particularly during inhalation.

Less severe phenotypical expressions of the reported genes involve individual variation of voice quality in the form of pitch variation (high in patients suffering from XYLT1 mutations) and sometimes hoarseness of the voice (reported for some patients with mutations of ACAN).

Whether this is due to a variation of the vocal tract and larynx anatomy influenced by the ACAN mutation or due to a scaled down vocal tract size in the case of the XYLT1 mutation which also causes primordial dwarfism is not yet clear.

We thank the reviewer for this clear and helpful summary. We believe this summary adds clarity to the phenotypes associated with these genes. We added this description, along with some elaboration on COL2A1 too to the “NFIX, COL2A1, SOX9, ACAN and XYLT1 phenotypes” chapter.

Extended table 5 – the title of this table is misleading or incomplete, as not all of these genes listed here were investigated for DMS in the current study as far as I can tell. If that is not the case and I got this wrong, I apologise but in that case, if only the 5 mentioned genes are significantly methylated, then why report all the other genes? In that case you can reduce the table to only the relevant genes.

Please alter the title to indicate that you compare your 5 genes with the rest and please highlight your 5 genes in the table (e.g. bold writing or coloured background).

The table reports all the DMRs in face-affecting and larynx-affecting genes. In this study, we identified DMRs in dozens of face- and larynx-affecting genes (see p.10: “Our analyses identified 56 DMRs in genes affecting the facial skeleton, and 32 in genes affecting the laryngeal skeleton.”, and this table describes all of the DMRs in these genes. The five genes we focus on later in the paper represent the genes with the most extensive methylation changes.

It might also be worth providing a definition for osteomalacia as I doubt most of your readers would know that this indicates a malformed laryngeal cartilaginous skeleton, particularly affecting the epiglottis and hence severely affecting normal breathing functions.

By “osteomalacia” we assume the reviewer meant “laryngomalacia”. We agree with the reviewer that most readers are probably unfamiliar with this term. Following the reviewer’s comment, we added a description of this phenotype to the table.

P 10, line 225

Following the suggested alterations above please delete sentence lines 225-226

Due to word limit considerations, we have added the paragraph above to the supplementary information chapter: “NFIX, COL2A1, SOX9, ACAN and XYLT1 phenotypes” and kept the shorter sentence in the main text.

P11, lines 229-232

This sentence is convoluted and developmentally not very clear. I would rephrase as follows: The cartilages of the larynx (thyroid, cricoid, arytenoids and epiglottis) share an origin from the somatic layer of the lateral plate mesoderm with the cartilaginous tissue of the limb bones prior to their ossification. Thus, it is likely that...

We thank the reviewer for his/her comment and agree that this sentence was not clear enough before. Following the reviewer’s comment, we changed the sentence to: “Importantly, the laryngeal skeleton, and particularly the arytenoid cartilage to which the vocal folds are anchored, share an origin from the somatic layer of the lateral plate mesoderm with the cartilaginous tissue of the limb bones prior to their ossification. Thus, it is likely that many of the DMRs identified here between limb samples also exist in their closest tissue – the laryngeal skeleton.”

P 11, line 239

Please rephrase as follows: ...in many face and vocal tract anatomy affecting genes have changed...

See comment above about voice-affecting vs vocal tract anatomy-affecting genes.

P 11, line 247

Please rephrase as follows: ...changes within face and vocal tract anatomy affecting genes is not different than expected

See comment above about voice-affecting vs vocal tract anatomy-affecting genes.

P 12, line 258

Please rephrase as follows: Strikingly, when ranking DMGs according to the fraction of AMH-derived CpGs, our top five skeleton-related DMGs (ACAN, COL2A1, NFIX, SOX9 and XYLT1) are evidenced to be involved in clinically relevant variation of the lower and midfacial protrusion and the variation of the vocal tract anatomy. This is very interesting

We thank the reviewer for their comment. We realize now that this sentence was unclear, and following the lines of the reviewer’s suggestion, we changed it to “when ranking DMGs according to the fraction of AMH-derived CpGs, all top five skeleton-related DMGs (ACAN, SOX9, COL2A1, XYLT1, and NFIX) are known to affect lower and midfacial protrusion, as well as the voice”

Changes to discussion section:

I am particularly glad that the authors have changed the manuscript in a way that speculations about the effect of variation in the genes described are put mainly in the discussion section. There are however still some improvements to be made.

P. 16, lines 342-364.

This is very well written and a very good account – at the right level of precision for this paper – what is known and what can be said.

We thank the reviewer for his/her comment.

Following this section, I miss how the authors link this excellent text with their speculations about the importance of NFIX in the next section. Why are you choosing NFIX for a good candidate to resolve the issues you just listed in the section before?

NFIX was chosen as our focus gene because within the top five genes, it shows the strongest link between methylation and expression. As NFIX phenotypes are tightly linked to facial and laryngeal anatomy, we also analyzed them further in this chapter and found that their predicted direction based on NFIX activity match observed Neanderthal-AMH differences, thus strengthening the possibility that NFIX might have played a role in shaping these features.

P.17, lines 365-386

Line 366-367 NFIX does not impair speech abilities per se but mutations of it result in a multitude of clinical features put together as either the Marshall-Smith or Soton-like syndromes, depending on where your NFIX mutation is located. Marshall-Smith syndrome (MSS) is characterised by a distinct facial configuration and often severe respiratory distress due to malformations of the upper airways and larynx. Speech in these patients is often, but not always, absent. If it is present, the patients do not seem to have too much problems producing speech sounds but see van Balkom et al. (2011) for a more detailed report on that. Absence of speech in many MSS patients is due to a combination of a pathological facial and nasopharyngeal configuration, larynx configuration and resulting respiratory deficits, severe cognitive dysfunctions, potential oromotor dysfunctions and aggravated by sensorineural/conductive hearing loss. This is a summary of the literature citation 41 (Shaw et al., 2010) given by the authors who have chosen only to pick out of it the one aspect that suits the interpretation of their data without acknowledging the rest. Since many MSS patients die very early from severe breathing issues, it would be more accurate to call this a breathing affecting gene. If you want to argue in this manner, then you might as well argue that NFIX is a gene that impairs bipedal walking abilities as many of the patients have a distinct gait due to common hip dysplasia and kyphoscoliosis of the spine. Needless to say that both the larynx pathologies and the hip and spine issues are due to affected bone and connective/cartilaginous tissue rather than directly affecting complex functions such as speech and walking.

We thank the reviewer for their comment. It is important to clarify that the focus of this study is on the larynx and face due to one reason only: these are the most enriched organs within differentially methylated genes in AMHs. We tested all organs on Gene ORGANizer, and if the most enriched organs were different, we would have focused on them. This focus is not driven by a hypothesis, but rather a report of the strongest trends. Similarly, our focus on the vocal related aspects of NFIX is because of the general trend we report in this paper. Indeed, like all of the other genes we report, NFIX affects other anatomical and physiological aspects. We acknowledge and describe this in the pleiotropy chapter in “Validation of face and larynx enrichment in Gene ORGANizer” chapter, in Extended Data Table 7 and 8, and in the sentence: “suggesting that although they could have additional effects, their

effect on the voice is their most shared function”. We also describe the larynx as a breathing and swallowing organ, and not only a speech organ: “Importantly, the larynx is an organ which is primarily involved in breathing and swallowing in mammals”. In p.17 we state that the speech deficit in these patients is driven by various factors, and that the laryngeal anatomical changes are only partly responsible for the speech delay: “Most skeletal disease phenotypes that result from NFIX dysfunction are craniofacial, as NFIX influences the balance between lower and upper projection of the face. In addition, mutations in NFIX were shown to impair speech capabilities. The exact mechanism is still unknown, but is thought to occur **partly** through skeletal alterations to the larynx”, and again in: “these **laryngeal and facial changes** are thought to underlie **some** of the limited speech capabilities observed in various patients”, in line with the reviewer’s description. We agree with the reviewer that NFIX can also be described as breathing-affecting gene, as well as brain-affecting, spine-affecting and hip-affecting. Also, the reviewer writes that the effect of NFIX on the face and larynx is through alterations to the bone and connective/cartilaginous tissues. We agree with the reviewer and describe this in the manuscript: “...through skeletal alterations to the larynx” and “Mutations in NFIX were shown to cause the Marshall-Smith and Malan syndromes, whose phenotypes include various skeletal alterations such as hypoplasia of the midface, retracted lower jaw, and depressed nasal bridge”. We agree with the reviewer that speech is a complex trait that is affected by many genes and in many different ways. We do not argue otherwise in the paper.

Importantly, it seems that some comments by the reviewer come from an interpretation of the term X-affecting gene that is different than the way this term is usually used in genetic studies. By X-affecting gene we do not mean that this is the only function of the gene, that this is the main function of the gene, or that this is always the function of the gene. We simply mean that this gene was shown to affect X, directly or indirectly, sometimes or always. To clarify this point, we added the sentence: “Hereinafter, we refer to genes as affecting an organ if they have been shown to have a phenotypic effect on that organ in some or all patients where this gene is dysfunctional.” to the paper (p.9). Following the reviewer’s comment, we also added the Balkom et al. citation to the description of speech alterations in MSS patients.

At the same time, NFIX is associated with more than one syndrome – and speech issues are reported for Marshall-Smith syndrome only Martinez et al. (2015). The Sotos-like syndrome patients do not have laryngomalacia or respiratory issues. If NFIX really were a speech-impairing gene, then this would have to have that effect on all patients who are affected by NFIX mutations.

Indeed, not all patients exhibit the same spectrum of phenotypes. This is often the case in such disorders, as a gene’s function is affected by other genes as well as by environmental factors. We only refer to speech with regard to MSS patients (e.g., “many cases of laryngeal malformations in the Marshall-Smith syndrome have been reported. Some of the patients exhibit positional changes to the larynx” and Extended Data Table 8). See comment above for the definition of X-affecting.

In the same vein of arguing, it is worth mentioning the following: ACAN (single locus, single gene) was linked to three different phenotypes, of which one reports a hoarse voice in 3 patients, whereas the rest of the phenotypes linked to 100+ patients for whom there is no mention of any voice quality alterations at all. The most commonly reported phenotypical traits for ACAN mutations are short stature, face morphology variations, respiratory distress and waddling gait. Hence the same issues of whether ACAN is a bipedal gait affecting gene applies as well whereas it actually is – like NFIX and SOX9 regulating chondrification and ossification processes. Furthermore, ACAN’s association with short stature (mostly of the

primordial dwarfism type in contrast to e.g. achondroplasia type dwarfisms) likely results in a scaled down vocal tract size which is generally associated with a high pitched voice. This might mean that you can have a non-altered larynx anatomy and still end up with a variation in voice quality. Yet this is not addressed either. I just detailed this here as a demonstration on just how difficult it is to actually support your bold statements about what your genes are actually doing or not.

See comment above

Lines 366-370

Sentence starting on 366-368 please remove it, you cannot support that statement.

The observation that mutations in NFIX result in speech impairment in some patients was reported by several previous studies, which we cite.

Line 368- 370

Sentence starting “To investigate...” at the end, please alter to “... examined its clinical relevant skeletal phenotypes”. Please make it clear here that the following are non-normal phenotypes.

Following the reviewer’s suggestion, we added “clinical” to this sentence

I can accept the statements in lines 375-386.

P. 18, line 387, please loose the interestingly at the beginning of the sentence.

We changed “interestingly” to “notably”

Section lines 388-392

This section is not acceptable in this form.

First, I read the reports on MSS laryngomalacia and the literature on laryngomalacia in detail again. There are many different types of laryngomalacia and most of them involve predominantly the size, shape and placement of the epiglottis and/or the arytenoids in relation to the epiglottis, particularly the soft tissues connecting them with the epiglottis. I did not find any literature which directly stated that the arytenoids are frequently malformed in Marshall-Smith syndrome. So please stay within what is known and report that some of the patients suffering from MSS do have an abnormal laryngeal cartilage framework and soft tissues which create severe respiratory distress and leave the whole speculation about arytenoids and vocal folds out of this. If you want to say something about NFIX regulation of the larynx, then state that it is likely due to the alteration this causes to the tissues NFIX codes for and that it can have a relationship with the facial structures being altered, that would be acceptable and would match better with the explanations you are putting forward for your adult methylation patterns observed. The rest of the discussion is ok as it is.

From Cullen et al. (citation 53): “Hassan et al. in 1976 reported a child as having an abnormal configuration of the larynx with an increased length of the arytenoids.”, and “In 1974 Visveshwara et al. reported... poor visualisation of the vocal cords and arytenoid folds”. It is unclear if this is frequent or not, as these phenotypes are not always examined, but it has nevertheless been reported that the arytenoids and vocal folds were affected in some patients. To clarify that this observation is more rare, we changed the sentence to: “Some of the patients exhibit positional changes to the larynx, changes in its width, and, **more rarely**, structural alterations to the arytenoid cartilage – the anchor point of the vocal folds, which controls their

movement”

Methods section

Validation of face and larynx enrichment in GeneORGANizer P 75, line 1470-75

Please remove the text “the DMGs which affect the voice” from this section. Otherwise I will insist on you giving a percentage separately for voice in line 1743 as well, not just for larynx, pelvis ect. I recommend replacing this with:

genes affecting the larynx and vocal tract anatomy and summarising these as 65.

Indeed, we now realize that this sentence could be interpreted as if we suggest that the differential methylation is affecting the voice, while we meant to say that it occurs in genes that are known to affect the voice. Therefore, we changed it to: “DMRs in voice-affecting genes.

P. 80 lines 1584-1602

First I would like to thank the authors that they acknowledge the importance of the breathing and feeding functions of the supralaryngeal vocal tract. I find it therefore a bit puzzling that this importance is then not presented accordingly in this section of the text. I therefore recommend the following:

Lines 1597-1598, loose the “Importantly” at the beginning of this sentence, then put it in the position of line 1584, at the beginning of this section.

We moved “importantly” to the beginning of this section, as suggested by the reviewer.

Line 1584, this lead in sentence to the section is misleading, there is no definition given for “vocal phenotype”. Vocal phenotype does not have any meaning in relation to evolution but it has a meaning in clinical studies. This needs to be addressed here. The term “brain disorder” is also not defined at all, neither is vocal alteration. I would leave this out or be more precise. More precision comes about by talking about modern human voice quality variation in pitch, timbre and volume and to stress that these are very individual.

By brain disorder we mean any clinical disorder that affects the brain, and changed the sentence to: “Some brain-related disorders (i.e., clinical disorders that affect the brain) result in vocal alterations to the voice”. “Vocal alteration” was changed to “alteration to the voice (either in its pitch, timbre, volume or range)”

Lines 1585-1587

I completely disagree with this sentence. Please either make it very clear what you mean by brain disorders, vocal phenotypes and voice alterations or delete it.

Following the reviewer’s comment, we removed this sentence.

Lines 1586-1487

The sentence on links between vocal tract anatomy and voice quality variation – the link is easier to establish – agreed but that concerns the link between the varied vocal tract anatomy and the speech sound output or voice quality, not the genes.

This was removed from the text, as suggested by the reviewer.

Lines 1590-1592

This sentence is not supported by any statement – please either delete it or provide the evidence for this.

Following the reviewer's suggestion, we added citations to HPO and Gene ORGANizer, which describe the phenotypes associated with these disorders.

Lines 1593-1596

This sentence reads as a fragment as the second half of it has no context to anything else – the families reported here are not introduced anywhere and nothing is in context.

Following the reviewer's comment, we changed "individuals from the same family" to "individuals from the same family as the individual with the disorder"

Lines 1600-1602

This sentence is again not indicating why alteration to voice quality would have an impact on speech evolution. Also still – I am really sorry about this – but I could not follow how a gene reported in Gene ORGANizer does provide the proof that any given gene does have a direct influence on voice quality. Either delete the sentence or explain it again.

We do not refer to speech evolution in this section, only to speech anatomy. With regard to Gene ORGANizer, gene-phenotype links that appear in this database are only in genes which were shown to affect an organ in an individual where the gene is dysfunctional. In other words, if a gene is linked to the larynx in Gene ORGANizer, it means that this gene was shown to phenotypically affect the larynx in individuals where the gene is dysfunctional. Importantly, this is not necessarily a direct effect. To clarify this, we changed the sentence to: "the genes reported by Gene ORGANizer and HPO were specifically associated with voice alterations, directly or indirectly, suggesting that although they could have additional effects, their effect on the voice is their most shared function."

References

Martinez, F. et al. (2015). Novel mutations of NFIX gene causing Marshall-Smith syndrome or Sotos-like syndrome: one gene, two phenotypes. *Translational investigation* 78(5): 533-539

Van Balkom I. et al. (2011) Development and behaviour in Marshall–Smith syndrome: an exploratory study of cognition, phenotype and autism. *Journal of Intellectual Disability Research* 55(10):973–987

Reviewers' Comments:

Reviewer #2:

Remarks to the Author:

The authors have addressed my concern regarding the specificity of the signal using the permutations as suggested. I think the manuscript has really improved since the first submission. I would have a final pledge to fix:

The main finding as in the title comes from an enrichment of DMRs on the modern human lineage in genes affecting vocal and facial anatomy. An enrichment is obviously relative to a baseline and currently it is difficult for the reader to understand from the main text to which baseline which enrichment belongs. For example the authors write: "Using Gene ORGANizer, we found 11 organs that are over-represented within the 588 AMH derived DMGs ..." One needs to make clear to what this overrepresentation relates. I assume these are all genes in Gene ORGANIZER or maybe these are all genes in Gene ORGANizer that have been assessed as DMGs? This needs to be made clear throughout the manuscript so that it is clear on which assumptions the enrichment is based.

Reviewer #4:

Remarks to the Author:

Dear Editor, dear authors,

Thank you very much for taking on board the last round of comments I made to the manuscript "Genes Affecting Vocal and Facial Anatomy Went Through Extensive Regulatory Divergence in Modern Humans" (NCOMMS-19-06061B).

I am glad to see that the main issues I had with translating highly specialised expert knowledge into a more accessible and more precise and/or differentiated presentation have been addressed.

Over and all, the explanation the authors have kindly and patiently provided have also clarified where I had problems following their arguments and I find the solutions the authors provide to address my concerns acceptable and in this form, I gladly accept the manuscript. I think by expanding on the complexity of gene interactions, your results actually do become stronger and more compelling. I particularly like the addition of the extended data table 8.

There are a very few minor items that might be considered. However, I for my part do not need to see the implementation of these again and would leave the decision of whether these should be added or not to the editor:

P. 8, line 171: Thank you very much for the introduction of the GO terms as codes of tools being used, this has clarified a lot of issues I had with your use of terminology and the explanations you provide for Gene ORGANIZER. Reading this now makes the use of it much clearer. If I may - for further support of the readers - you could consider referencing both your extended data table 3 and the phenotype descriptions of each of your genes from the supplementary material, those really helped making sense of this section.

Pp. 19(427), 28(630), 54(999) and 68(1313): In these four locations, you use the term "chimp", whereas throughout the rest of the manuscript, the term "chimpanzee" is used. I would suggest to exchange chimp with chimpanzee in these places as well.

Many thanks again for the opportunity to read this manuscript.

Reviewer #2 (Remarks to the Author):

The authors have addressed my concern regarding the specificity of the signal using the permutations as suggested. I think the manuscript has really improved since the first submission. We thank the reviewer for their constructive comments throughout this process.

I would have a final pledge to fix:

The main finding as in the title comes from an enrichment of DMRs on the modern human lineage in genes affecting vocal and facial anatomy. An enrichment is obviously relative to a baseline and currently it is difficult for the reader to understand from the main text to which baseline which enrichment belongs. For example the authors write: “Using Gene ORGANizer, we found 11 organs that are over-represented within the 588 AMH derived DMGs ...” One needs to make clear to what this overrepresentation relates. I assume these are all genes in Gene ORGANIZER or maybe these are all genes in Gene ORGANizer that have been assessed as DMGs? This needs to be made clear throughout the manuscript so that it is clear on which assumptions the enrichment is based.

We thank the reviewer for their suggestion. Indeed, this point needed clarification. Following the reviewer’s comment, we added the following clarification sentence to the beginning of the Gene ORGANizer analysis: “An enrichment or depletion in Gene ORGANizer is detected if the group of genes analyzed shows a significant deviation in the organs they are known to phenotypically affect, compared to the rest of the genome.”

Reviewer #4 (Remarks to the Author):

Dear Editor, dear authors,

Thank you very much for taking on board the last round of comments I made to the manuscript "Genes Affecting Vocal and Facial Anatomy Went Through Extensive Regulatory Divergence in Modern Humans" (NCOMMS-19-06061B).

We thank the reviewer for their thorough and thoughtful comments throughout this process.

I am glad to see that the main issues I had with translating highly specialised expert knowledge into a more accessible and more precise and/or differentiated presentation have been addressed. We are glad to see that the reviewer is satisfied with this multi-faceted translation process.

Over and all, the explanation the authors have kindly and patiently provided have also clarified where I had problems following their arguments and I find the solutions the authors provide to address my concerns acceptable and in this form, I gladly accept the manuscript. I think by expanding on the complexity of gene interactions, your results actually do become stronger and more compelling. I particularly like the addition of the extended data table 8.

We thank the reviewer.

There are a very few minor items that might be considered. However, I for my part do not need to see the implementation of these again and would leave the decision of whether these should be added or not to the editor:

P. 8, line 171: Thank you very much for the introduction of the GO terms as codes of tools being used, this has clarified a lot of issues I had with your use of terminology and the explanations you provide for Gene ORGANIZER. Reading this now makes the use of it much clearer. If I may - for further support of the readers - you could consider referencing both your extended data table 3 and the phenotype descriptions of each of your genes from the supplementary material, those really helped making sense of this section.

We thank the reviewer for this suggestion. Indeed, extended data table 3 is now referenced also in the supplementary material, in the section “Comparison to previous reports”.

Pp. 19(427), 28(630), 54(999) and 68(1313): In these four locations, you use the term "chimp", whereas throughout the rest of the manuscript, the term "chimpanzee" is used. I would suggest to exchange chimp with chimpanzee in these places as well.

We agree. Following the reviewer’s comment, we changed “chimp” to “chimpanzee”.

Many thanks again for the opportunity to read this manuscript.